# Integrative taxonomy clarifies the evolution of a cryptic primate clade

Global biodiversity is under accelerating threats, and species are succumbing to extinction before being described. Madagascar's biota represents an extreme example of this scenario, with the added complication that much of its endemic biodiversity is cryptic. Here we illustrate best practices for clarifying cryptic diversification processes by presenting an integrative framework that leverages multiple lines of evidence and taxon-informed cut-offs for species delimitation, while placing special emphasis on identifying patterns of isolation by distance. We systematically apply this framework to an entire taxonomically controversial primate clade, the mouse lemurs (genus *Microcebus*, family Cheirogaleidae). We demonstrate that species diversity has been overestimated primarily due to the interpretation of geographic variation as speciation, potentially biasing inference of the underlying processes of evolutionary diversification. Following a revised classification, we find that crypsis within the genus is best explained by a model of morphological stasis imposed by stabilizing selection and a neutral process of niche diversification. Finally, by clarifying species limits and defining evolutionarily significant units, we provide new conservation priorities, bridging fundamental and applied objectives in a generalizable framework.

It is well understood that Earth is facing a human-caused biodiversity extinction crisis[1–3]. What is less appreciated is that there are an untold number of species threatened with extinction that have yet to be recognized by science[4]. Two of the most critical factors contributing to this paradox are that the majority of extant species occur in remote areas where fieldwork is challenging[5] and that many of these species are 'cryptic' in the sense that while being genetically distinct, they are phenotypically indistinguishable to human eyes[6]. The accurate characterization of species[7], especially cryptic ones[8,9], is crucial for a comprehensive understanding of the biotic and abiotic forces that drive and maintain diversification[10,11], given that estimates of species richness, abundance and distribution are fundamental to macroevolutionary and ecological studies[12–14]. Species definitions are ultimately the foundation for conservation policies and action[7,15], and the accurate characterization of biodiversity is therefore a vital first step for comprehending and addressing the magnitude of the escalating extinction crisis.

Yet the delineation of biodiversity into species presents substantial challenges both operationally and philosophically[16]. Phylogenetic lineages belong to a diversification continuum, ranging from interconnected populations at one end to reproductively isolated species at the other. This makes their assignment to discrete categories difficult, particularly when species occur in allopatry and/or sampling is limited[17–19]. In addition, the concept of 'species' still lacks a widely accepted definition in the scientific community[20,21]. There is, however, increasing agreement among biodiversity investigators that the means for defining species must integrate multiple aspects of organismal phylogeny, geography, morphology and behaviour[22–24].

Mouse lemurs (genus *Microcebus*, family Cheirogaleidae) are a clade of cryptic primates endemic to Madagascar whose taxonomic treatment, like that of other lemur genera, has been criticized for overestimation of actual species diversity, also referred to as taxonomic inflation[25–27]. Though the diversity within the genus went unrecognized

✉e-mail: tob.velst@posteo.de; gabriele.sgarlata@gmail.com; dominik.schuessler@posteo.de; jordi.salmona@ird.fr

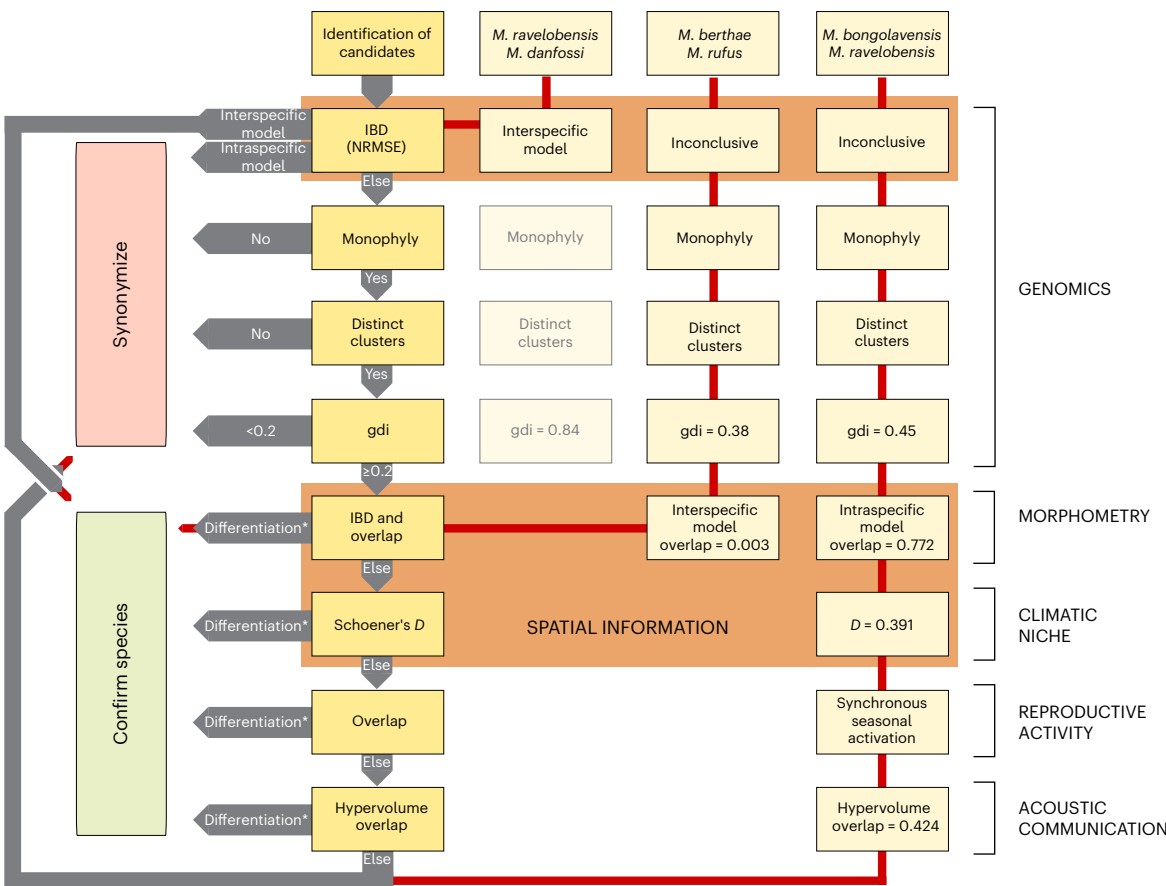

**Fig. 1 | Workflow for integrative taxonomy of cryptic taxa and its illustration in the genus *Microcebus*.** We first test whether genetic distances between candidates clearly reject or conform to an intraspecific model of isolation by distance, using a heuristic based on normalized root mean square error (NRMSE) distributions. If neither is the case, we test whether pairs of sister candidates are reciprocally monophyletic, form distinct genetic clusters and exhibit a genealogical divergence index (gdi) above or equal to 0.2. Failure to pass one of these criteria is sufficient to reject status as distinct species. If tests are passed, we explore whether candidates exhibit substantial differentiation in at least one other taxonomic character (morphometry, climatic niche, reproductive activity,

acoustic communication) that cannot be attributed to ecological flexibility, plasticity or similar factors (indicated by asterisks). If so, and only then, candidate species are confirmed. Three examples of pairs of candidate species in the genus *Microcebus* are presented to illustrate the workflow. Red arrows indicate the delimitation procedure. Additional taxonomic characters are not restricted to the examples given here. Brown boxes indicate which tests consider spatial variation. In principle, other taxonomic characters would benefit from being analysed in a spatial context as well. Details on how tests were conducted and differentiation was quantified can be found in the Methods.

for decades, largely due to cryptic morphology and allopatric distributions, the number of described species drastically increased from four to 25 within 2.5 decades following the introduction of mitochondrial DNA barcoding methods[28–30]. Accurate classification of the genus' diversity is urgently needed to enable effective conservation action and diversification research, given that many *Microcebus* species, along with most of the island's endemic mammals, are threatened with extinction due to habitat loss and degradation[31,32]. Moreover, the mechanisms that underlie the rapid evolutionary radiation of the genus *Microcebus* remain elusive[33]. For instance, it is presently unknown why and how *Microcebus* species diversified into distinct genetic clades in virtually all forest habitats across the island, while showing relatively little morphological divergence.

Here we present a practical framework following Padial et al.[34] that integrates multiple lines of evidence to distinguish interconnected populations from separately evolving metapopulation lineages (that is, distinct species sensu de Queiroz[21]) along the speciation continuum (Fig. 1). We prioritize genomic analyses to detect structure, differentiation and gene flow among hypothesized sister species (hereafter referred to as candidates), placing particular emphasis on identifying isolation by distance (IBD). To do so, we introduce a novel approach that takes genome-wide variation into account and tests whether genetic

distances between candidate individuals deviate from a model of intraspecific spatial structure (Extended Data Fig. 1). Instead of relying on arbitrary cut-offs to distinguish intra- from interspecific divergence, we derive genus-specific thresholds from variation observed among fragmented and continuous populations of two *Microcebus* species (*M. lehilahytsara* and *M. tavaratra*) with extensive sampling and well-characterized patterns of gene flow and IBD[35–39]. We also use additional lines of evidence to validate candidate species if an intraspecific model cannot clearly be rejected but genomic differentiation is identified. To do so, we compile available data on morphometry, climatic niche, reproductive activity and acoustic communication and quantify overlap in these traits, while also extending the use of our IBD-based approach to assess whether morphometric variation is structured in space. Accordingly, hypothesized species that do not show significant discontinuity in patterns of IBD are only confirmed if there is convincing evidence to reject the null hypothesis of a single-species model.

We systematically apply this framework to the genus *Microcebus*, including all 25 named species with extensive geographic sampling, thus accounting for both inter- and intraspecific variation within the clade. We demonstrate that its application enhances understanding of cryptic diversifications, temporal evolution of habitat and climatic niche and their combined impacts on morphological stasis through

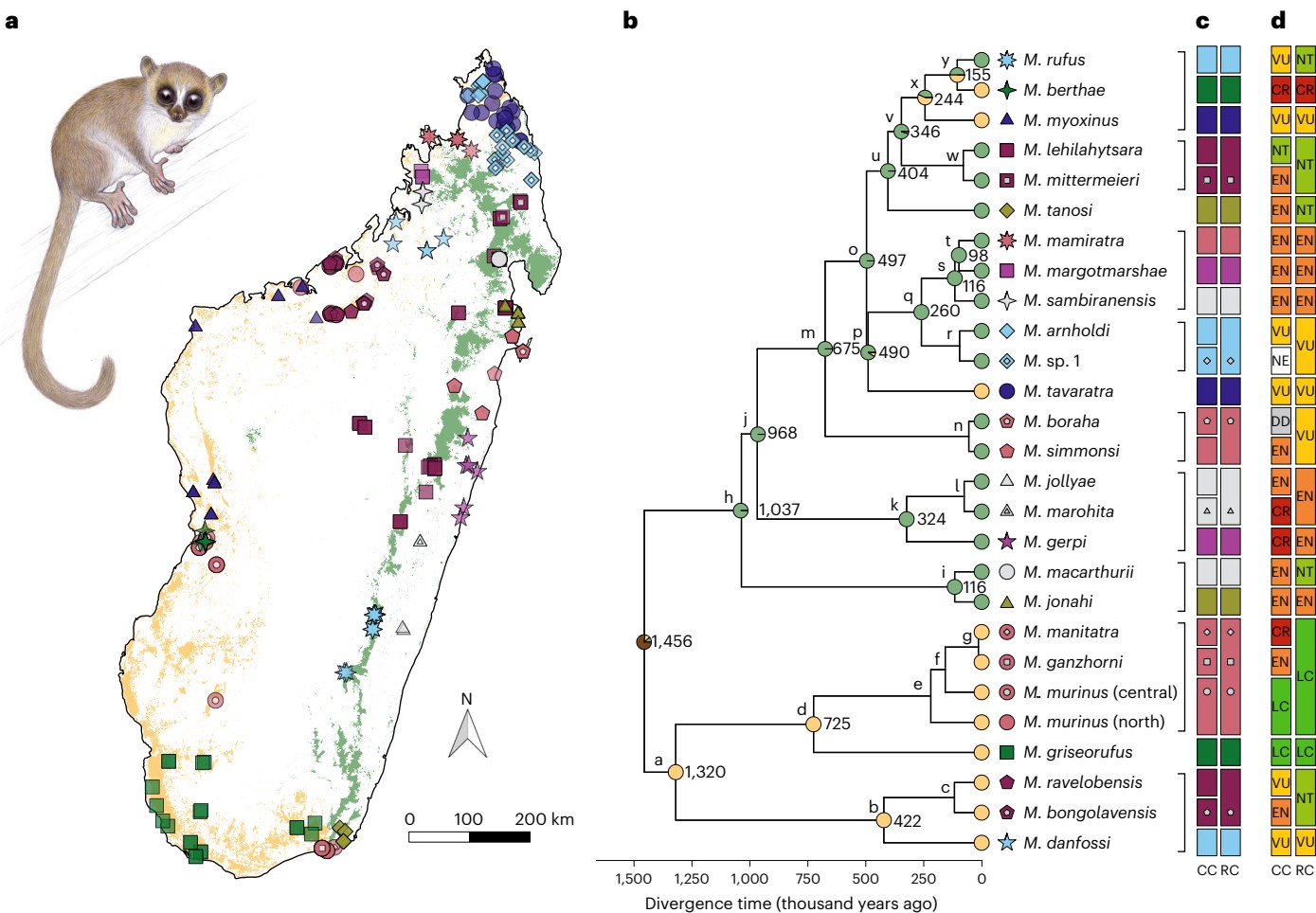

**Fig. 2 | Island-wide taxogenomics of the cryptic *Microcebus* radiation. a**, Map of genotyped *Microcebus* species (symbols correspond to **b**). Dry and humid forests are represented in yellow and green, respectively. Illustration represents *M. jonahi* (illustration copyright Stephen D. Nash; used with permission). **b**, *Microcebus* phylogeny with divergence times and ancestral habitats (node pies; yellow, dry; green, humid; brown, dry and humid). Candidate groups to which our delimitation framework was applied are indicated by black brackets. White centres in species symbols represent synonymized candidates following the revised classification shown in **c**. Divergence times among synonymized candidates are not reported. Nodes are labelled by lowercase letters for reference to downstream analyses. **c**, Comparison of the current (CC, 25 described and one putative species) and revised (RC; 19 species) *Microcebus* classification. **d**, Recommended changes in International Union for Conservation of Nature (IUCN) species conservation status after taxonomic revision (NE, not valuated; DD, data deficient; LC, least concern; NT, near threatened; VU, vulnerable; EN, endangered; CR, critically endangered).

time. Additionally, we highlight the consequences for conservation status and identify phylogeographic conservation units. Our work sheds light on the taxonomy and diversification of the genus *Microcebus*, while also providing an extended, generalizable framework for integrative taxonomy that will benefit studies on global biodiversity across phylogenetic lineages.

## Results and discussion

### An integrative framework for taxonomic re-evaluation

We demonstrate the applicability of our framework, treating the 25 currently recognized *Microcebus* species and one putative species (*M.* sp. 1 (ref. 38)) as candidates. By inferring a well-supported phylogeny from restriction site associated DNA (RAD) markers of 208 samples across all species of the GENUS (median = seven samples per species), we identified nine groups of allopatric sister candidate species within which pairwise delimitation tests were conducted (274 samples, median = ten samples per candidate; Fig. 2a,b and Supplementary Figs. 1–6). We propose the synonymization of seven candidates (*M. bongolavensis*, *M. boraha*, *M. ganzhorni*, *M. manitatra*, *M. marohita*, *M. mittermeieri* and *M.* sp.1) across six groups to their closest relatives, deflating the taxonomy of mouse lemurs from 26 to 19 species

(Fig. 2c and Supplementary Table 1). This is mostly due to strong influence of geographic structure on genomic differentiation, identified gene flow and/or low differentiation in morphometry, climatic niche, reproductive activity and acoustic communication (Fig. 3 and Extended Data Figs. 2–10; discussed in detail in Supplementary Results and Discussion: Species delimitation and diagnosis).

Here we highlight results for three exemplary candidate pairs, *M. berthae* vs *M. rufus*, *M. ravelobensis* vs *M. bongolavensis* and *M. ravelobensis* vs *M. danfossi*, to illustrate contrasting decision-making (that is, synonymizing vs retaining candidate species) in our framework (Fig. 1). Our IBD-based test statistic indicates that genetic distances between *M. danfossi* and *M. ravelobensis* are significantly higher than those found within taxa, even at similar geographic distances, clearly rejecting an intraspecific model of IBD and confirming their distinction as valid species (Figs. 1 and 3c, Extended Data Fig. 10d and Supplementary Table 2). This is not the case for the other two candidate pairs (Figs. 1 and 3c, Extended Data Fig. 2d and 10d and Supplementary Table 2) even though they exhibit clear genomic differentiation, indicated by reciprocal monophyly, distinct clusters in admixture analyses and intermediate mean genealogical divergence indices (gdi$_{M.ber./M.ruf.}$ = 0.38; gdi$_{M.bon./M.rav.}$ = 0.45; Fig. 3a,b,d and

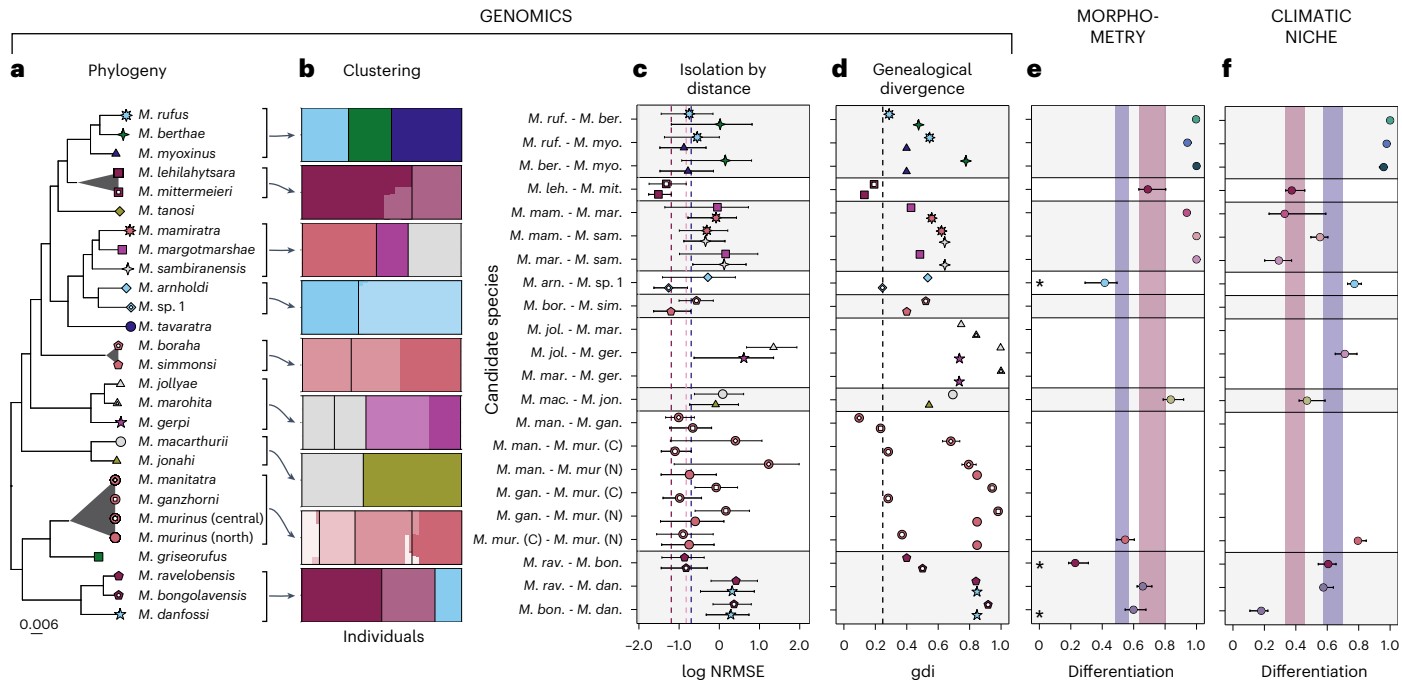

**Fig. 3 | Summary of species delimitation analyses in the genus *Microcebus*.**
**a**, Maximum likelihood phylogeny with non-monophyly indicated by triangles. Scale is substitutions per site. **b**, Admixture proportions (*y* axis), where the number of a priori clusters *K* equals the number of candidate species; candidate species are separated by black bars and ordered as in **a**. **c**, NRMSE distributions of isolation by distance (log scale) with 0.05 and 0.95 quantiles; symbols indicate focal taxon for calculation of within-candidate IBD; red, pink and blue dashed lines indicate 0.95 quantiles of NRMSE distributions based on IBD within *M. lehilahytsara*, *M. mittermeieri* and continuous *M. tavaratra* populations, respectively (Methods). **d**, Genealogical divergence index (gdi) with 95% highest posterior density interval based on a coalescent model of 6,000 loci and two individuals per species (one individual for *M. marohita*); symbols refer to **a**

and indicate which taxon's *θ* was used for estimation; taxon names refer to the first three letters of the candidate species epithet; the dashed line indicates threshold below which candidates are considered synonyms. **e**, Morphometric differentiation (1 − maximum hypervolume overlap) and 95% confidence intervals (CI); asterisks indicate fit to a model of intraspecific character variation, precluding the interpretation of the differentiation signal (Supplementary Table 5). **f**, Climatic niche differentiation (1 − *D* and 95% CIs. In **e** and **f**, red and blue areas represent 95% CIs of differentiation between *M. lehilahytsara* and *M. mittermeieri* and among fragmented *M. tavaratra* populations, respectively. Empty rows indicate a lack of data. Sample sizes per species for **c**, **e** and **f** are given in Supplementary Tables 2, 4 and 6, respectively.

Extended Data Figs. 2b,c,e and 10b,c,e and Supplementary Table 3). Therefore, we analysed additional lines of evidence to investigate whether they possess sufficient character differences to be considered distinct species. Whereas morphometric variation among *M. bongolavensis* and *M. ravelobensis* can be explained by an intraspecific model of IBD, such a model is clearly rejected for *M. berthae* and *M. rufus* (Fig. 3e, Extended Data Figs. 2f and 10f and Supplementary Tables 4 and 5). Similarly, climatic niche overlap (Schoener's *D*) between *M. bongolavensis* and *M. ravelobensis* (quantile range $Q_{0.05}$–$Q_{0.95}$: 0.34–0.46) resembles that found among populations of a similarly widely distributed mouse lemur species, *M. tavaratra* ($Q_{0.05}$–$Q_{0.95}$: 0.30–0.43) but is zero between *M. berthae* and *M. rufus* (Fig. 3f, Extended Data Figs. 2g and 10g and Supplementary Table 6). Finally, *M. bongolavensis* and *M. ravelobensis* exhibit similar timing of seasonal reproductive activity, whereas *M. berthae* and *M. rufus* show stable differences in female reproductive activation (Extended Data Figs. 2h and 10h). On the basis of these findings, we propose synonymizing *M. bongolavensis*, which was initially described based on three diagnostic sites in two mitochondrial loci and minor morphometric differentiation[40] under the senior name *M. ravelobensis*. Conversely, *M. berthae* and *M. rufus* should be maintained as distinct species due to their genomic, morphometric and niche differentiation.

Our findings demonstrate how detailed genomic analyses coupled with multivariate investigation of additional taxonomic characters enable consistent classification in a cryptic radiation. They confirm previous concerns of taxonomic inflation in this genus and provide a foundation for a wider application in other animal taxa with controversial taxonomies, such as the morphologically cryptic lemur

genera *Avahi*, *Cheirogaleus*, *Hapalemur* and *Lepilemur*, in which species have mostly been described based on the phylogenetic species concept[25,41–44]. The genus *Lepilemur*, for instance, also comprises 25 described species[45], with two already proposed to be synonymized (*L. milanoii* and *L. mittermeieri*)[39,46]. Finally, the systematics of various medium-sized vertebrates have been the topic of recent debates about species delimitation[47–50], illustrating appropriate applications of a systematic taxonomic approach beyond the lemurs of Madagascar.

Whereas our framework provides a generalizable way to integrate genomic data with the analysis of additional lines of evidence, we are aware that taxon-specific idiosyncrasies can present considerable challenges for application to other taxa. For example, gathering comprehensive data across taxonomic characters and candidate species may not be feasible, for instance, when associated populations are difficult to survey or encompass large distributions. Nonetheless, we emphasize that distinguishing intraspecific clinal variation and interspecific divergence, particularly in cryptic radiations, requires broad geographic sampling and multiple lines of evidence. The results of taxonomic studies that rely solely on a few genes or limited sampling for species delimitation should be considered provisional. We acknowledge that additional data are also required to definitively resolve the taxonomy of several *Microcebus* candidate groups and validate our conclusions, but systematically applying our framework across the entire genus yielded informed hypotheses and identified key areas where further sampling is necessary. Specifically, future work could, among others, be directed at the poorly studied *M. jollyae* and at addressing sampling gaps for *M. jonahi*, *M. macarthurii*, *M. murinus* and *M. simmonsi* (Supplementary Results and Discussion: Species delimitation and diagnosis).

In each application of the framework, it has to be decided which taxonomic characters are relevant for species delimitation, subsequent to the detection of genomic differentiation. For instance, reproductive traits (for example, seasonality, baculum morphology) may be better suited than morphometric traits related to body size in clades with phenotypic plasticity, whereas climatic niche dissimilarity may be misleading if taxa are ecologically flexible or constrained by geographic barriers instead of climate. Similarly, delimitation thresholds are subject to the degree of character variation in each system and have to be selected carefully. If available, we advocate the use of 'benchmark' taxa, that is, species with well-characterized population structure (for example, *M. tavaratra* herein; use of sympatric species in Tobias et al.[51]) and to which differentiation of candidates can be compared. Even with these caveats, our framework can serve as a heuristic model to facilitate consistent and quantitative classification of taxonomically challenging groups along the speciation continuum, while overcoming the oversplitting tendencies of the PSC and multispecies coalescent (MSC) approaches and potential biases from incomplete sampling and geographic clines in character variation[52–56].

## Coherent taxonomy informs evolution and conservation

The systematic application of our integrative framework to the genus *Microcebus* revealed a general tendency of misinterpreting geographic structure as interspecific variation. The proposed taxonomic changes have implications considering the geographic distributions of several species and their associated ecological correlates. For instance, *M. lehilahytsara*, once considered a highland specialist[57], is now demonstrated to be the second most widespread species, occurring also at low elevations[36]. Similarly, the microendemic, potentially specialized and threatened *M. ganzhorni* and *M. manitatra* are now best placed as synonyms of the most widespread generalist *M. murinus*. In the following sections, we therefore use the updated taxonomy (presenting a coherent characterization of patterns of species diversity) to identify the evolutionary processes underlying the cryptic diversification of this genus. We infer the spatiotemporal context of its diversification and test models of climatic niche and morphological evolution. Such models rely on the assumption that the species considered are accurately delineated[12], yet potential biases from treating divergent populations as distinct species remain to be assessed. Finally, by providing conservation status recommendations for all revised *Microcebus* species, we demonstrate the impacts of taxonomic inflation on conservation management.

## A Pleistocene diversification to dry and humid biomes

We estimated divergence times of the *Microcebus* phylogeny under an MSC model using a mutation rate calibration based on external evidence from per-generation de novo primate mutation rates, as no internal fossil calibrations are available for Lemuriformes. Using this method, we infer that the genus diverged from its sister lineage, the genus *Mirza*, about 2.3 million years (Ma) ago and started diversifying during the Mid-Pleistocene (~1.5 Ma ago; Fig. 2b, Supplementary Figs. 9–14 and Supplementary Table 7). Such a temporal framework (< 2 Ma ago) is supported by other MSC studies[36,37,58,59] and suggests that the diversification of the genus *Microcebus* fits a model of allopatric speciation in response to climatic fluctuations (that is, glacial–interglacial cycles). This interpretation agrees with studies that have posited that closed-canopy ecosystems converted to open vegetation during the Pleistocene in different areas of the island[60,61], forcing lineages to track forest habitats that shifted in elevation or to retreat to humid refugia[62,63]. Notably, the inferred divergence times differ markedly from dates obtained from concatenated likelihood analyses using fossil calibrations that placed the diversification of the genus at about 8–10 Ma ago during the Late Miocene[64–67]. This discrepancy may be expected, however, given the tendency of concatenated analyses to inflate divergence times by not accounting

for variation in genealogical histories[68], especially when using external and phylogenetically distant fossil calibrations[69].

Our phylogeny indicates that the earliest divergence among extant *Microcebus* species occurred between the *M. murinus* group, *M. griseorufus* and the clade comprised of *M. bongolavensis*, *M. danfossi* and *M. ravelobensis*, on the one hand, and all other *Microcebus* species, on the other. This agrees with Everson et al.[67] and Weisrock et al.[70] but contrasts with earlier multilocus studies[29,58,64,65,71] and recent work modelling reticulated evolution on orthologue genes[72] (Supplementary Results and Discussion: Divergence time estimation for details). Through ancestral state reconstruction, we show that this early bifurcation in the genus *Microcebus* coincides with habitat differentiation in humid eastern and dry western forests (Fig. 2b, Supplementary Fig. 15 and Supplementary Table 8), which has been shown for other lemur taxa as well (for example, the genus *Propithecus*[73]). Modelling ancestral habitats with finer-scale classifications also supports the major distinction between humid and dry conditions, while highlighting the evolution of more specialized niches (for example, in subhumid and arid habitats; Supplementary Figs. 16 and 17). At least two reversions to drier habitat occurred in the humid forest clade (Fig. 2b and Supplementary Fig. 15; *M. berthae*, *M. myoxinus*, *M. tavaratra*), indicating that ancestral humid forest-associated *Microcebus* lineages retained the evolutionary potential for niche shifts from humid to dry habitats. It has been suggested that bioclimatic disparities between eastern and western Madagascar may have promoted species formation, for example, by parapatric speciation through ecogeographic constraints[74,75]. It remains uncertain, however, whether the colonization of different habitats caused the early divergence in the genus or occurred subsequently.

## Morphological stasis and neutral climatic niche evolution

To identify the processes associated with lineage diversification, we reconstructed changes in morphometric and climatic niche overlap along the *Microcebus* phylogeny and compared the observed correlation of overlap values and node age to expectations given by trait simulations under different evolutionary models. We do not find a significant correlation (Spearman's correlation coefficient $r_s$) between node age and morphometric hypervolume overlap ($r_s = -0.015$, $P = 0.96$; Fig. 4a), using seven variables related to head and foot morphology that exhibit few missing data across species (Supplementary Table 9) and good reproducibility across researchers[76]. This indicates a temporal pattern of modest evolutionary change, in agreement with the concept of morphological stasis[77]. Various evolutionary processes have been proposed to explain stasis, including long periods of stabilizing selection[78,79] and neutral evolution with genetic and developmental constraints[80,81]. Our simulation-based analyses and cross-validation tests reveal that the observed relationships are better explained by a stabilizing selection (OU) than a neutral random walk (BM) or an early-burst (EB) model of evolution (Fig. 4b and Supplementary Fig. 18; Supplementary Results and Discussion: Morphological stasis and neutral climatic niche evolution contain details). Our results agree with studies of other taxa, which found substantial support for the OU model when investigating morphological stasis or evolution[82–86]. Comparing expectations from Lande's[87] stabilizing selection model with the inferred OU parameters and empirical estimates of morphological heritability in *M. murinus* provides further evidence that stabilizing selection is a reasonable model to explain morphological stasis in the genus *Microcebus* (Supplementary Results and Discussion: Morphological stasis and neutral climatic niche evolution).

Similarly, we do not observe a significant correlation between node age and two measures of climatic niche overlap (Schoener's *D* $r_s = 0.10$, $P = 0.69$; hypervolume overlap: $r_s = -0.268$, $P = 0.28$; Fig. 4c and Supplementary Fig. 29), using eight bioclimatic variables considered ecologically meaningful for *Microcebus* species (Supplementary Table 10)[88,89]. The simulation-based procedure reveals that a BM model

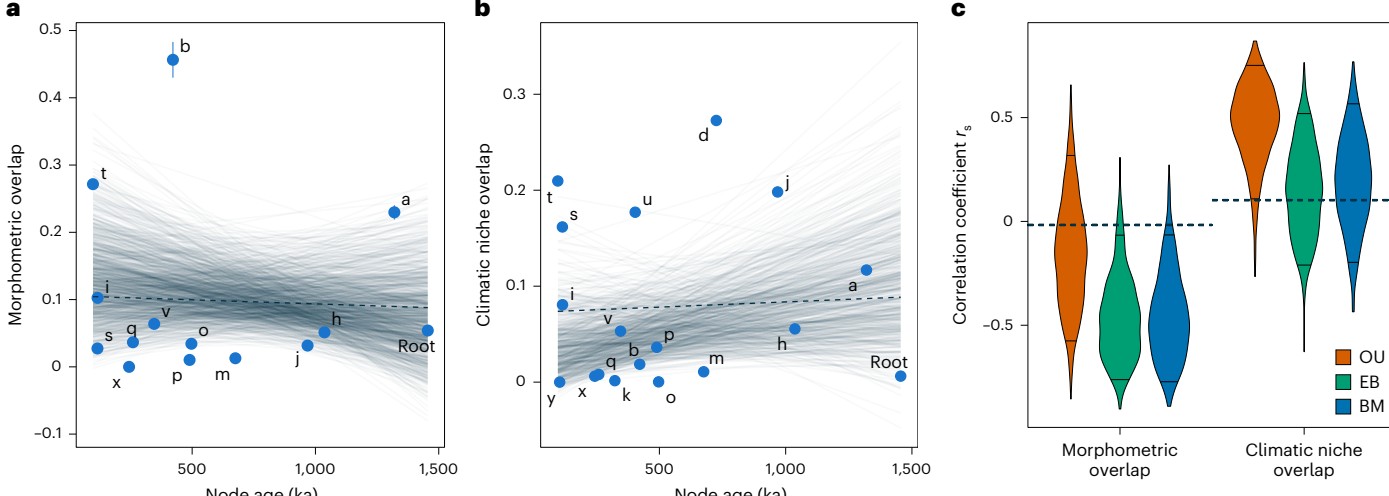

**Fig. 4 | Evolution of morphometry and climatic niche in the genus *Microcebus*. a,b**, Regression (dashed line) of morphometric (**a**) and climatic niche hypervolume (**b**) overlap through time, that is, across nodes of the tree in Fig. 2b. The vertical line of node b estimate represents the 95% confidence interval. Light lines represent linear regressions of 1,000 datasets simulated under the most likely models (Ornstein–Uhlenbeck (OU) for morphology and Brownian motion (BM) for climatic niche. **c**, Distributions of Spearman's correlation coefficient ($r_s$) between node age and morphometric or climatic niche hypervolume overlap, from 1,000 simulations under OU, BM and EB models of character evolution. Dashed horizontal lines indicate the observed $r_s$.

of evolution is more likely to reproduce the observed data than an OU model (stabilizing selection; Fig. 4d). When the EB model is fitted to the data, the estimated rate of change (*r*) is equal to zero, which is equivalent to a BM model (Supplementary Fig. 21a,b). This suggests that neither the EB nor the OU model could explain the data better than a simple neutral model of evolution, which is also supported by high estimates of phylogenetic signal across bioclimatic variables (Supplementary Table 11) and the overall low climatic niche overlaps observed among *Microcebus* species (for similar net rates of trait evolution, a BM model would lead to lower overlap than an OU model with convergence towards a single optimum; Supplementary Fig. 21c)[89,90]. Taken together, this indicates that the climatic niches of *Microcebus* species evolved through a neutral process of niche diversification, pointing to stochastic events of colonization of available climatic niches across the island, without being necessarily driven by systematic adaptation to specific niches. This recalls previous work showing that primate assemblages in Madagascar, and elsewhere, may have formed according to the neutral theory of community assembly—species of a community are ecologically equivalent, and their relative abundance is mainly the result of stochastic processes of extinction, immigration and speciation[91]. This does not preclude, however, that Pleistocene climatic fluctuations may have promoted geographic isolation among mouse lemur populations, as noted in the previous section, fostering genetic, ecological and/or climatic niche divergence, eventually leading to speciation.

Principles of primate community assemblages and interspecies competition provide a plausible explanation for the co-occurrence of morphological stasis and neutral climatic niche evolution in mouse lemurs. Despite high variability in climates and ecosystems throughout Madagascar, lemur community assemblages are notably similar across regions, inter-river systems and forest ecosystems (low beta functional and phylogenetic diversity)[91]. That is, regardless of habitat, one (or exceptionally two) species of the genera *Lepilemur*, *Microcebus*, *Propithecus*, *Avahi*, *Eulemur* and *Cheirogaleus*, respectively, can be found in nearly all inter-river systems[45]. Remarkably, lemur communities contain more distantly related species compared both to random expectation and to other primate assemblages in the world (relative nearest phylogenetic distance for Africa: −0.25 ± 0.74, Neotropics: −0.47 ± 0.7, Asia: −0.71 ± 0.78, Madagascar: −1.48 ± 1.1)[91–93]. This observation is consistent

with the idea that interspecific competition may have resulted in exclusion of closely related species[94,95], thus favouring communities with high levels of phylogenetic separation[96,97]. Given the similarity of lemur assemblages across regions and habitats, their trait diversity and niche partitioning may predate the actual diversification of various lemur genera[67]. Accordingly, there may have been general 'rules' of lemur community assembly that constrained *Microcebus* species to occupy a specific niche throughout their evolutionary diversification (that is, small size, nocturnality, omni-frugivory, fine branch niche), as they radiated alongside distantly related and larger-bodied lemur species[33]. The associated stabilizing selection may have intensified competitive exclusion among closely related species, resulting in low levels of co-occurrence of congeneric species[38] and promoting phylogenetically overdispersed lemur communities.

### Conservation implications

The increased sampling effort and taxonomic deflation presented in our study have implications for the conservation status of previously recognized species. Specifically, we propose to synonymize two micro-endemic Critically Endangered (CR), three Endangered (EN), one Data Deficient (DD) and one not yet evaluated candidate species, resulting in a lower recommended level of endangerment for six previously assessed lineages (Fig. 2d, Supplementary Fig. 22 and Supplementary Table 12; Supplementary Results and Discussion: Change in conservation status provides details). Furthermore, our extensive data collection suggests revising the conservation status of five other species to a lower level despite substantial habitat loss in the recent past. Perceived advantages of taxonomic inflation are often linked to the idea that species should be the primary units for setting conservation priorities rather than populations (for example, primates[27], African ungulates[50] and others[98], but see Creighton et al.[99]). We argue, however, that the concept of species as used in conservation policy decision-making often neglects pronounced intraspecific partitioning of genetic, morphological and ecological diversity[100]. For example, our analyses combined with more detailed population genetic studies[36,38,59] allow the identification of at least 39 genetically differentiated, reciprocally monophyletic clades within the 19 *Microcebus* species (Supplementary Results and Discussion: Species delimitation and diagnosis). To prevent the permanent loss of irreplaceable genetic and other biological diversity, these

populations demand separate conservation attention, particularly in light of the high rates of habitat loss and fragmentation on the island today (Supplementary Table 12)[101]. Currently, at least 12 of these (31%) occur outside formally protected areas. Conversely, our findings also imply that *M. manitatra*, *M. ganzhorni* and southeastern *M. murinus* populations, for instance, constitute a single lineage (Extended Data Fig. 9), which could indicate that these three groups no longer need to be treated as independent entities for conservation purposes. Such mismatches between patterns of diversity and conservation efforts due to a focus on species-level classification are expected to be common in many taxa with strong population structure, particularly when species differentiation is cryptic (for example, primates[45,102], small mammals and reptiles[63,103]). We therefore conclude that it is crucial for conservation programmes to also prioritize the preservation of divergent intraspecific lineages as evolutionarily significant units[104,105]. Comprehensive phylogenetic, population genetic and taxonomic investigations will be essential for their identification.

## Methods

### Data collection
We compiled a comprehensive set of genomic, morphometric, bioclimatic and behavioural data across all *Microcebus* species from the literature and our own research (below; Supplementary Fig. 22 and Supplementary Tables 13–17). All field procedures were approved by Malagasy authorities and adhered to Malagasy regulations, standards of the International Primatological Society[106] and the 'proposal for ethical research conduct in Madagascar'[107]. Species assignments were based on geographic location, preliminary identification of the respective field primatologist and in part on previous sequencing activities in different laboratories.

**Genomics.** Our genomic dataset comprised *SbfI* RAD sequencing data for 300 *Microcebus* samples across all 25 described and one putative species (range: 2–35 samples per species; Fig. 2a, Supplementary Fig. 22a and Supplementary Table 13). Three *Cheirogaleus* and *Mirza zaza* individuals, respectively, were added as outgroups. Data were already published for 81 samples[36,37,59,108] or newly generated from tissues collected between 1995 and 2018 (225 samples). Animals were captured with Sherman Life traps or directly by hand during the night. Ear biopsies (~2 mm²) were taken and stored in Queen's lysis buffer[109] until DNA extraction. Animals were released at their capture location within 24 h. DNA was extracted using a modified QIAGEN DNeasy Blood and Tissue Kit protocol[110] or a standardized phenol/chloroform extraction technique[111]. RAD library preparation and sequencing followed the three protocols described in Poelstra et al.[37] (Supplementary Methods "Library preparation").

**Morphometry.** We obtained data for 13 morphometric variables (ear length, ear width, head length, head width, snout length, intraorbital distance, interorbital distance, body length, tail length, lower leg length, hind foot length, third toe length, body mass) across 1,673 adult *Microcebus* specimens (range: 2–351 specimens per species; Supplementary Fig. 22b and Supplementary Table 14) from Schüßler et al.[76], accounting for measurement and observer bias.

**Distribution and bioclimate.** We assembled occurrence data for *Microcebus* species, resulting in 373 spatially filtered records that could be assigned to a particular species (range: 1–41 records per species; Supplementary Fig. 22c and Supplementary Table 15). We extracted eight bioclimatic variables that are considered ecologically meaningful for lemurs (that is, isothermality, temperature seasonality, maximum temperature of warmest month, minimum temperature of coldest month, annual precipitation, precipitation seasonality, precipitation of wettest and driest quarter)[88,89] with a resolution of 30 arcseconds (1 km) for each record from the CHELSEA v2.1 database[112].

**Reproductive activity.** We assembled 2,354 assessments of reproductive state (that is, presence/absence of oestrus, pregnancy or lactation in females and of enlarged testes in males at the time of capture; 1,006 male and 1,348 female records) across 24 described *Microcebus* species from our own research and the literature (range: 4–376 assessments per species; Supplementary Fig. 22d and Supplementary Table 16).

**Acoustic communication.** We obtained data on 623 alert and advertisement calls across five described *Microcebus* species from the sound archive of the Institute of Zoology of the University of Veterinary Medicine Hannover, Foundation (range: 91–157 calls per species; Supplementary Fig. 22e and Supplementary Table 17).

### RAD genotyping
Raw RAD reads were demultiplexed with the process_radtags function of Stacks v2.0b[113], trimmed with Trimmomatic v0.39[114] (Leading: 3, Trailing: 3, Slidingwindow: 4:15, Minlen: 60) and aligned against the *M. murinus* reference genome (Mmur 3.0)[115] with BWA-MEM v0.7.17[116]. Reads not mapping to autosomal scaffolds or with a mapping quality below 20 were removed using SAMtools v1.11[117]. Paired-end reads were also filtered for proper pairing and deduplicated. RAD sequencing statistics are given in Supplementary Table 18.

We created distinct datasets specifically tailored to each analysis, using called genotypes and genotype likelihoods to ensure robustness of our results (datasets and associated analyses are described in Supplementary Table 19). First, genotypes of 214 individuals with mean forward read depth across RAD sites larger than five were called using GATK v4.1.9.0[118]. After removing indels, only sites with a global sequencing depth between ten and the sum of the 0.995 quantiles of per-individual depth distributions and represented in at least three individuals were retained. In addition, for each individual, sites with a sequencing depth lower than two or larger than the maximum 0.995 quantile among per-individual depth distributions were masked. Subsequently, a minor allele count filter of two was applied. Sites satisfying one of the following conditions were removed using VCFtools v0.1.17[119]: FS > 60.0; MQ < 40.0; MQRankSum < −12.5; ReadPosRankSum < −8.0; ABHet < 0.2 or ABHet > 0.8. Finally, we created five genotype sets with varying amounts of maximum missing data per site (5%, 25%, 50%, 75%, 95%). Second, we followed Poelstra et al.[37] to convert called genotypes to phased RAD loci for a subset of two samples with decent read depth and geographic representativeness per *Microcebus* lineage and two *Mirza zaza* samples, which served as the outgroup. Extracted orthologues were re-aligned with MUSCLE v3.8.31[120]. Third, we estimated genotype likelihoods (GL) with the SAMtools model in ANGSD v0.92[121,122] for nine sample sets consisting of the species pairs and triplets for which species delimitation tests were conducted (Supplementary Table 13). We retained only (1) sites with a total sequencing depth larger than twice the number of focal individuals and smaller than the sum of the 0.995 quantiles of per-individual depth distributions, (2) sites with an individual depth larger than two and smaller than the maximum 0.995 quantile among per-individual depth distribution, (3) sites present in at least 75% of focal individuals, (4) bases with a mapping quality larger than 20, (5) uniquely mapping and properly paired reads with a minimum mapping quality of 20, (6) biallelic variants with a probability below 1e−5 and (7) sites with a minor allele frequency (MAF) larger than 0.05.

### Phylogenetic inference
We used two complementary approaches for phylogenetic inference from single nucleotide polymorphisms (SNPs) and explored the effects of missing data by using genotype call sets with varying amounts of maximum missing data per site (above). First, we performed maximum likelihood (ML) inference with IQ-TREE v2.2.0[123], using the GTR + Γ model of sequence evolution and correcting for ascertainment bias. We used 1,000 replicates to perform a SH-like approximate likelihood ratio

test (SH-aLRT)[124] and estimate ultrafast bootstrap support optimized by nearest neighbour interchange. Second, we used the coalescent-based algorithm SVDquartets[125] implemented in PAUP* v4.0a (build 168)[126] on a SNP set thinned every 10,000 bp to ensure site independence. We evaluated 20,000,000 quartets, estimated support over 100 standard bootstraps and assigned either individuals or described species as tips. As an exception, we subdivided *M. lehilahytsara*, *M. murinus* and *M. simmonsi* into populations (Supplementary Fig. 6) because these species were not recovered as monophyletic by ML inference (Supplementary Figs. 1–5). Associated alignment statistics are given in Supplementary Table 20.

### Species delimitation

Similar to Padial et al.[34], we present a practical integrative framework to systematically delimit species across the cryptic genus *Microcebus* (Fig. 1). We considered the 25 currently described and one putative *Microcebus* species as species hypotheses (that is, candidates). As an exception to this, *M. murinus* was split into two candidates (north vs central) due to its wide distribution and phylogenetic structure (Fig. 2a,b). We then applied our framework separately to nine groups of allopatric sister candidate species (Fig. 2b), within which we performed pairwise delimitation tests to identify pairs not representing separately evolving metapopulations (that is, distinct species). Accordingly, the framework is specifically designed to delimit allopatric sister species, for which species status cannot be verified through cases of syntopic occurrence without interbreeding. It characterizes genomic differentiation while integrating available additional lines of evidence (that is, morphometry, climatic niche, reproductive activity, acoustic communication) as proxies for reproductive isolation and/or measures of trait divergence to synonymize or confirm candidates as distinct species (below).

Our approach places a particular focus on identifying intraspecific geographic structure (Fig. 1). To do so, we derived genus-specific thresholds for spatial analyses from variation observed among populations of *M. tavaratra* (detailed for each analysis below), an extensively sampled species with well-characterized population structure. It comprises both fragmented and continuous populations that display IBD and are unlikely to represent diverging lineages or potential candidate species (Supplementary Fig. 23)[35,38,39]. Accordingly, this species can provide estimates of intraspecific variation expected in a spatially structured yet interconnected *Microcebus* species. To obtain an additional reference, we also compared differentiation of candidate species to that of the widely distributed *M. lehilahytsara* and *M. mittermeieri*, which were recently proposed to be synonymized based on evidence of gene flow and a cline in genomic and morphometric diversity (Extended Data Fig. 3)[30,36,37]. More details on why we consider these species appropriate references are given in the Supplementary Methods: Species delimitation. Notably, the estimated thresholds do not necessarily apply to other study systems because they likely depend on features shared within a genus, such as life-history traits (for example, dispersal), population size and genome architecture.

**Genomics.** Because species often show spatial patterning of variation[127], which can confound species delimitation if ignored or not represented adequately in the sampling[54,55], we first tested whether genetic distances between candidate individuals could be explained by a model of intraspecific geographic structure. To do so, we developed a heuristic approach based on IBD, consisting of the following four steps (Extended Data Fig. 1):

1. We divided the genotype set with a maximum of 5% missing data per site into windows containing a fixed number of SNPs with the function *vcf_windower* of the R package 'lostruct' v0.0.0.9000[128]. This resulted in a set of 104,000 SNPs across 104 windows (Supplementary Table 19). We used SNP number and not window length in bp to divide genomic data because of the scattered nature of RADseq data. We selected the appropriate number of SNPs per window (that is, 1,000 SNPs; Supplementary Fig. 24) by minimizing the difference between signal and noise (calculated as in Li and Ralph[128]). The impact of window size selection on delimitation results is illustrated in Supplementary Fig. 25.

2. For each candidate pair and genomic window, we computed IBD within and between candidates by correlating individual genetic distances with geographic distances (on log scale). Genetic distances were calculated using a custom R script based on the pixy algorithm[129] to obtain an unbiased estimate of the average number of nucleotide differences per site between two individuals ($\pi$). However, unlike pixy, we took only variant sites into account.

3. We used the normalized root mean square error (NRMSE) to quantify deviations of observed genetic distances between candidates from those predicted by the within-candidate geographic clines in genetic distance (Supplementary Methods: Species delimitation provides details). Accordingly, two NRMSE values were obtained for each genomic window, one for each candidate. The rationale behind the NRMSE is that we can control for within-candidate genetic variation, so that NRMSE distributions are comparable across all candidate pairs, regardless of the associated extent of spatial structure.

4. To test whether the obtained NRMSE distributions across genomic windows were consistent with a null hypothesis of intraspecific geographic structure, we compared them to empirical NRMSE distributions (treated as null distributions) inferred from *M. tavaratra* and *M. lehilahytsara* (including the former *M. mittermeieri*). For *M. tavaratra*, individual genetic distances between fragmented and within continuous populations were considered as between- and within-candidate comparisons, respectively (Supplementary Methods: Species delimitation provides details). For *M. lehilahytsara*, we used as between-candidate pairwise comparisons those between *M. mittermeieri* and *M. lehilahytsara* individuals (Extended Data Fig. 3d). The proposed heuristic test rejected the intraspecific clinal variation model, if the 0.05 quantiles of both NRMSE distributions of a candidate pair were above the 0.95 quantiles of the reference null NRMSE distributions, indicating that genetic distances between candidates could not be explained by a geographic cline (evidence for retaining candidate species). Conversely, if the 0.95 quantiles of a single NRMSE distribution of a candidate pair was below the 0.95 quantiles of the null NRMSE distributions, we considered genetic distances to be congruent with a model of intraspecific structure (evidence for synonymization). Cases that were neither rejecting nor congruent with the intraspecific model were considered inconclusive.

In cases where the IBD-based approach was inconclusive for species delimitation, we considered (1) the absence of reciprocal monophyly in the inferred *Microcebus* phylogeny, (2) the presence of individuals with admixed ancestry and (3) a genealogical divergence index (gdi)[130] smaller than 0.2 as proxies for a lack of genomic independence and therefore as sufficient evidence to synonymize candidates. Individual ancestries were estimated from genotype likelihoods, using NGSadmix v32[131] and setting the number of a priori clusters ($K$) from two to five. Ten independent runs were conducted. The gdi was calculated as $\mathrm{gdi} = 1 - e^{\frac{-2\tau}{\theta}}$, where $\tau$ and $\theta$ represent the posterior parameter means of the MSC models built for divergence time estimation (below). The gdi helps to differentiate population structure from speciation by quantifying the degree of genetic divergence of candidates due to genetic isolation and gene flow. Because the MSC is prone

to oversplitting[52,53], we used the gdi only to synonymize but not to confirm candidate species. For the same reason, we did not apply other coalescent-based species delimitation algorithms (for example, Yang[132]). Whereas we followed Jackson et al.[130] in adopting a gdi threshold of 0.2 for species synonymy, this value is likely too low for mammals (Supplementary Methods: Species delimitation provides details), potentially hindering accurate distinction between intraspecific lineages with limited divergence and those undergoing speciation. Because the three criteria were only employed if the IBD-based test failed to be conclusive, our framework was able to confirm candidates exhibiting introgression and hybridization if genetic distances between candidates were significantly higher than those within candidates while accounting for geography (that is, the IBD-based test clearly rejected an intraspecific clinal variation model).

Because all sister candidate species considered here occur allopatrically and genomic data can be extremely powerful at displaying differentiation even among distinct localities of the same population[49], we confirmed candidates with inconclusive analyses of IBD and no other evidence for synonymization (such as a lack of reciprocal monophyly, admixed ancestry or low gdi) as valid species only if they exhibited substantial differentiation in at least one additional taxonomic character for which data were available (that is, morphometry, climatic niche, reproductive activity, acoustic communication) and if other explanations for such differentiation (for example, plasticity or local adaptation) were unlikely[22,34]. In other words, these proxies for reproductive isolation and/or measures of trait divergence were only employed to confirm two candidates as distinct species subsequent to the detection of genomic differentiation, but an overlap in such characters was not used as direct evidence for synonymization.

**Morphometry.** Even though the genus *Microcebus* is considered cryptic, quantitative analyses can reveal consistent morphometric differences between lineages[76]. We considered such differences if accompanied by genomic differentiation (and accounting for geographic variation) as evidence to confirm candidate species.

Similar to the analysis of IBD based on genomic data, we leveraged the NRMSE to test if variation in morphometry can be explained by IBD. Instead of resampling diversity along genome segments, we resampled morphometric variables. Species candidates with less than five individual morphometric records were not considered. For each comparison, morphometric variables missing in at least one candidate were discarded as were individuals with more than 70% missing data. Because compared candidates did not always share the same number of variables, we created 200 resampled replicates (without replacement) for each candidate comparison while maximizing the number of variables, using the *combn* function in R. For comparisons for which the number of shared variables enabled less than 50 resampled replicates, we resampled individuals instead of morphometric variables.

The results were interpreted using the *p*-value distribution of the correlations between pairwise geographic and morphometric distances (1-hypervolume overlap) among all considered samples as well as the NRMSE distributions (calculated as in the genomic procedure), which were obtained from the resampled data. *M. tavaratra* was used as reference. Here the pair *M. lehilahytsara*/*M. mittermeieri* was not considered as reference because it did not exhibit a significant pattern of morphometric IBD. Candidate pairs with a *p*-value 0.95 quantile above 0.05 were not considered at IBD (that is, not fitting an intraspecific model). Candidate pairs with at least one NRMSE 0.50 quantile (that is, the median) below the reference's 0.95 quantile were considered fitting an IBD pattern of intraspecific character variation. Here we used the median (instead of the 0.95 quantile as in the genomic procedure) to account for the lower number of variables and the high inter-observer effect of morphometric data. For candidate pairs not matching a model of intraspecific character variation, morphometric hypervolume overlap (below) was subsequently considered for species delimitation.

In other words, only if the IBD test showed that the species candidates did not form a continuous morphometric cline across space, we used morphometric differences to inform the taxonomic procedure.

We quantified pairwise overlap in morphometry between candidates using the maximal value of asymmetric overlap in *n*-dimensional hypervolumes (where *n* relates to the number of morphological variables) with the function *dynRB_VPa* in the R package 'dynRB' v0.18[133,134], setting 'product' as aggregation method and using 51 dynamic range boxes. Confidence intervals were estimated by jackknife, resampling 90% of the individuals 100 times. Confidence intervals of morphometric overlap between *M. lehilahytsara* and *M. mittermeieri* and between fragmented *M. tavaratra* populations were taken as reference for species delimitation. To warrant comparability of overlap values across the dataset, we chose four morphometric variables with high ecological relevance that were present in most candidate species for these analyses (that is, ear length, head length, body mass, tail length)[76]. Finally, because *M. tavaratra*, *M. ravelobensis* and *M. murinus* (north) had much larger sample sizes than the other candidate species, we randomly subsampled 150 individuals for each of these 100 times and used average values across replicates.

**Climatic niche.** Most described *Microcebus* species are confined to relatively small geographic areas (that is, they are micro-endemics, but see *M. murinus* and *M. lehilahytsara*[45], which correspond to specific bioclimatic conditions. Whereas most allopatric sister lineages occupy neighbouring regions and are therefore expected to share most of their climatic niche, sister lineages using drastically different bioclimatic niches may show different adaptations. We therefore considered pronounced differences in climatic niche space if accompanied by genomic differentiation as potential evidence to confirm candidate species.

We estimated climatic niche models for each candidate species based on extracted bioclimatic variables (above) using the MaxEnt algorithm as implemented in the R package 'ENMtools' v1.0.7[135]. To do so, we transformed the bioclimatic data via principal component analysis and used only the first three principal components (PCs; explaining 93.1% of the variation) to reduce multicollinearity and to accommodate low sample sizes for some candidate species. Parameters (that is, feature classes and regularization multipliers) were independently tuned based on lowest Akaike Information Criterion (AIC) value, using 10,000 background points. Model validation was performed based on the area under the receiver operating curve (AUC) and the continuous Boyce index (CBI), using a leave-one-out cross-validation approach in the R package 'ENMeval' v2.0[136,137].

Niche overlap among sister candidate species was subsequently quantified with Schoener's $D$[138], which ranges from 0 (no overlap) to 1 (complete overlap). Confidence intervals were estimated by jackknife, resampling 90% of the individuals 100 times. Using identity tests as implemented in 'ENMtools', we tested for significant deviations of the empirical estimate of niche overlap from a null distribution. Confidence intervals of niche overlap between *M. lehilahytsara* and *M. mittermeieri* and among *M. tavaratra* populations were taken as reference for species delimitation.

**Reproductive activity.** Whereas differentiation in reproductive activity can directly preclude interbreeding and lead to speciation, it can also emerge as a consequence of reproductive isolation and divergence, making it a valuable proxy for species delimitation. We therefore considered consistent differences in reproductive activity as strong evidence to confirm candidate species if accompanied by genomic differentiation.

For each candidate species and month of the year, we estimated the proportion of reproductively active individuals and total individuals surveyed, using the presence of oestrus, pregnancy and lactation in females and the presence of enlarged testes in males as reproductive indicators (during the non-breeding season testes are regressed[139]).

Records of pregnancy and lactation were adjusted to obtain the approximate timing of oestrus, considering that these can be diagnosed about 2 and 2–3.5 months after oestrus, respectively[140,141]. Details are given in the Supplementary Methods: Species delimitation. Subsequently, we assessed qualitatively whether there was evidence for asynchronous reproductive schedules, as the quantification of pairwise overlap values was impeded by the large variation in sampling effort and period across candidate species.

**Acoustic communication.** Similar to reproductive activity, acoustic communication is directly associated with reproduction and therefore a valuable proxy for species delimitation. We therefore quantified pairwise overlap in alert and advertisement calls of candidate species using the maximal value of asymmetric overlap in $n$-dimensional hypervolumes (Supplementary Methods: Species delimitation for details).

### Divergence time estimation

To determine the temporal context of diversification in the genus *Microcebus*, we estimated divergence times among species under a MSC model in BPP v4.4.1[132]. We aimed to avoid biases of concatenation and phylogenetically distant, external fossil calibrations[68,69] (no fossil calibrations are available in Lemuriformes; Supplementary Methods: Divergence time estimation provides details) by accounting for incomplete lineage sorting and transforming branch lengths from substitutions per site to substitutions per absolute time units based on external evidence from per-generation de novo primate mutation rates and *Microcebus* generation times. Four independent chains of BPP (analysis A00) were run for 1,000,000 generations with a burn-in of 20% on the tree topology estimated with IQ-TREE and using the 6,000 extracted RAD locus alignments with the least amount of missing data (Supplementary Table 20 provides statistics) to decrease computational burden. We set a gamma prior for $\theta$ ($\alpha = 2$; $\beta = 2,000$) and an inverse gamma prior for $\tau$ ($\alpha = 3$; $\beta = 0.0041$). Convergence of chains and effective sample size were checked with Tracer v1.7.2[142]. Final estimates were obtained by averaging across the four chains, which were largely congruent (Supplementary Figs. 9–12). Following Poelstra et al.[37], we used a mutation rate of $1.236 \times 10^{-8}$ per site per generation and a generation time of 3.5 years to convert $\tau$ to years (Supplementary Methods "Divergence time estimation for details"). To explore how uncertainty in these estimates affected inferred divergence times, we also did the conversion using a gamma distribution with a mean of $1.236 \times 10^{-8}$ and a variance of $0.107 \times 10^{-8}$ and a lognormal distribution with a mean of $\ln(3.5)$ and a standard deviation of $\ln(1.16)$ for mutation rate and generation time, respectively.

### Biogeographic reconstruction

We reconstructed ancestral habitats along the *Microcebus* phylogeny (that is, the spatial context of diversification) using trait-dependent dispersal models in the R package 'BioGeoBears' v1.1.2[143]. For this, recent distributions of species retained in our taxonomic revision were related to biogeographic regions following three different classifications: (1) dry vs humid forest, (2) five major ecoregions[58] and (3) the Köppen–Geiger climate classification[144]. For each classification, we fitted a Dispersal–Extinction–Cladogenesis model[145] and models analogous to the Bayesian Inference of Historical Biogeography for Discrete Areas[146] and the Dispersal-Vicariance[147] models with (+J) and without jump dispersal. Model fit was evaluated with the AIC corrected for sample size (AICc).

### Modelling morphological and climatic niche evolution

We aimed to identify the evolutionary processes that best explain the diversification of morphometric traits and climatic niche along the *Microcebus* phylogeny. To do so, we considered three evolutionary models that have often been compared for understanding evolutionary divergence of traits in extant and fossil lineages[83,84]: (1) a neutral model of genetic drift where trait differences among lineages accumulate over time (random walk), modelled as a multivariate Brownian Motion (BM) process; (2) rapid evolution followed by stasis, where the rate of trait diversification among lineages decreases exponentially over time, equivalent to a BM process with a time-dependent rate of change and modelled as a multivariate Early-Burst (EB) process and (3) stabilizing selection (random walk with a single stationary peak), where a trait can randomly change over time although it will tend to return to an optimum trait value (that is, the stationary peak), modelled by a single-rate multivariate Ornstein–Uhlenbeck process (OU)[148,149]. The root state and the optimum of the OU model are distributed according to the stationary distribution of the process (that is, they have the same value), because they are not identifiable on ultrametric trees[150,151].

The morphometric dataset considered for this analysis consisted of seven variables (out of 13) across 15 recognized *Microcebus* species (out of 19), chosen to minimize the amount of missing data across individuals (Supplementary Table 9) and exhibiting good measurement reproducibility across researchers[76]. The bioclimatic dataset comprised the eight bioclimatic variables used for niche modelling (Supplementary Table 10). Phylogenetic signal was estimated for each bioclimatic variable through Blomberg's $K$ and Pagel's $\lambda$, using the function *phylosig* of the R package 'phytools' v2.3-0[152]. For each species, we computed the mean and the squared standard error of every variable and the covariance matrix between variables.

Because we were interested in identifying the evolutionary process that is most likely to reproduce the observed changes in morphometric and climatic niche overlap along the *Microcebus* phylogeny, we considered as observed data (or test statistic) the non-parametric Spearman's correlation coefficient ($r_s$) between node age and overlap, a summary statistic describing these temporal changes. Pairwise overlaps between species were quantified as the maximum of asymmetric overlap of the respective $n$-dimensional hypervolumes, using the R package 'dynRB'. For niche data, overlap was additionally quantified as Schoener's $D$ of climatic niches (Species delimitation). The correlation of node ages and overlap values was computed using the *age.range.correlation* function of the R package 'phyloclim' v0.9.5[153]. This metric computes nested averages of pairwise overlaps between species in each clade to account for their phylogenetic relatedness, providing an estimate of the average overlap for each node in the tree without having to reconstruct ancestral morphological traits[138,154].

For identifying the evolutionary process that best explained the data (that is, morphometry or climatic niche), we used the following steps: (1) fitting evolutionary models to the data; (2) simulating data under the inferred model parameters and (3) comparing the observed correlation of node ages and overlap values with the distribution of this statistic in each simulated evolutionary model:

1. We fitted evolutionary models to both datasets using maximum likelihood (accounting for measurement error and using the L-BFGS-B and subplex algorithms) as implemented in the R package 'mvMORPH' v1.1.9[151]. We used the *mvBM* function (model = 'BM1'; trend = FALSE) to model random walk, the *mvEB* function (setting the upper bound for the $r$ parameter to zero) to model a burst of morphological diversification, which decreases exponentially over time, and the *mvOU* function (model = 'OU1' and root = FALSE) to model stabilizing selection on trait variance around a single optimum. All model functions account for trait correlation by modelling the covariance matrix. We ensured reliable parameter estimation by checking the eigendecomposition of the Hessian matrix. The relative fit of each of the three models was assessed using the AICc.
2. We simulated data along the *Microcebus* phylogeny with the *mvSIM* function of the R package 'mvMORPH'. For each of the three models (BM, EB and OU), we simulated 1,000 independent datasets, using estimates of the previous step

(obtained with the *mvBM*, *mvEB* and *mvOU* functions) as model parameters and using the squared standard error matrix computed from the observed data as measurement error. For each simulation, we checked that the data would include only positive trait values. Because *mvSIM* simulates the trait means of species in a tree, but pairwise overlap was measured from the *n*-dimensional hypervolume of trait values of sampled individuals, we used the *rtmvnorm* function of the R package 'tmvtnorm' v1.6[155] to simulate trait values of individuals from trait means obtained in *mvSIM* simulations. For each species, the *rtmvnorm* function randomly samples trait values of individuals from a truncated multivariate normal distribution with mean equal to the simulated species trait mean and covariance structure given by the covariance matrix estimated from the observed data. Across species, the sample size of the simulated traits of individuals was equal to that in the real dataset. We chose to use a truncated multivariate normal distribution for three main reasons. First, most morphometric and climatic traits are normally distributed as shown by the Shapiro–Wilk test (Supplementary Figs. 28 and 29 and Supplementary Tables 21 and 22). Second, the truncation avoids simulating negative trait values as we set the lower limit to zero. Finally, the covariance matrix enables consideration of trait covariation, which is key for reproducing multivariate trait evolution.

3. For each simulated dataset, we computed overlaps between species pairs as described above and quantified their correlation with node age through non-parametric Spearman correlation. Ultimately, we compared the observed correlation coefficients ($r_s$) to the distribution of this statistic obtained from the 1,000 simulations of the tested evolutionary models. If the observed statistic was above the 0.95 quantile or below the 0.05 quantile of the simulated distribution, we rejected the model underlying the simulated data.

To assess the rejection power of the test statistic $r_s$, we carried out a cross-validation analysis on morphometric data (Supplementary Fig. 18a,c). We randomly subsampled 100 out of 1,000 datasets simulated under both the BM and OU models. We excluded the EB model because from the fitted parameter values (for example, pattern of rate change $r = 0$), it was not distinguishable from a classical BM model (Supplementary Fig. 19). Each randomly sampled dataset was then fitted to the two alternative models (step 1) and the estimated parameter values were used to simulate 500 independent datasets (step 2). The observed test statistic for each of the 100 simulations was then compared to the BM- and OU-based distributions of this statistic obtained from the additional 500 simulations (step 3). The results of the cross validation were classified into four categories: (1) reject the BM model, (2) reject the OU model, (3) reject both BM and OU models or (4) reject neither the BM nor the OU model. We considered a specific model rejected when the observed statistic was above the 0.95 quantile or below the 0.05 quantile of the simulated distribution. For comparison, we also assessed the probability of identifying the correct model when using the AIC. We did not carry out a cross-validation analysis on climatic niche overlap data because neither the EB model, which converged to a BM model, nor the OU model, for which we could not find reliable solutions during model fitting, were sufficiently supported based on our data.

### Conservation reassessment

On the basis of the extensive sampling and updated taxonomy presented here, we provide new conservation status recommendations for all valid *Microcebus* species following International Union for Conservation of Nature (IUCN) guidelines[156]. To do so, we first produced binary distribution maps in ArcGIS Pro v3.1.0 based on climatic niche models by applying the 10-percentile training presence as a threshold above which areas were deemed suitable for presences[157]. Next, we excluded areas separated by known geographic barriers across which species could not be detected (for example, rivers[59]), resulting in a more accurate estimate of the Extent of Occurrence as defined by the IUCN. The Extent of Occurrence was further refined by considering only forest cover in 2017[158], representing the actual inhabitable area for *Microcebus* species or the Area of Occupancy. Finally, we estimated Area of Occupancy loss over the past three generations (that is, 11.5 years, assuming a generation time of 3.5 years[58,159,160]) by comparing forest cover in 2017 to that in 2005[158].

### Reporting summary

Further information on research design is available in the Nature Portfolio Reporting Summary linked to this article.

## Data availability

All new sequencing data have been made available through NCBI Bio-Projects PRJNA560399 and PRJNA807164. Individual BioSample accessions are given in Supplementary Table 13. Analysis input, output and configuration files are available via Dryad at https://doi.org/10.5061/dryad.b2rbnzsp3 (ref. 161).

## Code availability

Analysis scripts can be found via Github at https://github.com/t-vane/van_Elst_et_al_2024_Cryptic_diversification.

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

## Acknowledgements

This study was supported by the German Research Foundation DFG (Ra 502/23-1, Ra 502/7-1, Ra 502/7-3, Ra 502/20-1 and Ra 502/20-3 to U.R.; Ka 1082/8-1, Ka 1082/8-2 and Ka 1082/19-1 to P.M.K.; Ga 342/19 to J.U.G.), the Agence Nationale de la Recherche ANR (LABEX CEBA: ANR-10-LABX-25-01, LABEX TULIP: ANR-10-LABX-0041 and a visiting scientist grant to A.D.Y., L.C. and J.S.), the National Science Foundation NSF (NSFDEB-NERC: 2148914 to A.D.Y.), the Portuguese Foundation for Science and Technology FCT (PTDC/BIA-BEC/100176/2008, PTDC/BIA-BIC/4476/2012 and BIODIVERSA/0003/2015 to L.C.; PD/BD/114343/2016 to G.M.S.; SFRH/BD/64875/2009 to J.S.), the German National High Performance Computing Alliance NHR@ Göttingen (nib00015 to U.R. and T.v.E.), the German Academic Exchange Service DAAD (91529232 to A.F.H.; 91565325 to M.R.E.; fellowships to B.R. and S.R.), the Bauer Foundation of the German Foundation Center DSZ (T237/22985/2012/kg to D.S.), the Rufford Foundation (RSG-10941-1 to J.S.; RSG-12973-1 to M.T.I.; RSG-15472-1 to A.F.H.), the French National Centre for Scientific Research CNRS (BEEG-B IRP to L.C.), the German Federal Ministry of Education and Research BMBF (01LC1617A to U.R.), the ERA-NET biodivERsA (2015-138 to U.R.), the Lemur Conservation Action Fund (to M.B.B.), Operation Wallacea (to U.R.), Landesforschungsförderung Hamburg (to J.U.G.), the German Federal Agency for Nature Conservation BfN (to U.R.), Gesellschaft der Freunde der TiHo e.V. (to G.O.), the Institute of Zoology of the University of Veterinary Medicine Hannover (to U.R.), and the National Institutes of Health NIH (R35 GM136290 to Graham Coop, which in part supported G.M.S.). We thank the Direction Générale du Ministère de l'Environnement et des Forêts de Madagascar and Madagascar's Ad Hoc Committee for Fauna and Flora and Organisational Committee for Environmental Research (CAFF/CORE) for allowing us to conduct fieldwork. This study benefited from the continuous support of the Department of Animal Biology and Ecology, University of Mahajanga, the Department of Animal Biology, University of Antananarivo, Madagascar National Parks (MNP), the NGO Fanamby, the Groupe d'étude et de recherche sur les primates de Madagascar (GERP) and the MacArthur Foundation. We are grateful to J.B. Andriambeloson, R. Andriantompohavana, B. Andriatsitohaina, S. Atsalis, M. Barnavon, E.B. Beanaka, A. Beck, A. Bensouleimany, M. Dammhahn, R. Ernest, V. Gabillaud, N.K. Guthrie, T. Hafen, Irène, S. Kessler, F. Kiene, A. Klein, C. Kun-Rodrigues, A. Miller, D.S.A. Ousseni, N. Rabemananjara, D. Rabesamihanta, Z. Radavison, T. Radriarimanga, S.M. Rafamantanantsoa, F.D. Rakotoarizaka, A. Rakotomanantena, H. Rakotondramanana, T.N. Ralantoharijaona, M.L. Ramilison, M. Ramsay, R. Randriamampionona, F.J. Randriamaroson, E. Rasoazanabary, D.W. Rasolofoson, J.H. Ratsimbazafy, S.H. Roberts, V.S.T. Rovanirina, Tropical and Social Enterprise, V. Zietemann and the late E. Zimmermann for providing samples. We acknowledge E. Foss for performing preliminary bioinformatic analyses and P.D. Etter and E.A. Johnson for conducting RAD sequencing. We also warmly thank numerous Malagasy M.Sc. students, field assistants, volunteers, local guides and cooks for their help in the field and for sharing their incomparable expertise of the forest. *Misaotra betsaka anareo jiaby*.

## Author contributions

J.S., T.v.E., G.M.S., D.S., J.W.P., G.P.T., U.R., A.D.Y. and L.C. conceptualized the study. J.S., U.R., L.C., A.D.Y., E.E.L., D.S., R.M.R., M.B.B., G.M.S., S.M.G., G.O., R.R., B.R., S.R., J.R.Z., B.L.P., E.R., F.J., P.M.K., J.U.G., J.F.R., S.J.R., M.R.E., A.F.H., J.M.R., H.T., D.H., T.v.E., M.T.I. and A.N.R. performed and/or supervised fieldwork (that is, collection of tissue samples and morphometric, bioclimatic and reproductive data). D.S. compiled additional morphometric, bioclimatic and reproductive records from the literature. J.S., P.A.H., S.M., I.G.A.-P., J.F.R. and A.I. extracted DNA and/or constructed RADseq libraries. M.S. and A.F.H. collected and/or compiled data on acoustic communication. T.v.E., G.M.S., D.S., J.S., J.W.P., G.P.T., M.S. and D.H. analysed the data. T.v.E., J.S., G.M.S., D.S., A.D.Y. and M.S. wrote the paper. U.R., L.C., A.D.Y., J.S., J.U.G., E.E.L., M.B.B., P.M.K., S.M.G., B.R., S.R., D.S., T.v.E., M.R.E., A.F.H., M.T.I. and G.O. secured funding. All authors read, revised and approved the final paper.

## Funding

## Competing interests

The authors declare no competing interests.

## Additional information

**Extended data** is available for this paper at https://doi.org/10.1038/s41559-024-02547-w.

**Correspondence and requests for materials** should be addressed to Tobias van Elst, Gabriele M. Sgarlata, Dominik Schüßler or Jordi Salmona.

Tobias van Elst [1,28] ✉, Gabriele M. Sgarlata [2,3,28] ✉, Dominik Schüßler [4,28] ✉, George P. Tiley [5,6], Jelmer W. Poelstra [6,7], Marina Scheumann [1], Marina B. Blanco [6], Isa G. Aleixo-Pais [8], Mamy Rina Evasoa[1,9], Jörg U. Ganzhorn[10], Steven M. Goodman [11,12], Alida F. Hasiniaina [1,13], Daniel Hending [14], Paul A. Hohenlohe [15], Mohamed T. Ibouroi [2,16], Amaia Iribar[17], Fabien Jan[2], Peter M. Kappeler [18,19], Barbara Le Pors[2], Sophie Manzi[17], Gillian Olivieri[1,20], Ando N. Rakotonanahary[9], S. Jacques Rakotondranary [21], Romule Rakotondravony[9,22], José M. Ralison[23], J. Freddy Ranaivoarisoa[21], Blanchard Randrianambinina[9,22], Rodin M. Rasoloarison[18], Solofonirina Rasoloharijaona[9], Emmanuel Rasolondraibe[2], Helena Teixeira [1,24], John R. Zaonarivelo[25], Edward E. Louis Jr.[26], Anne D. Yoder[6], Lounès Chikhi[2,17,27], Ute Radespiel [1] & Jordi Salmona [17] ✉

[1]Institute of Zoology, University of Veterinary Medicine Hannover, Hannover, Germany. [2]Instituto Gulbenkian de Ciência, Oeiras, Portugal. [3]Department of Evolution and Ecology, University of California, Davis, CA, USA. [4]Institute of Biology and Chemistry, University of Hildesheim, Hildesheim, Germany. [5]Royal Botanic Gardens, Kew, Richmond, UK. [6]Department of Biology, Duke University, Durham, NC, USA. [7]Molecular and Cellular Imaging Center, The Ohio State University, Columbus, OH, USA. [8]Centro de Investigação de Montanha (CIMO), Instituto Politécnico de Bragança, Campus de Santa Apolónia, Bragança, Portugal. [9]Faculté des Sciences, de Technologies et de l'Environnement, Université de Mahajanga, Mahajanga, Madagascar. [10]Department of Biology, Universität Hamburg, Hamburg, Germany. [11]Field Museum of Natural History, Chicago, IL, USA. [12]Association Vahatra, Antananarivo, Madagascar. [13]School for International Training, Antananarivo, Madagascar. [14]John Krebs Field Station, Department of Biology, University of Oxford, Wytham, UK. [15]Department of Biological Sciences, University of Idaho, Moscow, ID, USA. [16]Université de La Réunion, Saint-Denis de La Réunion, France. [17]Centre de Recherche sur la Biodiversité et l'Environnement (CRBE), UMR5300 Université Toulouse, CNRS, IRD, Toulouse INP, Université Toulouse 3 Paul Sabatier (UT3), Toulouse, France. [18]Department Sociobiology/Anthropology, Johann-Friedrich-Blumenbach Institute of Zoology and Anthropology, University Göttingen, Göttingen, Germany. [19]Behavioral Ecology and Sociobiology Unit, German Primate Center, Leibniz Institute for Primate Research, Göttingen, Germany. [20]University of Warwick, Coventry, UK. [21]Mention Anthropobiologie et Développement Durable, Faculté des Sciences, Université d'Antananarivo, Antananarivo, Madagascar. [22]Ecole Doctorale Ecosystèmes Naturels (EDEN), Université de Mahajanga, Mahajanga, Madagascar. [23]Département de Biologie Animale, Université d'Antananarivo, Antananarivo, Madagascar. [24]UMR ENTROPIE (Université de La Réunion, IRD, CNRS, IFREMER, Université de Nouvelle-Calédonie), Saint-Denis de La Réunion, France. [25]Département des Sciences de la Nature et de l'Environnement, Université d'Antsiranana, Antsiranana, Madagascar. [26]Madagascar Biodiversity Partnership, Antananarivo, Madagascar. [27]Centre for Ecology, Evolution and Environmental Changes (cE3c), Faculdade de Ciências da Universidade de Lisboa, Lisboa, Portugal. [28]These authors contributed equally: Tobias van Elst, Gabriele M. Sgarlata, Dominik Schüßler. ✉e-mail: tob.velst@posteo.de; gabriele.sgarlata@gmail.com; dominik.schuessler@posteo.de; jordi.salmona@ird.fr

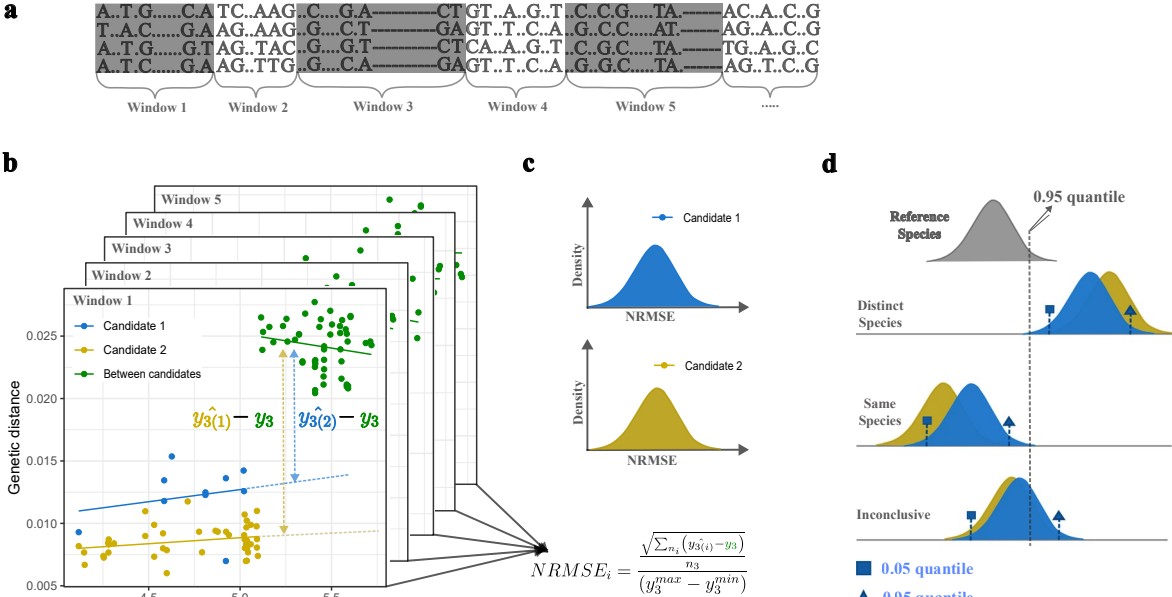

**Extended Data Fig. 1 | Statistical test to distinguish intra- from interspecific divergence for two candidate species based on patterns of isolation by distance. a**, SNP data are divided into windows comprising 1,000 SNPs. **b**, For each window, individual geographic distances are correlated with genetic distances and classified as distances among candidate 1 individuals (blue), among candidate 2 individuals (yellow) or between individuals of the two candidates (green). **c**, Deviations of observed genetic distances between candidates from those predicted by the within-candidate geographic clines in genetic distance are calculated. Accordingly, two normalized root mean square error (NRMSE) values are obtained for each genomic window *j*, one for each candidate *i*, resulting in two NRMSE distributions across genomic windows.

**d**, The resulting distributions are compared to the 0.95 quantiles of reference distributions (taken from *M. tavaratra* and *M. lehilahytsara* in this work). The intraspecific clinal variation model is rejected if the 0.05 quantiles of both NRMSE distributions of a candidate pair are above the 0.95 quantiles of the reference distributions, indicating that genetic distances between candidates cannot be explained by a geographic cline. Conversely, if the 0.95 quantile of a single NRMSE distribution of a candidate pair is below the 0.95 quantiles of the reference distributions, genetic distances are considered to be congruent with a model of intraspecific structure. Cases that are neither rejecting nor congruent with the intraspecific model are considered inconclusive.

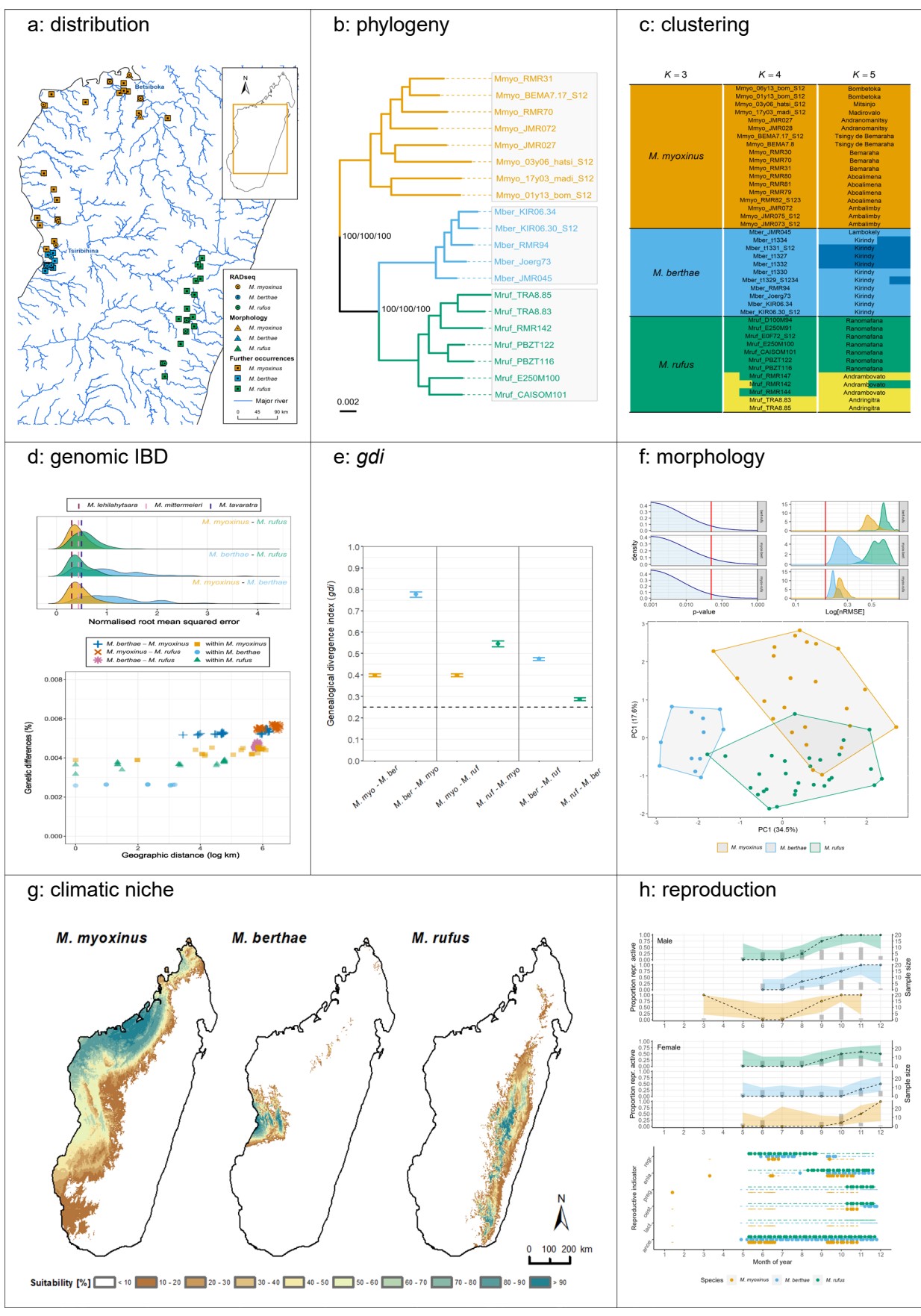

**Extended Data Fig. 2 | See next page for caption.**

**Extended Data Fig. 2 | Summary of species delimitation results for the candidates *M. rufus*, *M. berthae* and *M. myoxinus*. a**, Sampling map. **b**, Phylogeny (node labels represent percent SH-aLRT/ultrafast bootstrap support in IQ-TREE/bootstrap support in SVDquartets and are only given for divergences between candidates; scale is substitutions per site; grey shading indicates evolutionarily significant units). **c**, Admixture proportions assuming 3 to 5 clusters (labels in columns represent candidates, sample names and localities from left to right). **d**, Top: Normalised root mean square error (NRMSE) distributions of within and between candidate isolation by distance (IBD) across genomic windows (colour indicates focal taxon for within candidate IBD; vertical lines indicate different thresholds for species delimitation); bottom: genome-wide patterns of IBD in the candidate group. **e**, Genealogical divergence indices (*gdi*) with 95% highest posterior density (HPD) intervals based on a coalescent model of 6,000 loci and two individuals per species (one individual for

*M. marohita*). **f**, Top: *p*-value distributions of Mantel tests for IBD (left) and NRMSE distributions (log scale) of within and between candidate IBD (right) across morphological resampling (colour indicates focal taxon for within candidate IBD; vertical lines indicate threshold for species delimitation); bottom: PCA bidimensional representation of the morphological variability within and among candidates. **g**, Climatic niche models. **h**, Top: proportion of reproductive individuals for males and females after correction (see Supplementary methods: Species delimitation); grey histograms indicate sample size; bottom: reproductive indicators of sample individuals (dots and dashes indicate presence and absence, respectively; regr.: regressed testes; enla.: enlarged testes; preg.: pregnant; oest.: oestrous; lact.: lactating; anoe.: anoestrous). Sample sizes per species for panels **d**, **f** and **g** are given in Supplementary Tables 2, 4/5 and 6, respectively.

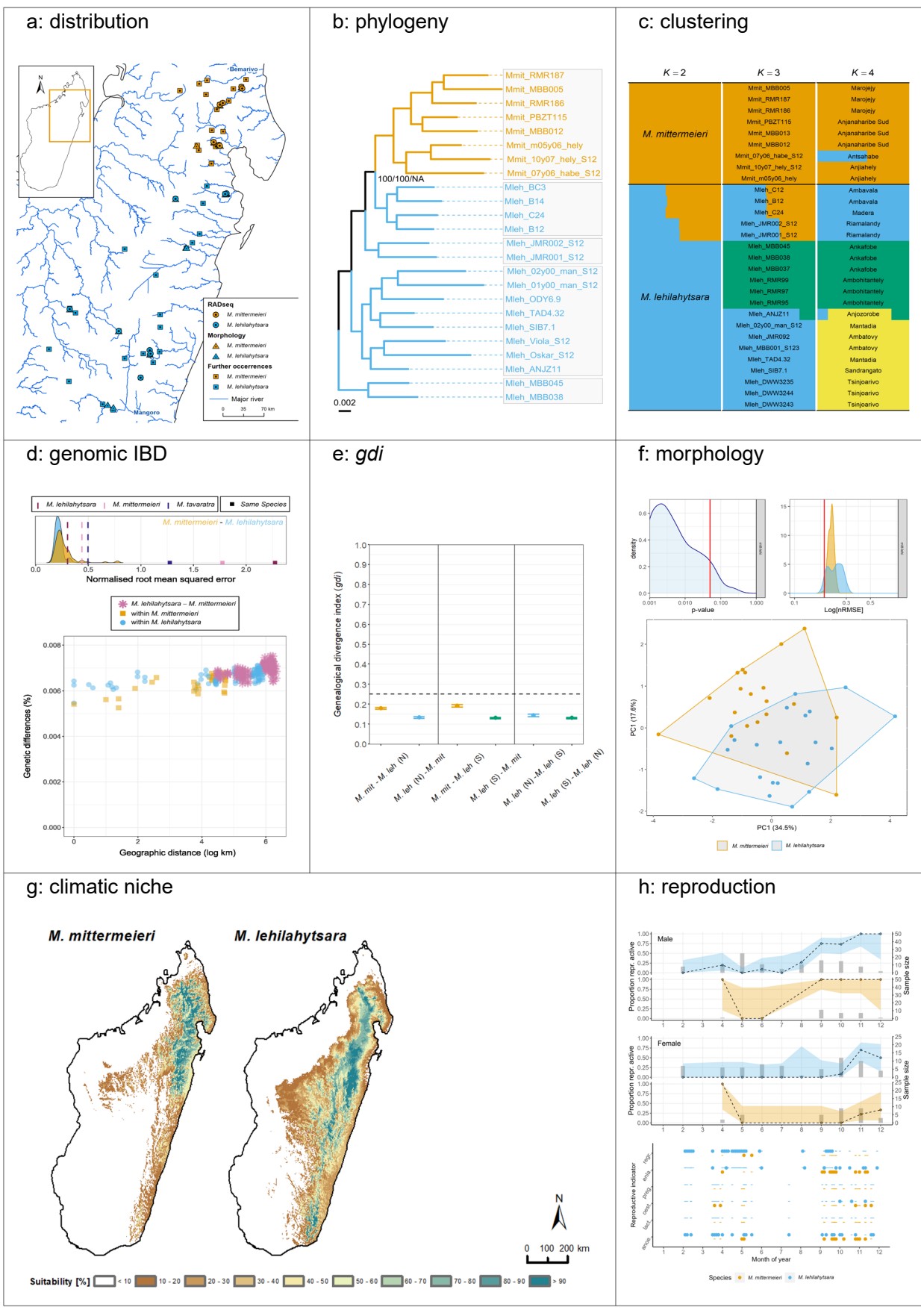

**Extended Data Fig. 3 | See next page for caption.**

**Extended Data Fig. 3 | Summary of species delimitation results for the candidates _M. lehilahytsara_ and _M. mittermeieri_. a**, Sampling map. **b**, Phylogeny (node labels represent percent SH-aLRT/ultrafast bootstrap support in IQ-TREE/bootstrap support in SVDquartets and are only given for divergences between candidates; scale is substitutions per site; grey shading indicates evolutionarily significant units). **c**, Admixture proportions assuming 2 to 4 clusters (labels in columns represent candidates, sample names and localities from left to right). **d**, Top: Normalised root mean square error (NRMSE) distributions of within and between candidate isolation by distance (IBD) across genomic windows (colour indicates focal taxon for within candidate IBD; vertical lines indicate different thresholds for species delimitation); bottom: genome-wide patterns of IBD in the candidate group. **e**, Genealogical divergence indices (_gdi_) with 95% highest posterior density (HPD) intervals based on a coalescent model of 6,000 loci and two individuals per species (one individual for

_M. marohita_). **f**, Top: _p_-value distributions of Mantel tests for IBD (left) and NRMSE distributions (log scale) of within and between candidate IBD (right) across morphological resampling (colour indicates focal taxon for within candidate IBD; vertical lines indicate threshold for species delimitation); bottom: PCA bidimensional representation of the morphological variability within and among candidates. **g**, Climatic niche models. **h**, Top: proportion of reproductive individuals for males and females after correction (see Supplementary methods: Species delimitation); grey histograms indicate sample size; bottom: reproductive indicators of sample individuals (dots and dashes indicate presence and absence, respectively; regr.: regressed testes; enla.: enlarged testes; preg.: pregnant; oest.: oestrous; lact.: lactating; anoe.: anoestrous). Sample sizes per species for panels **d**, **f** and **g** are given in Supplementary Tables 2, 4/5 and 6, respectively.

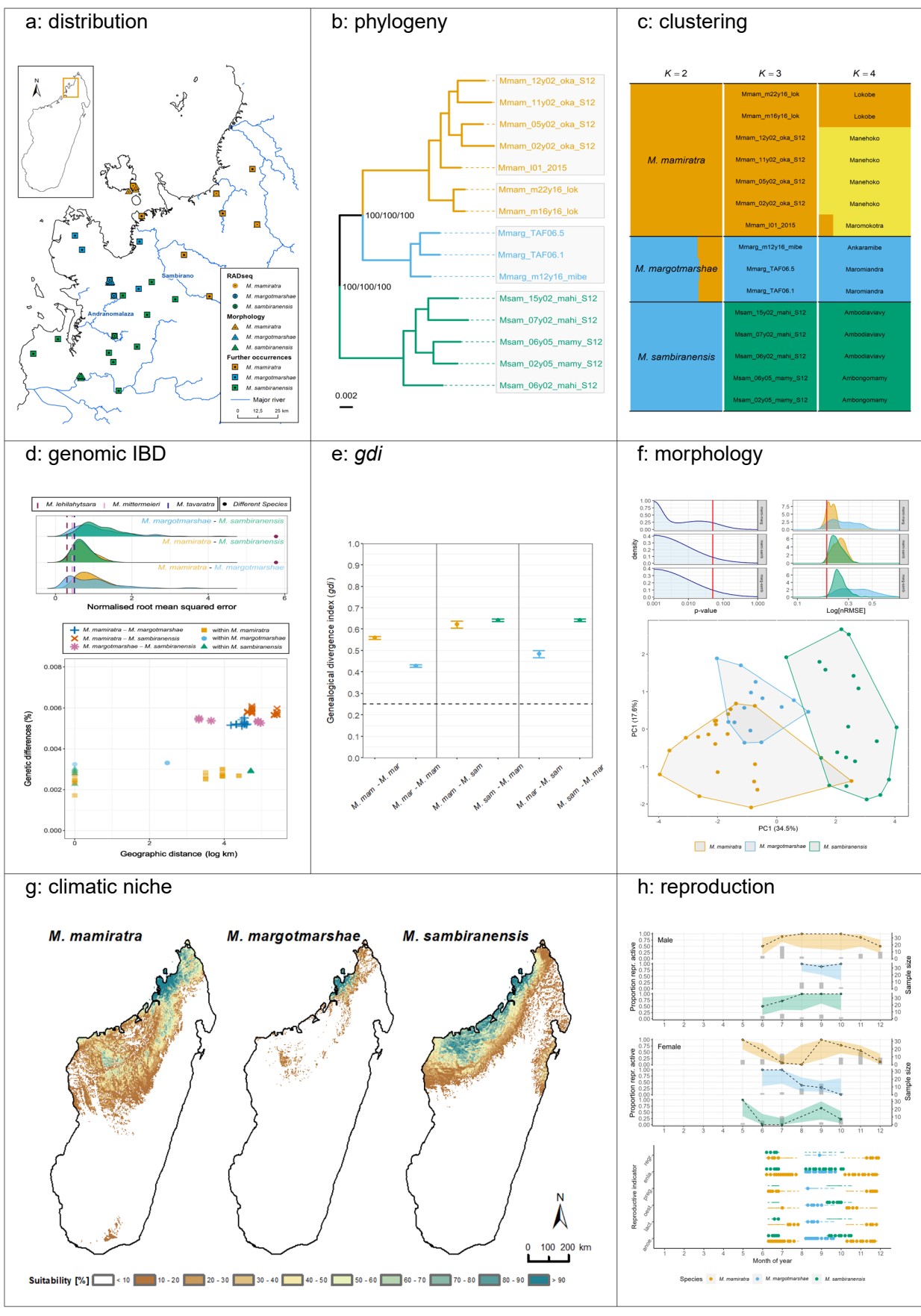

**Extended Data Fig. 4 | See next page for caption.**

**Extended Data Fig. 4 | Summary of species delimitation results for the candidates *M. mamiratra*, *M. margotmarshae* and *M. sambiranensis*.**
**a**, Sampling map. **b**, Phylogeny (node labels represent percent SH-aLRT/ultrafast bootstrap support in IQ-TREE/bootstrap support in SVDquartets and are only given for divergences between candidates; scale is substitutions per site; grey shading indicates evolutionarily significant units). **c**, Admixture proportions assuming 2 to 4 clusters (labels in columns represent candidates, sample names and localities from left to right). **d**, Top: Normalised root mean square error (NRMSE) distributions of within and between candidate isolation by distance (IBD) across genomic windows (colour indicates focal taxon for within candidate IBD; vertical lines indicate different thresholds for species delimitation); bottom: genome-wide patterns of IBD in the candidate group. **e**, Genealogical divergence indices (*gdi*) with 95% highest posterior density (HPD) intervals based on a coalescent model of 6,000 loci and two individuals per species

(one individual for *M. marohita*). **f**, Top: *p*-value distributions of Mantel tests for IBD (left) and NRMSE distributions (log scale) of within and between candidate IBD (right) across morphological resampling (colour indicates focal taxon for within candidate IBD; vertical lines indicate threshold for species delimitation); bottom: PCA bidimensional representation of the morphological variability within and among candidates. **g**, Climatic niche models. **h**, Top: proportion of reproductive individuals for males and females after correction (see Supplementary methods: Species delimitation); grey histograms indicate sample size; bottom: reproductive indicators of sample individuals (dots and dashes indicate presence and absence, respectively; regr.: regressed testes; enla.: enlarged testes; preg.: pregnant; oest.: oestrous; lact.: lactating; anoe.: anoestrous). Sample sizes per species for panels **d**, **f** and **g** are given in Supplementray Tables 2, 4/5 and 6, respectively.

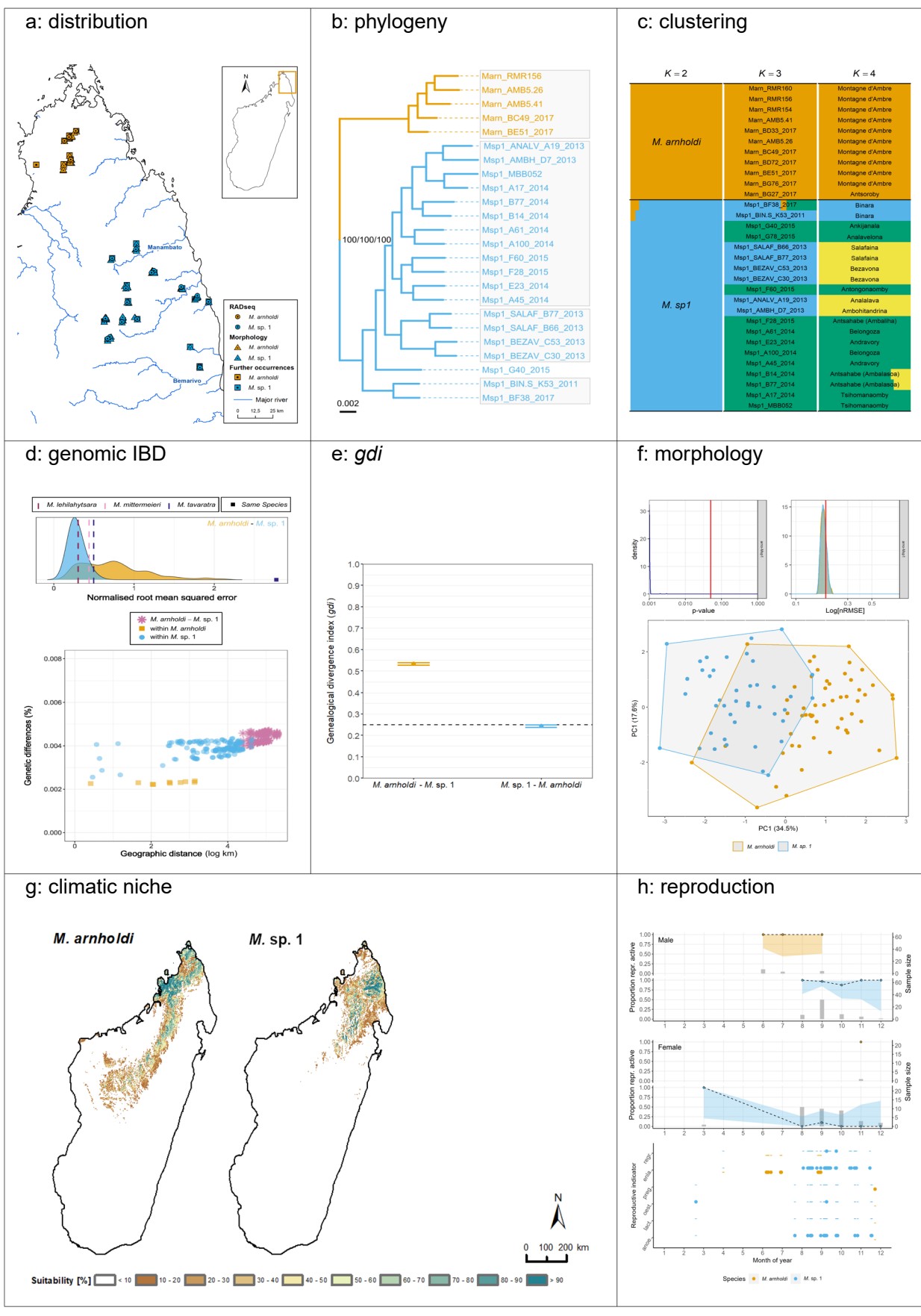

**Extended Data Fig. 5 | See next page for caption.**

**Extended Data Fig. 5 | Summary of species delimitation results for the candidates *M. arnholdi* and *M.* sp. 1. a**, Sampling map. **b**, Phylogeny (node labels represent percent SH-aLRT/ultrafast bootstrap support in IQ-TREE/ bootstrap support in SVDquartets and are only given for divergences between candidates; scale is substitutions per site; grey shading indicates evolutionarily significant units). **c**, Admixture proportions assuming 2 to 4 clusters (labels in columns represent candidates, sample names and localities from left to right). **d**, Top: Normalised root mean square error (NRMSE) distributions of within and between candidate isolation by distance (IBD) across genomic windows (colour indicates focal taxon for within candidate IBD; vertical lines indicate different thresholds for species delimitation); bottom: genome-wide patterns of IBD in the candidate group. **e**, Genealogical divergence indices (*gdi*) with 95% highest posterior density (HPD) intervals based on a coalescent model of 6,000 loci

and two individuals per species (one individual for *M. marohita*). **f**, Top: *p*-value distributions of Mantel tests for IBD (left) and NRMSE distributions (log scale) of within and between candidate IBD (right) across morphological resampling (colour indicates focal taxon for within candidate IBD; vertical lines indicate threshold for species delimitation); bottom: PCA bidimensional representation of the morphological variability within and among candidates. **g**, Climatic niche models. **h**, Top: proportion of reproductive individuals for males and females after correction (see Supplementary methods: Species delimitation); grey histograms indicate sample size; bottom: reproductive indicators of sample individuals (dots and dashes indicate presence and absence, respectively; regr.: regressed testes; enla.: enlarged testes; preg.: pregnant; oest.: oestrous; lact.: lactating; anoe.: anoestrous). Sample sizes per species for panels **d**, **f** and **g** are given in Supplementary Tables 2, 4/5 and 6, respectively.

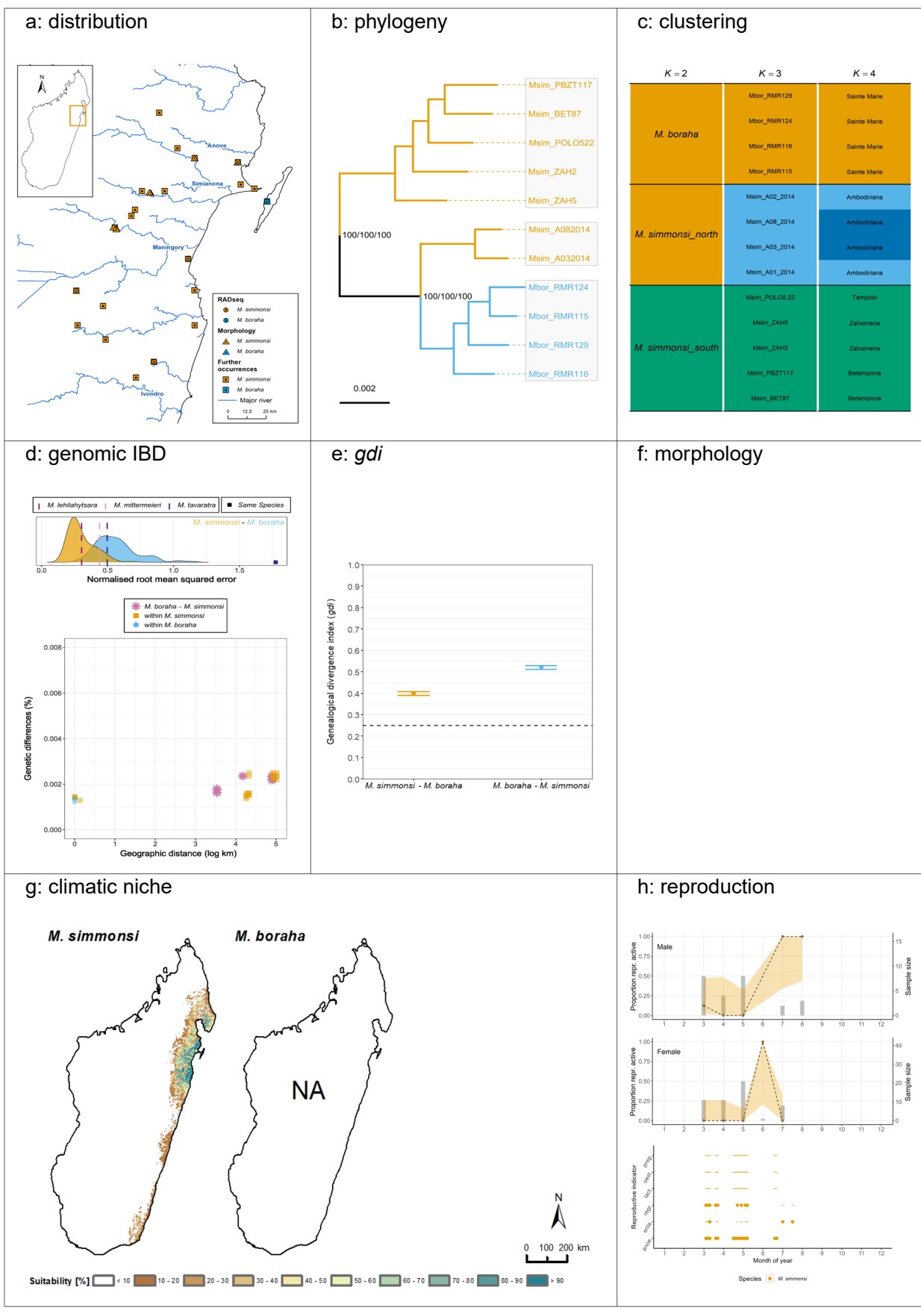

**Extended Data Fig. 6 | See next page for caption.**

**Extended Data Fig. 6 | Summary of species delimitation results for the candidates *M. boraha* and *M. simmonsi*. a**, Sampling map. **b**, Phylogeny (node labels represent percent SH-aLRT/ultrafast bootstrap support in IQ-TREE/ bootstrap support in SVDquartets and are only given for divergences between candidates; scale is substitutions per site; grey shading indicates evolutionarily significant units). **c**, Admixture proportions assuming 2 to 4 clusters (labels in columns represent candidates, sample names and localities from left to right). **d**, Top: Normalised root mean square error (NRMSE) distributions of within and between candidate isolation by distance (IBD) across genomic windows (colour indicates focal taxon for within candidate IBD; vertical lines indicate different thresholds for species delimitation); bottom: genome-wide patterns of IBD in the candidate group. **e**, Genealogical divergence indices (*gdi*) with 95% highest posterior density (HPD) intervals based on a coalescent model of 6,000 loci and two individuals per species (one individual for *M. marohita*). **f**, Comprehensive morphometric data are lacking for these candidates. **g**, Climatic niche models. **h**, Top: proportion of reproductive individuals for males and females after correction (see Supplementary methods: Species delimitation); grey histograms indicate sample size; bottom: reproductive indicators of sample individuals (dots and dashes indicate presence and absence, respectively; regr.: regressed testes; enla.: enlarged testes; preg.: pregnant; oest.: oestrous; lact.: lactating; anoe.: anoestrous). Sample sizes per species for panels **d**, **f** and **g** are given in Supplementary Tables 2, 4/5 and 6, respectively.

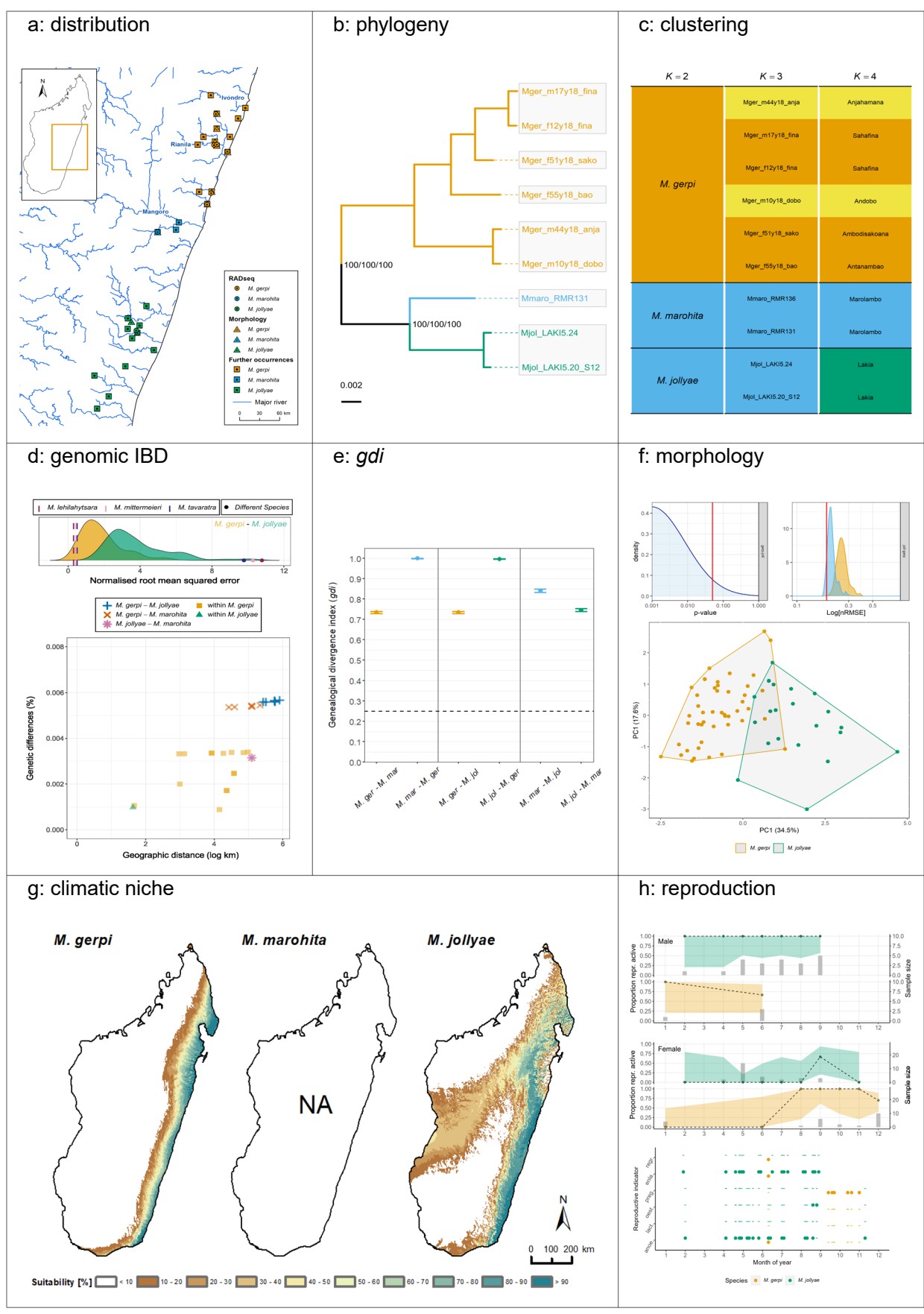

**Extended Data Fig. 7 | See next page for caption.**

**Extended Data Fig. 7 | Summary of species delimitation results for the candidates *M. jollyae*, *M. marohita* and *M. gerpi*. a**, Sampling map. **b**, Phylogeny (node labels represent percent SH-aLRT/ultrafast bootstrap support in IQ-TREE/bootstrap support in SVDquartets and are only given for divergences between candidates; scale is substitutions per site; grey shading indicates evolutionarily significant units). **c**, Admixture proportions assuming 2 to 4 clusters (labels in columns represent candidates, sample names and localities from left to right). **d**, Top: Normalised root mean square error (NRMSE) distributions of within and between candidate isolation by distance (IBD) across genomic windows (colour indicates focal taxon for within candidate IBD; vertical lines indicate different thresholds for species delimitation); bottom: genome-wide patterns of IBD in the candidate group. **e**, Genealogical divergence indices (*gdi*) with 95% highest posterior density (HPD) intervals based on a coalescent model of 6,000 loci and two individuals per species (one individual for

*M. marohita*). **f**, Top: *p*-value distributions of Mantel tests for IBD (left) and NRMSE distributions (log scale) of within and between candidate IBD (right) across morphological resampling (colour indicates focal taxon for within candidate IBD; vertical lines indicate threshold for species delimitation); bottom: PCA bidimensional representation of the morphological variability within and among candidates. **g**, Climatic niche models. **h**, Top: proportion of reproductive individuals for males and females after correction (see Supplementary methods: Species delimitation); grey histograms indicate sample size; bottom: reproductive indicators of sample individuals (dots and dashes indicate presence and absence, respectively; regr.: regressed testes; enla.: enlarged testes; preg.: pregnant; oest.: oestrous; lact.: lactating; anoe.: anoestrous). Sample sizes per species for panels **d**, **f** and **g** are given in Supplementary Tables 2, 4/5 and 6, respectively.

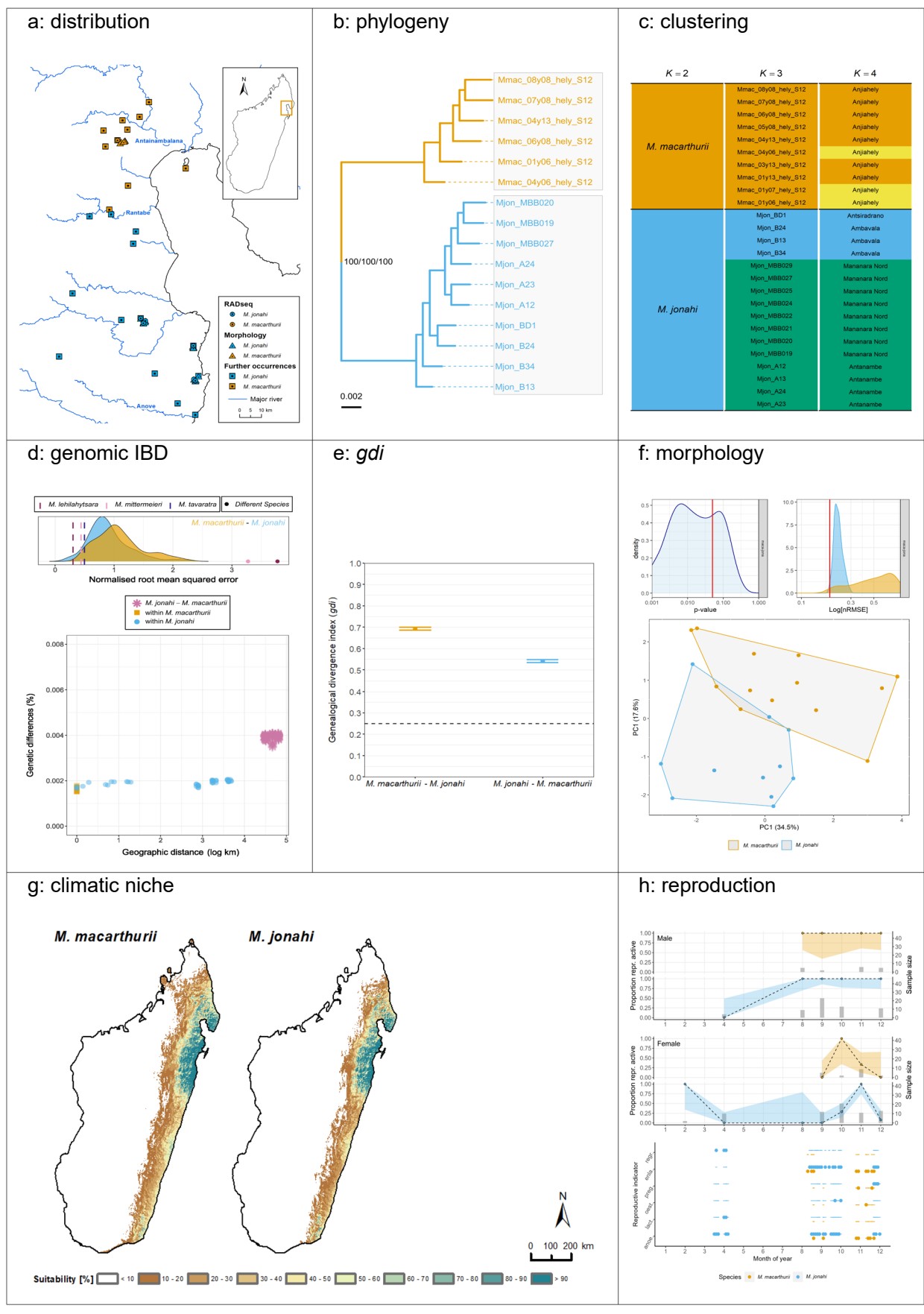

**Extended Data Fig. 8 | See next page for caption.**

**Extended Data Fig. 8 | Summary of species delimitation results for the candidates _M. macarthurii_ and _M. jonahi._ a**, Sampling map. **b**, Phylogeny (node labels represent percent SH-aLRT/ultrafast bootstrap support in IQ-TREE/ bootstrap support in SVDquartets and are only given for divergences between candidates; scale is substitutions per site; grey shading indicates evolutionarily significant units). **c**, Admixture proportions assuming 2 to 4 clusters (labels in columns represent candidates, sample names and localities from left to right). **d**, Top: Normalised root mean square error (NRMSE) distributions of within and between candidate isolation by distance (IBD) across genomic windows (colour indicates focal taxon for within candidate IBD; vertical lines indicate different thresholds for species delimitation); bottom: genome-wide patterns of IBD in the candidate group. **e**, Genealogical divergence indices (_gdi_) with 95% highest posterior density (HPD) intervals based on a coalescent model of 6,000 loci

and two individuals per species (one individual for _M. marohita_). **f**, Top: _p_-value distributions of Mantel tests for IBD (left) and NRMSE distributions (log scale) of within and between candidate IBD (right) across morphological resampling (colour indicates focal taxon for within candidate IBD; vertical lines indicate threshold for species delimitation); bottom: PCA bidimensional representation of the morphological variability within and among candidates. **g**, Climatic niche models. **h**, Top: proportion of reproductive individuals for males and females after correction (see Supplementary methods: Species delimitation); grey histograms indicate sample size; bottom: reproductive indicators of sample individuals (dots and dashes indicate presence and absence, respectively; regr.: regressed testes; enla.: enlarged testes; preg.: pregnant; oest.: oestrous; lact.: lactating; anoe.: anoestrous). Sample sizes per species for panels **d**, **f** and **g** are given in Supplementary Tables 2, 4/5 and 6, respectively.

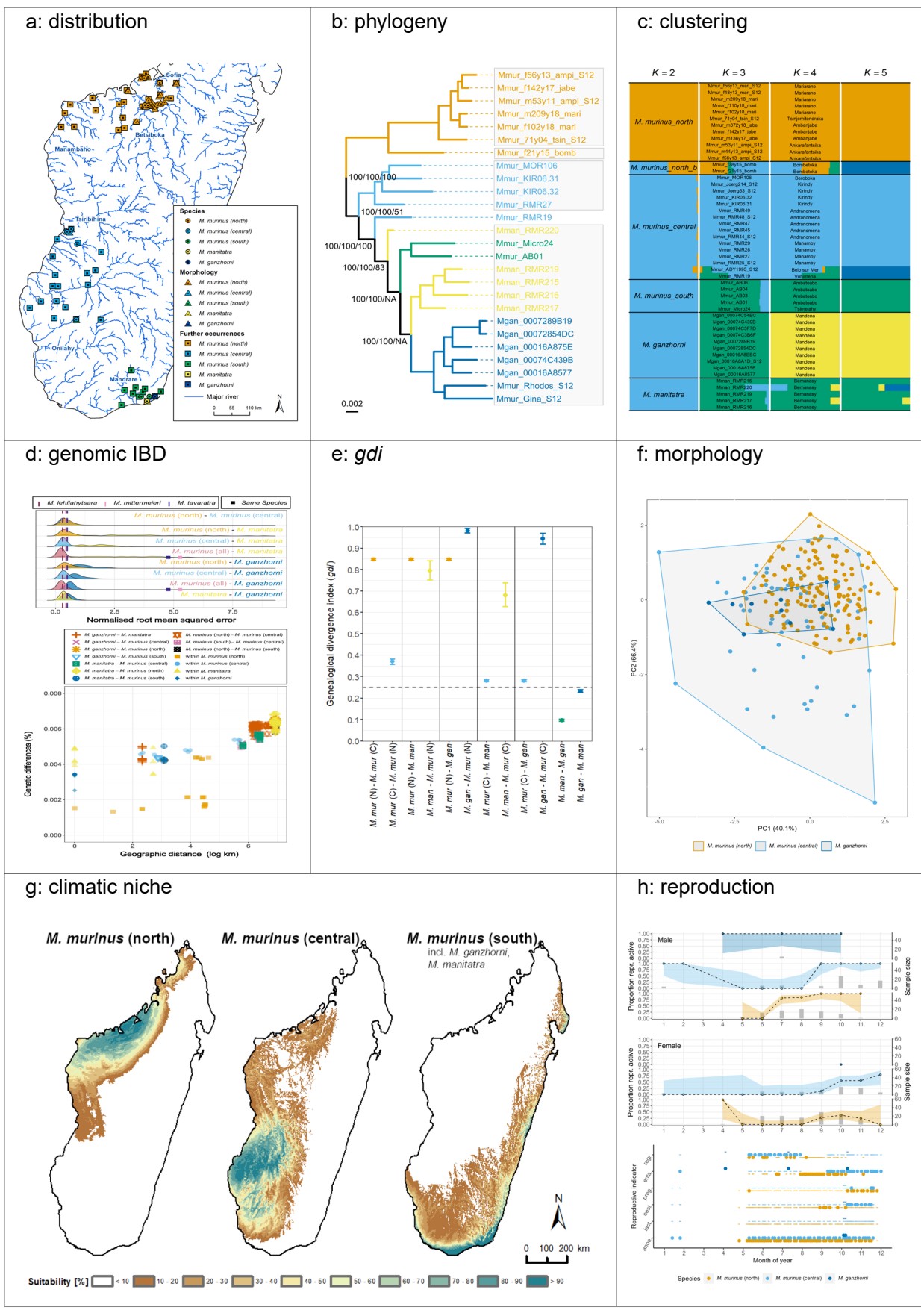

**Extended Data Fig. 9 | See next page for caption.**

**Extended Data Fig. 9 | Summary of species delimitation results for the candidates *M. manitatra*, *M. ganzhorni* and *M. murinus*. a**, Sampling map. **b**, phylogeny (node labels represent percent SH-aLRT/ultrafast bootstrap support in IQ-TREE/bootstrap support in SVDquartets and are only given for divergences between candidates; scale is substitutions per site; grey shading indicates evolutionarily significant units). **c**, Admixture proportions assuming 2 to 5 clusters (labels in columns represent candidates, sample names and localities from left to right). **d**, Top: Normalised root mean square error (NRMSE) distributions of within and between candidate isolatino by distance (IBD) across genomic windows (colour indicates focal taxon for within candidate IBD; vertical lines indicate different thresholds for species delimitation); bottom: genome-wide patterns of IBD in the candidate group. **e**, Genealogical divergence indices (*gdi*) with 95% highest posterior density (HPD) intervals based on a coalescent model of 6,000 loci and two individuals per species (one individual for *M. marohita*). **f**, PCA bidimensional representation of the morphological variability within and among candidates; analyses of IBD of morphometry were not conducted due to lack of data. **g**, Climatic niche models. **h**, Top: proportion of reproductive individuals for males and females after correction (see Supplementary methods: Species delimitation); grey histograms indicate sample size; bottom: reproductive indicators of sample individuals (dots and dashes indicate presence and absence, respectively; regr.: regressed testes; enla.: enlarged testes; preg.: pregnant; oest.: oestrous; lact.: lactating; anoe.: anoestrous). Sample sizes per species for panels **d**, **f** and **g** are given in Supplementary Tables 2, 4/5 and 6, respectively.

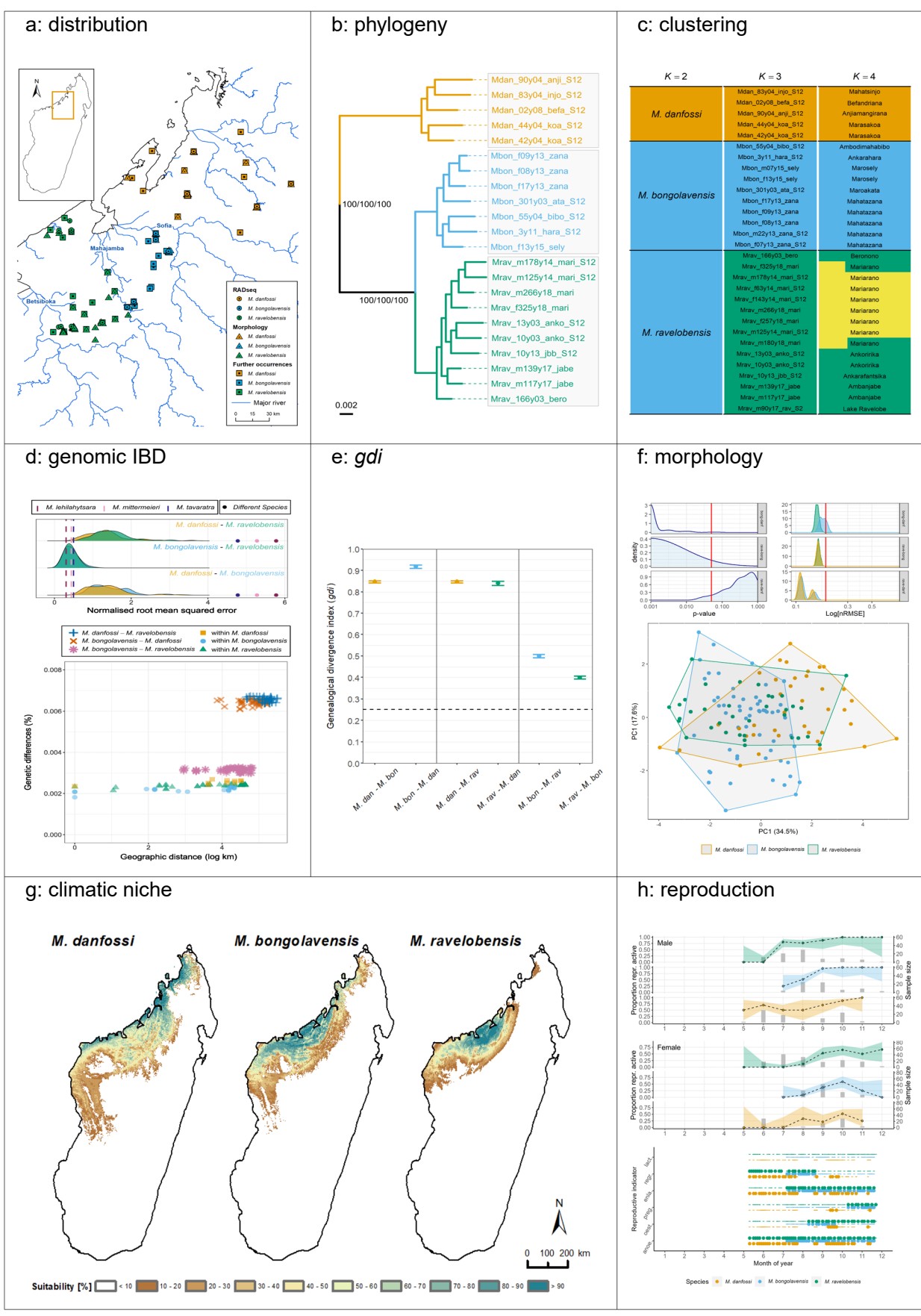

**Extended Data Fig. 10 | See next page for caption.**

**Extended Data Fig. 10 | Summary of species delimitation results for the candidates *M. ravelobensis*, *M. bongolavensis* and *M. danfossi*. a**, Sampling map. **b**, Phylogeny (node labels represent percent SH-aLRT/ultrafast bootstrap support in IQ-TREE/bootstrap support in SVDquartets and are only given for divergences between candidates; scale is substitutions per site; grey shading indicates evolutionarily significant units). **c**, Admixture proportions assuming 2 to 4 clusters (labels in columns represent candidates, sample names and localities from left to right). **d**, Top: Normalised root mean square error (NRMSE) distributions of within and between candidate isolation by distance (IBD) across genomic windows (colour indicates focal taxon for within candidate IBD; vertical lines indicate different thresholds for species delimitation); bottom: genome-wide patterns of IBD in the candidate group. **e**, Genealogical divergence indices (*gdi*) with 95% highest posterior density (HPD) intervals based on a coalescent model of 6,000 loci and two individuals per species (one individual for

*M. marohita*). **f**, Top: *p*-value distributions of Mantel tests for IBD (left) and NRMSE distributions (log scale) of within and between candidate IBD (right) across morphological resampling (colour indicates focal taxon for within candidate IBD; vertical lines indicate threshold for species delimitation); bottom: PCA bidimensional representation of the morphological variability within and among candidates. **g**, Climatic niche models. **h**, Top: proportion of reproductive individuals for males and females after correction (see Supplementary methods: Species delimitation); grey histograms indicate sample size; bottom: reproductive indicators of sample individuals (dots and dashes indicate presence and absence, respectively; regr.: regressed testes; enla.: enlarged testes; preg.: pregnant; oest.: oestrous; lact.: lactating; anoe.: anoestrous). Sample sizes per species for panels **d**, **f** and **g** are given in Supplementary Tables 2, 4/5 and 6, respectively.

# Reporting Summary

## Statistics

For all statistical analyses, confirm that the following items are present in the figure legend, table legend, main text, or Methods section.

| n/a | Confirmed | |
|---|---|---|
| ☐ | ☒ | The exact sample size (*n*) for each experimental group/condition, given as a discrete number and unit of measurement |
| ☐ | ☒ | A statement on whether measurements were taken from distinct samples or whether the same sample was measured repeatedly |
| ☐ | ☒ | The statistical test(s) used AND whether they are one- or two-sided *Only common tests should be described solely by name; describe more complex techniques in the Methods section.* |
| ☐ | ☒ | A description of all covariates tested |
| ☐ | ☒ | A description of any assumptions or corrections, such as tests of normality and adjustment for multiple comparisons |
| ☐ | ☒ | A full description of the statistical parameters including central tendency (e.g. means) or other basic estimates (e.g. regression coefficient) AND variation (e.g. standard deviation) or associated estimates of uncertainty (e.g. confidence intervals) |
| ☐ | ☒ | For null hypothesis testing, the test statistic (e.g. *F*, *t*, *r*) with confidence intervals, effect sizes, degrees of freedom and *P* value noted *Give P values as exact values whenever suitable.* |
| ☐ | ☒ | For Bayesian analysis, information on the choice of priors and Markov chain Monte Carlo settings |
| ☒ | ☐ | For hierarchical and complex designs, identification of the appropriate level for tests and full reporting of outcomes |
| ☐ | ☒ | Estimates of effect sizes (e.g. Cohen's *d*, Pearson's *r*), indicating how they were calculated |

*Our web collection on statistics for biologists contains articles on many of the points above.*

## Software and code

Policy information about availability of computer code

| Data collection | No software was used for data collection. |
|---|---|
| Data analysis | RAD genotyping: Stacks v2.0b, Trimmomatic v0.39, BWA-MEM v0.7.17, SAMtools v1.11, GATK v4.1.9.0, VCFtools v0.1.17, MUSCLE v3.8.31, ANGSD v0.92<br>Phylogenetic inference: IQ-TREE v2.2.0, PAUP* v4.0a<br>Species delimitation: NGSadmix v32, PRAAT v5.4.0.4, R packages 'lostruct' v0.0.0.9000, 'dynRB' v0.18, 'ENMtools' v1.0.7, 'ENMeval' v2.0, 'binom' v1.1-1.1<br>Divergence time estimation: BPP v4.4.1, Tracer v1.7.2<br>Biogeographic reconstruction: R package 'BioGeoBears' v1.1.2<br>Modelling morphological and climatic niche evolution: R packages 'phytools' v2.3-0, 'dynRB' v0.18, 'phyloclim' v0.9.5, 'mvMORPH' v1.1.9, 'tmvtnorm' v1.6<br>Conservation reassessment: ArcGIS Pro v3.1.0<br>All custom analysis scripts can be found at https://github.com/t-vane/van_Elst_et_al_2024_Cryptic_diversification. |

For manuscripts utilizing custom algorithms or software that are central to the research but not yet described in published literature, software must be made available to editors and reviewers. We strongly encourage code deposition in a community repository (e.g. GitHub). See the Nature Portfolio guidelines for submitting code & software for further information.

# Data

Policy information about availability of data

All manuscripts must include a data availability statement. This statement should provide the following information, where applicable:
- Accession codes, unique identifiers, or web links for publicly available datasets
- A description of any restrictions on data availability
- For clinical datasets or third party data, please ensure that the statement adheres to our policy

All new sequencing data have been made available through NCBI BioProjects PRJNA560399 and PRJNA807164. Individual BioSample accessions are given in Supplementary Table 13.
Bioclimatic data were extracted from CHELSEA database v2.1 (https://envicloud.wsl.ch/#/?prefix=chelsa%2Fchelsa_V1%2Fclimatologies).
Occurrence records are given in Supplementary Table 15.
Reproductive data are given in Supplementary Table 16.
Morphometric and acoustic raw data as well as all analysis input, output and configuration files are available at Dryad (https://doi.org/10.5061/dryad.b2rbnzsp3).

# Research involving human participants, their data, or biological material

Policy information about studies with human participants or human data. See also policy information about sex, gender (identity/presentation), and sexual orientation and race, ethnicity and racism.

| Reporting on sex and gender | Not applicable. |
| Reporting on race, ethnicity, or other socially relevant groupings | Not applicable. |
| Population characteristics | Not applicable. |
| Recruitment | Not applicable. |
| Ethics oversight | Not applicable. |

Note that full information on the approval of the study protocol must also be provided in the manuscript.

# Field-specific reporting

Please select the one below that is the best fit for your research. If you are not sure, read the appropriate sections before making your selection.

☐ Life sciences  ☐ Behavioural & social sciences  ☒ Ecological, evolutionary & environmental sciences

For a reference copy of the document with all sections, see nature.com/documents/nr-reporting-summary-flat.pdf

# Ecological, evolutionary & environmental sciences study design

All studies must disclose on these points even when the disclosure is negative.

| Study description | We developed a practical integrative framework following that considers genomic, bioclimatic, morphometric, and behavioural data to consistently delimit species across taxonomically challenging groups. We applied this framework to the genus Microcebus, including all 25 named species with extensive geographic sampling. We inferred the temporal evolution of habitat and climatic niche, and their combined impacts on morphological stasis through time. Additionally, we highlighted the consequences for conservation status and identified phylogeographic conservation units. |
| Research sample | The genus Microcebus was chosen as model to illustrate the presented integrative approach to taxonomy and diversification research due to its cryptic diversity, controversial taxonomy and relative wealth of available data. We compiled all available information across multiples lines of evidence (genomics, morphometrics, bioclimate, reproduction, accoustic communication) for the genus from the literature and our own research. Accordingly, the research sample aimed to be as comprehensive as possible, covering all described and one putative Microcebus species. |
| Sampling strategy | Sampling aimed to be comprehensive accounting for the full geno- and phenotypic diversity known in the genus Microcebus. It therefore covered all described and assumed distributions of Microcebus species across Madagascar. |
| Data collection | Microcebus samples were obtained by numerous researchers (given in Supplementary Tables 13 to 17). This included taking tissues (i.e.,ear biopsies) for genomic analyses, morphometric measurements, bioclimatic data, sex, and data on acoustic communication. Sampling procedures for each of these are given in the Methods in the manuscript. |
| Timing and spatial scale | Microcebus individuals were sampled between 1994 and 2022 across all forested ecosystems of Madagascar. Sampling localities and dates are given in Tables 13 to 17. |

| | |
|---|---|
| Data exclusions | For genomic analyses, several commonly used filters were applied on both individual and genotype level to limit bias from data quality (sequencing depth and quality; described in Methods). Morphometric data were filtered following Schüßler et al. (2023, AJBA; see Methods). Bioclimatic data was filtered on individual level to reduce autocorrelation. |
| Reproducibility | All scripts, input, output and configuration files as well as intermediate results are made available through online repositories, promoting full reproducibility of our study. |
| Randomization | Samples were not randomized as no manipulative experiments were carried out. Individuals were assigned to candidate species based on geographic location, preliminary identification of the respective field primatologist and on previous sequencing activites in different laboratories. |
| Blinding | Not applicable. |

Did the study involve field work?  ☒ Yes  ☐ No

# Field work, collection and transport

| | |
|---|---|
| Field conditions | Field work was carried out over the course of approx. 30 years during both dry and rainy season. Detailed sampling dates are given in Supplementary Tables 13 to 17. |
| Location | Field work was carried out in forested ecosystems across Madagascar. Detailed sampling locations are given in Supplementary Tables 13 to 17. |
| Access & import/export | Research was conduted with the permission of the following authorities and institutions: The Direction Générale du Ministère de l'Environnement et des Forêts de Madagascar, and Madagascar's Ad Hoc Committee for Fauna and Flora and Organisational Committee for Environmental Research (CAFF/CORE). Associated permit numbers are: [87,233,329]/06/10/MEF/SG/DCB.SAP/SCB 121/07/MEF/SG/DCB.SAP/SCB [026,227]/08/10/MEF/SG/DCB.SAP/SCB [218,220]/10/MEF/SG/DCB.SAP/SCB [113,118,186]/11/MEF/SG/DCB.SAP/SCB [074,099,124]/12/MEF/SG/DCB.SAP/SCB [137,175,178,179,186]/13/MEF/SG/DCB.SAP/SCB [150,168,169,137]/13/MEF/SG/DGF/DCB.SAP/SCB 175/14/MEF/SG/DCB.SAP/SCB 072/15/MEEMF/SG/DGF/DCB.SAP/SCB 074/15/MEEEMEF/SG/DGF/DCB.SAP/SCD [169,270]/16/MEEF/SG/DGF/DSAP/SCB.Re 130/16/MEEF/SG/DGF/DAPT/SCBT.Re [044,078,079,136,197]/17/MEEF/SG/DGF/DSAP/SCB.Re [080,151]/17/MEF/SG/DGF/DSAP/SCB.Rc [035,084,093]/18/MEF/ SG/DGF/DSAP/SCB.Rc. 013/19/MEEF/SG/DGF/DSAP/SCB.Re 169/19/MEDD/SG/DGEF/DGRNE 349/21/MEDD/SG/DGGE/DAPRNE/SCBE.Re [015,030]/22/MEDD/SG/DGGE/DAPRNE/SCBE.Re [023,037,071,072,181]/MINENV.EF/SG/DGEF/DADF/SCB [093,179,230]/MINENV.EF/SG/DGEF/DPB/SCBLF [075,177]/MINENV.EF/SG/DGEF/DPB/SCBLF/RECH [87,233,329]/06/10/MEF/SG/DCB.SAP/SCB 121/07/MEF/SG/DCB.SAP/SCB [026,227]/08/10/MEF/SG/DCB.SAP/SCB |
| Disturbance | Disturbance was minimized by releasing captured Microcebus individuals within 24 hours of capture. |

# Reporting for specific materials, systems and methods

We require information from authors about some types of materials, experimental systems and methods used in many studies. Here, indicate whether each material, system or method listed is relevant to your study. If you are not sure if a list item applies to your research, read the appropriate section before selecting a response.

## Materials & experimental systems

| n/a | Involved in the study |
|-----|----------------------|
| ☒ | Antibodies |
| ☒ | Eukaryotic cell lines |
| ☒ | Palaeontology and archaeology |
| ☐ | ☒ Animals and other organisms |
| ☒ | Clinical data |
| ☒ | Dual use research of concern |
| ☒ | Plants |

## Methods

| n/a | Involved in the study |
|-----|----------------------|
| ☒ | ChIP-seq |
| ☒ | Flow cytometry |
| ☒ | MRI-based neuroimaging |

# Animals and other research organisms

Policy information about studies involving animals; ARRIVE guidelines recommended for reporting animal research, and Sex and Gender in Research

| Laboratory animals | No laboratory animals were used in this study. |
|---|---|
| Wild animals | Wild mouse lemurs were captured during night by hand or with Sherman traps baited with banana. Capture procedures are reported and referenced in the Methods of the manuscript. After transport to the camping site in Sherman traps, Microcebus individuals were measured (morphometrics) and tissue samples (i.e., ear biopsies) were taken. Individuals were released within 24 hours of capture at the respective capture location. Species and/or developmental stage of sampled mouse lemurs are provided in Supplementary Tables 13 to 17. |
| Reporting on sex | Sex of animals was only relevant to the analysis of reproductive activity and is reported for all sampled animals in Supplementary Tables 13 to 16. |
| Field-collected samples | Tissue samples were stored in Queen's lysis buffer or 70% ethanol. They were preserved at room temperature during the field season and were frozen at -20°C afterwards and until DNA extraction. |
| Ethics oversight | This study was conducted in agreement with the laws of Madagascar and adhered to the principles of the Code of Best Practices for Field Primatology of the International Primatological Society and the ethical guidelines of the Association for the Study of Animal Behaviour and the Animal Behavior Society. All capture and handling procedures followed routine protocols and were approved by the Malagasy Authorities. |

Note that full information on the approval of the study protocol must also be provided in the manuscript.

# Plants

| Seed stocks | Not applicable. |
|---|---|
| Novel plant genotypes | Not applicable. |
| Authentication | Not applicable. |

