## [Peer Review File · Nature Ecology & Evolution]

Peer Review Information

Journal: Nature Ecology & Evolution

Manuscript Title: Integrative taxonomy clarifies the evolution of a cryptic primate clade

Corresponding author name(s): Gabriele M. Sgarlata, Dominik Schüßler, Jordi Salmons

Editorial Notes:

Reviewer Comments & Decisions:

Decision Letter, initial version:

28th March 2023

Dear Dr Salmona

Thank you very much for your enquiry about submitting a manuscript to Nature Ecology & Evolution.

I've now had a chance to discuss your work with my colleagues, and although we think that it sounds very interesting, we are still uncertain as to the degree to which the study will be a good fit for the journal.

Therefore, we would like to invite you to submit the full manuscript to Nature Ecology & Evolution so that we can examine the data before deciding whether to send the paper out to review.

If this is acceptable to you, you can submit the complete manuscript using the link below:

[REDACTED]

If you have any questions, please feel free to contact me.

[REDACTED]

Decision Letter, first revision:

10th January 2024

Dear Dr Salmona,

2Your manuscript entitled "An integrative and generalizable approach to elucidate cryptic diversifications sheds light on mouse lemur taxonomy and evolution" has now been seen by three reviewers, whose comments are attached. The reviewers have raised a number of concerns which will need to be addressed before we can offer publication in Nature Ecology & Evolution. We will therefore need to see your responses to the criticisms raised and to some editorial concerns, along with a revised manuscript, before we can reach a final decision regarding publication.

While we feel one of the strengths of the paper is its combined exploration of cryptic species & species delimitation and evolutionary dynamics (and do not advise you to split the paper up), we agree with reviewer 2 that these sections could be better integrated and the importance of what one aspect has to say about another could be more clearly described.

We therefore invite you to revise your manuscript taking into account all reviewer and editor comments. Please highlight all changes in the manuscript text file.

* If you have not done so already please begin to revise your manuscript so that it conforms to our Article format instructions at <http://www.nature.com/natecolevol/info/final-submission>. Refer also to any guidelines provided in this letter.

[REDACTED]

Nature Ecology & Evolution is committed to improving transparency in authorship. As part of our efforts in this direction, we are now requesting that all authors identified as 'corresponding author' on published papers create and link their Open Researcher and Contributor Identifier (ORCID) with their account on the Manuscript Tracking System (MTS), prior to acceptance. ORCID helps the scientific community achieve unambiguous attribution of all scholarly contributions. You can create and link your ORCID from the home page of the MTS by clicking on 'Modify my Springer Nature account'. For more information please visit please visit www.springernature.com/orcid.

[REDACTED]

Reviewer expertise:

Reviewer #1: statistical genetics and phylogenetic inference

Reviewer #2: cryptic speciation and genetics

Reviewer #3: prosimian systematics, morphometrics and ecology

Reviewers' comments:

Reviewer #1 (Remarks to the Author):

In this paper Van Elst assembled and analyzed large sets of genomic, bioclimatic, morphometric, behavioural and acoustic data to characterize the diversity in the mouse lemurs (*Microcebus*), in particular to fix the problem of taxonomic inflation in the group. I think a main strength of the paper is the amount and diversity of data, and the use of a great many software tools.

I have a few minor comments. I should note that I am not knowledgeable enough to comment on many aspects of the paper, and my comments are limited to the analysis of genomic data.

a. The authors identified two well-recognised cases of within-species population structure (instead of distinct species) in the group: (i) *M. lehilahytsara* and *M. Mittermeieri*, which have been synonymized, and (ii) widely distributed populations of *M. tavaratra*. These are used to generate null distributions, with the rationale that a pair of candidate species/hypothesised species/populations showing similar population structures, isolation-by-distance patterns, etc. to those two cases should be lumped into one species. I suggest that the authors state this analysis framework explicitly in the Introduction, so that biologists working on other species may be able to judge whether the same strategy is applicable to their own data. Given that the mouse lemurs are very close, this framework may be justifiable. However, there may be huge variations in population size, in geographic distributions and overlaps etc., between pairs of good species even within the same genus, so that cutoffs identified for one good species may not apply to another.

b. The filters used to compile the genomic data are not always clearly described. Some of them seem unnecessarily complicated and ritualistic. I have a few examples below.

My other comments are sorted by page.

p.11 l.389- “After removing indels, only sites with a minimum global depth of 10, with a maximum global depth of the sum of the 0.995 quantiles of per-individual sequencing depth distributions, and represented in at least three individuals, were retained. In addition, sites with a minimum per-individual depth lower than two or a maximum per-individual depth larger than the maximum 0.995 quantile of per-individual sequencing depth distributions”

Is “global depth” the sum of read depths over all samples, and if so, over how many samples? The read depth seems low, although we do not have a good understanding of how the downstream analysis is

4affected by low coverage.

“the maximum 0.995 quantile of per-individual sequencing depth distributions” sounds very complicated.

p.13 l.443 “The absence of reciprocal monophyly was considered sufficient evidence for synonymization.”

p.13 l.447 “The presence of individuals with admixed ancestry was considered sufficient evidence for synonymization.”

Perhaps some simple justifications or comments are called for here. Some good species do not show reciprocal monophyly, and some good species are involved in gene flow.

p.13 l.455 “we used the gdi only to synonymize (if below 0.25) but not to confirm candidate species.” Jackson et al.’s rule of thumb is that “gdi values less than 0.2 suggest that a single species exists”. Here the authors used 0.25. Is this because the objective of the paper is to “synonymize” candidate species?

p.13 l.462 “We divided genomic data into contiguous windows containing a fixed number of SNPs ...” It should be better to use a fixed window size (10kb, 100kb, etc.). Choosing the window size to have a fixed number of SNPs complicates the statistic test, although this is a problem with Li & Ralph (2019).

p.14 l.470 “the average number of nucleotide differences between two individuals (π)” Should this be “the average proportion of nucleotide differences”?

p.15 l.519 “by resampling 90% of the data 100 times. Confidence intervals of morphometric overlap...” Is this some kind of jackknifing. If so, how do you apply a correction (for the smaller jackknife samples than the original data)?

Reviewer #2 (Remarks to the Author):

This manuscript combines newly-collected genetic data with phenotypic data to provide an integrative taxonomy of the mouse lemurs of Madagascar. The study then uses this new taxonomy to understand the diversification

of the mouse lemurs, including their phenotypic evolution, and to propose revised conservation aims.

Overall, this paper was exceptionally well-written. The ideas were well-organized, the flow was coherent

5throughout, and the language was thoughtful and well-reasoned. Most approaches seem well-considered and properly implemented. In particular, I enjoyed how the study presents clade-by-clade analyses in the Suppl Info & then highlights a few exemplar clades in the main text.

I have a few major comments for consideration.

First, this study felt like two separate papers to me. The section on cryptic species & species delimitation felt a bit disconnected from the section on evolutionary dynamics. The study could simply be split in two, but I suspect the authors combined these two sections for a reason. So, perhaps instead more could be done to discuss how the revised taxonomy influences our interpretation of evolutionary dynamics. (The study has started to do this by exploring how the revised taxonomy affects conservation priorities.)

Second, the study mentions several times that their taxonomic approach is "novel". For example, in L190 - 192, they suggest this framework could also be applied to other controversial delimitations. I am not sure I understand what aspect of it is novel, as most of what they implement is in line with previously published approaches. That said, I happen to think novelty is underrated - what is more valuable is to implement a method with coherence and rigor, which I think this study does.

Third, if I understood the study's framework correctly, taxonomic units that showed insufficient genomic differentiation but had marked phenotypic differentiation would be synonymized. Is this correct? Or does this scenario never happen? In either case, it might be help clarifying the logic.

Fourth, I was surprised to see the strict requirement for reciprocal monophyly. One could easily imagine a case of restricted geographical introgression leading to non-monophyly. Should these two hybridizing species be considered the same species based on this alone? More generally, can the study do more to justify why evidence for gene flow is used to argue for synonymization (e.g., no admixed individuals as in L445), given extensive evidence that gene flow can occur between "good" species? Perhaps I am misunderstanding these criteria, but I would appreciate hearing more about the logic. Importantly, as is always going to be the case, sampling is not quite even across species, and some species complexes do not appear to have much sampling close to parapatric species boundaries. Is this a valid concern?

Fifth, as an extension of my previous comment, many of these criteria are being used as proxies for reproductive isolation - e.g., divergence in mating calls perhaps might influence mate choice. What evidence is there - if any - for reproductive barriers between these species? Some of this is buried in the SI, but I think it would be useful to unearth this discussion and place it in context of the criteria in Figure

1. Such discussion would help this work be in greater dialogue with broader conversations in the speciation community.

MINOR NOTES

- L71: Does the study actually provide new conservation priorities? Given that the taxonomy has not formally changed, and given that they make some argument for evolutionary significant units (ESUs), it seems like conservation priorities might not have changed at all.
- L95 - 96: Not clear why primates need special care - perhaps briefly expand or strike?
- L133 - 136: Accounting for genome-level variation in IBD is a fun idea. I am curious if there are enough data given that this study is using RAD data. How many SNPs were found for a given species complex? Most RAD studies typically report ~20 - 30K, which would be only 20 or 30 windows. Not an issue - more curious. In any case, I recommend that the SI be amended to include the number of SNPs used for inference in each species complex.
- L147: I would recommend in this first major data paragraph mentioning the average number of genetic samples (and perhaps unique number of sampling points) per species.
- L148: Consider briefly mentioning what data were used to construct this phylogeny - the wording suggested whole-genome data, but it appears it was RAD?
- L150: Was delimitation done only pairwise? Imagine a case where A & B form a clade sister to C. If A & B were compared & synonymized, is AB now compared to C?
- L164 - 165: Given evidence that levels of within-taxa IBD can vary significantly across species, can the study clarify how a threshold level of IBD was identified?
- L175: Here, I think it could be useful to explain why *M. tavaratra* is the appropriate comparison point. (See again L483 - 485 - I am curious how it can be so clearly known that something is structure rather than speciation.)
- L177 - 179: Based on what the manuscript said earlier, I assume that these to-be synonymized species were originally defined based solely on mtDNA data? It might be helpful to remind the reader that the initial evidence for these species being distinct was modest.
- L187: "cathemeral" -> is this a commonly used term by non-primatologists? I had never heard it before! In any case, is there a reason why species with this irregular pattern of activity would be relevant?
- L217: Perhaps briefly mention where the mutation estimate came from, given how central that is to these claims. (This could be as simple as saying "We used a mutation rate inferred from pedigree data.")
- L226: It might be helpful here to explain that the authors chose not to use fossils because of their phylogenetic distance. (I assume?) Otherwise, fossils are generally superior to mutation rates ...
- L262 - 263: This is a fairly non-standard approach so it would be useful to provide a bit more context here until the readers can get the full details in the Methods. Also, what is the null expectation? The

authors say the correlation is non-significant, but relative to what evolutionary model?

- L282 - 284: I had trouble squaring this idea of a "neutral process of niche diversification" with the discussion in the previous section of niche divergence driving speciation. I am sure this two findings are not mutually exclusive, but it would be helpful to put them in context of each other.
- L284 - 5: Under a neutral process of niche divergence we would expect low overlap? Wouldn't that depend on the rate of neutral drift?
- L287 - 8: I can see the logic of neutral niche divergence linking to neutral theory of community assembly, but I think it would be helpful to more explicitly make these connections.
- L289 - 291: It seems to me that the morphological stasis & niche conservatism could also explain principles of community assembly, right? For example, in L299 - 302: perhaps communities are phylogenetically overdispersed because of high stasis between closely related species.
- L289 - 309: This entire section felt superfluous to me, only perhaps because it isn't so strongly grounded in data collected in this paper.
- L311 - 334: How do these different approaches to thinking about species units affect which geographic regions are prioritized for conservation?
- L337: I suggest listing the range of # of inds sampled per species for a given data type.
- L422: These species were the nominal species, right?
- L455: How was gdi of 0.25 chosen as a threshold?
- L602: I would be pretty wary of a diversification analysis on such a small phylogenetic scale - the difference in rates is likely to be a false positive, a notorious problem of these methods (see Rabosky & Goldberg 2015; 10.1093/sysbio/syu131).
- L622: I would imagine the correlation across these variables is high - is that an issue for interpretation?
- L1205: Consider putting a picture of a mouse lemur in this figure. If you work on something cute, might as well take advantage. :)
- L1205: Check spelling of Acoustic in figure.
- L1225: I was surprised by how young these species splits are - is this typical of these taxa?
- L1240: The triangles are hard to see - perhaps color could be used instead, given how shallow some of these node heights are?
- Suppl Info: I really enjoyed how the Suppl Info was formatted, with the extra methods immediately followed by the figures. It made it much more digestible. Thanks!
- SI L152: These species by species accounts are very helpful.
- SI L172: I am not a primatologist, so pardon my ignorance - but let's say a female from Species X has a non-overlapping oestrus with a female from Species Y. Does that necessarily mean that a male from Species Y won't try to mate with a female from Species X? In other words, I am curious how well the non-overlapping oestrus patterns reflect reproduction patterns.
- SI: On the Figures showing climatic niche (e.g., Fig 7) consider putting the occurrence points on the

ENMs so that readers can assess under- and over-fitting.

Reviewer #3 (Remarks to the Author):

In this manuscript the authors re-assess the species diversity of the genus *Microcebus* using a new and integrative framework incorporating multiple types of genomic data alongside morphological, bioclimatic, reproductive and acoustic data. Their stepwise approach and accompanying flowchart seem to be a well-thought-out and appropriate approach that will recognize good species while being conservative and avoiding over-inflation. They find that only 19 of the 26 candidate species are supported and recommend taxonomic deflation. Further, the manuscript yields important insights on phylogeny/paleobiogeography, divergence times and speciation rates, geographic structure of genomic and morphological variables, and the pace of evolution (supporting stabilizing selection and a neutral process of niche diversification).

I feel this is a critically important study that has been executed well and presented clearly. Many of us are tired of the arguments between splitters and lumpers, and this paper is a tangible and important step towards finding meaningful middle ground, and providing a model that could be profitably applied to other taxonomic groups. I was impressed and the comments I offer below are relatively minor. One larger point I thought could be addressed is the potential stability of the findings in the face of future evidence yet to be gathered. For example, what if currently an assessment of a pair of candidate species is based on 2 local populations for one species and 3 for another. Will future sampling raising this to 3 and 5 be likely to affect the results? Similarly, how about increased morphometric sampling from those new sites and new accumulation of other types of data. I know there are various aspects of the methods where randomization approaches are used, I think in order to address this issue. But, when a geographic range is relatively large and you're able to do those randomizations only among the individuals you've sampled from 2 or 3 localities, it wouldn't accomplish the same thing as truly randomly selecting individuals from across the entire geographic range. I wonder if this could be addressed or acknowledged.

Line-by-line comments:

60: I rather disagree with the term “behavioural” (see further comments below) and I think it's clearer if split here into reproductive and acoustic data.

187: lower case for mittermeieri.

251: Replace dryer with drier.

303: Is it not a bit contradictory to say “the trait diversity of lemur communities may have relatively

ancient origins” just after showing that the entire *Microcebus* genus is only 1.5 million years old?

315: I feel “mitigation” is not the right word choice here. Wouldn’t it be better to express that the lumping results in a species that is at the same level of endangerment, or a lower level of endangerment, than the least endangered of the two candidate species that were lumped? That, to me, is the only logical and consistent thing you can say.

323: The authors refer to “supplementary results and discussion”, yet I only see two supplementary files: 1) tables and 2) text and figures.

373: It is an odd categorization to call everything on this list “Behaviour”. Male testis size is an anatomical characteristic, and the others seem more like life history characteristics. Nothing on the list feels very behavioural. Fig S32D refers to them as “data on reproductive activity” and Table S15 refer to them as “reproductive activity data. In addition, the description in the main text is a bit misleading. For example, I would naturally think of “pregnancy” as some kind of length of time, or timing throughout the year, but what I see in the supplemental table is simply pregnant or not on a specific day of capture. The main text should be clearer.

486: Lower case for *mittermeieri*.

489: *lehilahytsara*.

540: NRMSE?

565: As with previous comment, I feel “behaviour” is not the right heading to use.

567: oestrus?

704: I disagree with the tone taken here in the “Conservation reassessment” section and the way it’s explained. First, a published article can’t officially “update” an IUCN endangerment status – only an IUCN-published assessment can do that. The tone here implies that the authors are doing this. I would suggest a re-wording using the wording “new suggested conservation statuses”, or “likely new conservation status”. Secondly, looking at Table S10, there is quite a mix of things going on. For *M. lehilahytsara*, Dolch et al already did the reassessment officially; this should be mentioned somewhere as a counterpoint to the others, which have not been reassessed. There is also quite a wide range of cases. Sometimes (e.g. *murinus+ganzhorni+manitatra*) it is fairly straightforward: a widespread LC species that now has two relatively tiny chunks of habitat added will certainly come out as a LC species. However, for *simmonsii+boraha*, you have an EN species plus a DD species, respectively. Table S10 tells us these authors have decided to make it a VU species, with no explanation. Given the (not perfect but) quantitative and repeatable IUCN endangerment criteria, this comes across as relatively sloppy/arbitrary. I would say there either needs to be justification given for things like this, or the authors back off on the proclamations.

1203: The wording “largely overlapping” in the figure feels a bit arbitrary/qualitative compared to the other, more quantitative criteria. Could we be more objective/quantitative?

1203: Correct “Acoustic” in the figure

*****END*****

Author Rebuttal, first revision:Reviewers' comments:

Reviewer #1 (Remarks to the Author):

In this paper van Elst assembled and analyzed large sets of genomic, bioclimatic, morphometric, behavioural and acoustic data to characterize the diversity in the mouse lemurs (*Microcebus*), in particular to fix the problem of taxonomic inflation in the group. I think a main strength of the paper is the amount and diversity of data, and the use of a great many software tools.

Reply: We thank reviewer #1 for the positive feedback and the detailed and constructive comments on our manuscript.

I have a few minor comments. I should note that I am not knowledgeable enough to comment on many aspects of the paper, and my comments are limited to the analysis of genomic data.

2

a. The authors identified two well-recognised cases of within-species population structure (instead of distinct species) in the group: (i) *M. lehilahytsara* and *M. Mittermeieri*, which have been synonymized, and (ii) widely distributed populations of *M. tavaratra*. These are used to generate null distributions, with the rationale that a pair of candidate species/hypothesised species/populations showing similar population structures, isolation-by-distance patterns, etc. to those two cases should be lumped into one species. I suggest that the authors state this analysis framework explicitly in the Introduction, so that biologists working on other species may be able to judge whether the same strategy is applicable to their own data. Given that the mouse lemurs are very close, this framework may be justifiable. However, there may be huge variations in population size, in geographic distributions and overlaps etc., between pairs of good species even within the same genus, so that cutoffs identified for one good species may not apply to another. Reply: First, we appreciate the reviewer's comment highlighting the potential for emphasizing a key aspect of our analytical framework earlier in the manuscript. To address this suggestion, we included a mention of this novelty in the Abstract and Introduction and a brief description in the first paragraph of the Results and discussion section, specifically within the delineation framework subsection (lines 62-66, 115-118, 151-154). Additionally, we expanded upon its explanation in the Methods (lines 501-508). Although the reviewer suggested a short explanation in the introduction, we believe its current placement provides better context.

Second, the reviewer raises valid points regarding the generalizability of the cutoffs identified with *M. tavaratra* or *M. mittermeieri*/*M. lehilahytsara*. These depend on species-specific factors like life history traits (e.g., dispersal and population size) or genomic architecture, which we now mention in the Methods as well (lines 508-511). In addition to the reviewer's remark that this limitation is less concerning for mouse lemurs due to their relatively recent diversification and shared life history traits, we would like to point out that we were quite cautious in our use of null distributions inferred from these "reference" systems (see also reply to reviewer #2's comment on line 175).

First, we used two species to generate these null distributions (i.e., *M. lehilahytsara* and *M. tavaratra*) and not just a single one. Second, we only rejected an intraspecific clinal variation model for genomic data, for instance, if the 0.05 quantiles of both NRMSE distributions of a candidate pair were above the 0.95 quantiles of the reference null NRMSE distributions. Conversely, if the 0.95 quantiles of a NRMSE distribution of a candidate pair was below the 0.95 quantiles of the null NRMSE distributions, we considered genetic distances to be congruent with a model of intraspecific structure. Cases that were neither rejecting nor congruent with the intraspecific model were considered inconclusive (i.e., inside a "gray zone") and required additional evidence. These two precautions were specifically chosen to buffer and account for heterogeneity in population size, structure, etc. among species of a genus, increasing generalizability also to other systems. Finally, we would like to highlight that one of the aims of this study was to provide a framework to consistently delimit species of cryptic genera (in our case *Microcebus*), where "good" and "bad" species are often not well-defined beforehand, e.g., due to a lack of phenotypic differentiation, and little other evidence than gene flow and/or genetic distances is available for delimitation. In these cases, we believe null distributions inferred from "benchmark" species provide a useful objective heuristic for systematic delimitation across the entire genus. In fact, we believe the use of these "benchmarks" is a strength of our approach, as it allows us to account for variation between genera in genetic distances etc., which is not the case when using hard cut-offs, e.g., as in Jackson et al.'s rule of thumb for the gd^{\dagger} .

b. The filters used to compile the genomic data are not always clearly described. Some of them seem unnecessarily complicated and ritualistic. I have a few examples below.

Reply: We agree with the reviewer that some filters used to compile genomic data were not clearly presented. We thank the reviewer for the effort made to systematically report the unclear sentences, which was very helpful to address these issues. We detail below the clarifications that were brought to each of them.

My other comments are sorted by page.

p.111.389- “After removing indels, only sites with a minimum global depth of 10, with a maximum global depth of the sum of the 0.995 quantiles of per-individual sequencing depth distributions, and represented in at least three individuals, were retained. In addition, sites with a minimum per-individual depth lower than two or a maximum per-individual depth larger than the maximum 0.995 quantile of per-individual sequencing depth distributions”

Is “global depth” the sum of read depths over all samples, and if so, over how many samples? The read depth seems low, although we do not have a good understanding of how the downstream analysis is affected by low coverage.

Reply: Yes, “global depth” refers to the sum of read depths over all samples for which genotypes were called, which are 214 in total (mentioned in line 449; Table S17 also provides information on the number of samples and sites used in each analysis). Indeed, given the number of samples, the threshold is low. We used relaxed depth filters to keep private loci for taxa represented by very few samples and with potentially limited sequencing data, which is unavoidable for such a comprehensive study aiming to cover all species of the genus. For instance, if a site was present at low coverage and only in the clade composed of *M. gerpi*, *M. jollyae* and *M. marohita* (which is represented by few samples), we would have lost this information with a high global minimum depth threshold.

It is also true that allowing for low coverages might affect downstream analyses, even though it is unclear to what extent. Nevertheless, we are confident that our results are robust for several reasons. First, among all samples available for this project, we only used those with mean forward read coverage across RAD sites larger than 5 for genotype calling (compare Tables S11 and S16), which should alleviate effects of low coverage at some sites given the large number of SNPs used in total. Second, we obtained congruent results for phylogenetic inference when using SNP sets with largely variable number of retained sites (allowing for differing amounts of missing data, ranging from 5% to 95%), when thinning these data sets every 10k bp, and when using different inference algorithms. In addition, we also recovered a congruent topology when running phylogenetic inference on the 6,000 best extracted full locus sequences used for BPP and the estimation of *gdi* (which we did to ensure that the underlying data for these analyses supported the same topology as our comprehensive SNP-based inference). Finally, results of genetic clustering analyses, which were carried out on genotype likelihoods, alleviating some effects of low sequencing depth (e.g., Lozier, 2014; Pedersen et al., 2018; Warmuth and Ellegren, 2019), were also largely congruent with delimitation results obtained on the basis of SNP calls.

“the maximum 0.995 quantile of per-individual sequencing depth distributions” sounds very complicated.

Reply: We agree with the reviewer that the expression sounds rather complicated. This filter was applied as a way to exclude sites with exceedingly high sequencing depth (i.e., outliers) compared to the average depths found among considered individuals. We tried to simplify the presentation of these filters as much

as possible (lines 451-455, 465-468), but felt only able to do so to a limited degree given their technical nature. We would be grateful for any suggestions that would further simplify these expressions without losing information.

p.13 l.443 “The absence of reciprocal monophyly was considered sufficient evidence for synonymization.”
 p.13 l.447 “The presence of individuals with admixed ancestry was considered sufficient evidence for synonymization.”

Perhaps some simple justifications or comments are called for here. Some good species do not show reciprocal monophyly, and some good species are involved in gene flow.

Reply: We thank the reviewer for pointing out the need for clearer justifications in these two sentences. Indeed, reciprocal monophyly and admixed ancestry may not be sufficient evidence for synonymization in general and across all taxa (e.g., incomplete lineage sorting or introgression do take place between well-identified species). The criteria of reciprocal monophyly and absence of admixed ancestry are here used mainly in relation to previous mouse lemur systematics work, where most species descriptions were based on molecular diagnosis in few mitochondrial (mtDNA) and/or nuclear loci alone (now detailed in the species-by-species accounts in the Supplementary results and discussion). It is in this context that we think that these criteria (inferred from RAD sequencing) provide solid evidence to overrule the earlier splitting based on limited genetic evidence. We thus challenge previous splits between neighbouring mouse lemur populations which were likely affected by sparse sampling and the smaller effective size of mtDNA.

Another important result of nearly two decades of genetic research in Madagascar is that there is no evidence of ongoing gene flow between pairs of sympatric mouse lemurs. The only two studies that claimed to have identified cases of ongoing hybridization among sympatric species (Hapke et al.² for *M. murinus* and *M. griseorufus* and Teixeira et al.³ for *M. murinus* and *M. ravelobensis*) have been refuted in recent years by Poelstra et al.⁴ and Teixeira et al.⁵, respectively. This suggests that signatures of hybridization or incomplete lineage sorting between allopatric mouse lemur taxa can be explained by the fact that they are part of one metapopulation. We believe that for the sake of scientific rigor we should consider such metapopulations species until otherwise proven.

One could argue that what we identify are rare hybridization events (or incomplete lineage sorting) that should not lead us to lump populations into one species. We believe that this can most likely be excluded in our specific study case. In particular, only two cases of non-reciprocal monophyly have been identified in our study:

- In *M. boraha* vs. *M. simmonsii*, the non-reciprocal monophyly shows 100% bootstrap support in all analyses and we do not find any evidence for admixture (Fig. S11b). Such strong support would not generally be expected if this finding resulted from hybridization, as hybrid genomes typically exhibit admixed sequences from both parental species.
- Regarding *M. murinus* vs. *M. manitatra* vs. *M. ganzhorni*, if the non-reciprocal monophyly was the result of hybridization between two “good species”, we would expect distinct clades representing the parental species and some “hybrid” individuals clustering with either clade. However, the observed “nested phylogenetic gradient” (Fig. S14) lacks such clear distinctions, making hybridization an unlikely explanation.

Finally, in cryptic lineages, by definition, little other criteria are available aside from genomics to sort ‘good species’ from ‘bad candidates’, which makes reciprocal monophyly and a lack of gene flow useful, conservative (and potentially necessary) heuristics for their delimitation (similar to the use of benchmark systems mentioned in a previous comment).

To address the reviewer's point, we clarified these justifications in the Methods (lines 518-543). We would like to also refer the reviewer to the answer to a relatively similar and complementary comment given by reviewer #2 (top of page 10 of this document).

p.13 l.455 "we used the *gdi* only to synonymize (if below 0.25) but not to confirm candidate species." Jackson et al.'s rule of thumb is that "gdi values less than 0.2 suggest that a single species exists". Here the authors used 0.25. Is this because the objective of the paper is to "synonymize" candidate species?

Reply: The reviewer is right that, according to Jackson et al.'s rule of thumb¹, "*gdi* values less than 0.2 suggest that a single species exists". Our deviation from this rule (i.e., using 0.25 instead of 0.2) was not properly justified. We now decided to follow their suggestion and use a threshold value of 0.2 in the revised version of our manuscript because the scope of this work is not to re-evaluate the *gdi*'s rule of thumb. This modification does not change the conclusions of our species delimitation analyses. However, we think that it would be important to clarify our original rationale. We therefore explain, in the Supplementary material why the lower rule of thumb threshold (0.2) may potentially be too low and inappropriate for mammals (Supplementary methods: lines 1054-1068). Furthermore, we added a brief mention of our concern in the main text (lines 550-554), pointing to the Supplementary material.

p.13 l.462 "We divided genomic data into contiguous windows containing a fixed number of SNPs ..."

It should be better to use a fixed window size (10kb, 100kb, etc.). Choosing the window size to have a fixed number of SNPs complicates the statistical test, although this is a problem with Li & Ralph (2019).

Reply: We appreciate the reviewer's comment on the window size definition. While we adopted the general concept from Li & Ralph⁶, who used full-genome data, our approach differs due to the nature of our data. That is, RAD-seq markers are randomly and sparsely distributed across the genome, making it challenging to find long, contiguous stretches of DNA. Therefore, we chose to define window size based on a set number of SNPs instead of a fixed window size in kilobases. It is important to note that we also incorporated Li & Ralph's method for selecting the optimal window size by assessing the signal-to-noise ratio across different window sizes. This step helps to ensure the robustness of the signal within each window and partially addresses concerns about potential SNP dependence within the same window. Following the reviewer's suggestion, we have updated the main text to provide a clear explanation for using a SNP-based window size ("We used SNP number and not window length in bp to divide genomic data because of the scattered nature of RADseq data" in lines 563-564).

p.14 l.470 "the average number of nucleotide differences between two individuals (π)" Should this be "the average proportion of nucleotide differences"?

Reply: Recognizing that our previous explanation might not have been fully accurate, we've revised it to use the precise terminology: "the average number of nucleotide differences per site between two individuals (π)" (lines 571-572). This aligns with the formal definition established in the initial study describing this nucleotide diversity statistic (Nei and Li⁷; pg. 5273 line 8).

p.15 l.519 "by resampling 90% of the data 100 times. Confidence intervals of morphometric overlap..."

Is this some kind of jackknifing. If so, how do you apply a correction (for the smaller jackknife samples than the original data)?

Reply: For each comparison of two mouse lemurs species, we first calculated the overlap of hypervolumes based on morphometric parameters using the full dataset. Then, we used a jackknifing approach to generate

the 95% confidence interval (CI) by re-calculating these overlaps 100 times, each time based on randomly selected 90% of the data (i.e., *Microcebus* individuals). Because overlaps are measured as the proportion of the total hypervolume space that is overlapping (see Junker et al.⁸: “The aggregation method ‘product’ measures the overlap port(A, B), which is A in B or the portion of the trait space of B that is covered by A, in the n-dimensional hypervolume (sensu Hutchinson 1957)”), these relative overlap values should not be downward biased due to a reduced sample size, which is also indicated by CI values that are both higher and lower than the point estimate obtained using 100% of the data (Figure 3). We clarified this point now in the Methods (lines 644-645, 666-667). In case we misunderstood the reviewer’s request, we kindly ask for clarification.

Reviewer #2 (Remarks to the Author):

This manuscript combines newly-collected genetic data with phenotypic data to provide an integrative taxonomy of the mouse lemurs of Madagascar. The study then uses this new taxonomy to understand the diversification of the mouse lemurs, including their phenotypic evolution, and to propose revised conservation aims.

Overall, this paper was exceptionally well-written. The ideas were well-organized, the flow was coherent throughout, and the language was thoughtful and well-reasoned. Most approaches seem well-considered and properly implemented. In particular, I enjoyed how the study presents clade-by-clade analyses in the Suppl Info & then highlights a few exemplar clades in the main text.

Reply: We thank reviewer #2 for the evaluation of our manuscript and appreciate the detailed, constructive feedback. Your comments have undeniably helped us enhance the quality of our work. It is especially rewarding to receive such positive recognition for the effort invested in assembling and extensively analysing this comprehensive data set.

I have a few major comments for consideration.

First, this study felt like two separate papers to me. The section on cryptic species & species delimitation felt a bit disconnected from the section on evolutionary dynamics. The study could simply be split in two, but I suspect the authors combined these two sections for a reason. So, perhaps instead more could be done to discuss how the revised taxonomy influences our interpretation of evolutionary dynamics. (The study has started to do this by exploring how the revised taxonomy affects conservation priorities.)

Reply: We appreciate the reviewer’s comment and agree that other readers may also feel the same. We have thus included a short paragraph in the Results and discussion (lines 235-251) to better integrate the cryptic species delimitation with the evolutionary dynamics section. The reviewer is also right that the section on evolutionary dynamics was included in this manuscript because the goal of this first extensive phylogenomic work on mouse lemurs, besides providing an accurate taxonomic framework, was to investigate the origins, processes and temporal scale of their genomic and phenotypic diversification. In other words, while the sections on cryptic species delimitation investigates observed *patterns*, the following sections aim to unravel the *processes* underlying their cryptic radiation. After all, to infer these processes correctly, it is essential to first have an accurate understanding of the partitioning of biodiversity (e.g., to not violate model assumptions). The link between defining the accurate taxonomy using patterns of

evolution and assessing the processes underlying the evolutionary trajectories is also emphasised throughout the introduction in lines 86-88, 106-109, 120-122, and 126-127.

Second, the study mentions several times that their taxonomic approach is "novel". For example, in L190 - 192, they suggest this framework could also be applied to other controversial delimitations. I am not sure I understand what aspect of it is novel, as most of what they implement is in line with previously published approaches. That said, I happen to think novelty is underrated - what is more valuable is to implement a method with coherence and rigor, which I think this study does.

Reply: We appreciate the reviewer's feedback and agree that the overall framework might not be considered as novel (e.g., we mention in the introduction that our framework generally follows Padiál et al.⁹; line 113) and that its true value lies in the systematic implementation of a comprehensive suite of these analyses across all species comparisons. Still, we believe there are the following unique aspects of our framework:

- A quantitative assessment of the covariation between geographic and genomic/phenotypic structure among candidate species pairs: Assessing the effect of geography on genomic and morphological diversity is not a new idea (e.g., see Mayr¹⁰ for a conceptual framing, and more recent technical work by Hausdorf and Hennig¹¹ and Bradburd et al.¹²). Similarly, the integration of covariation between geography and genetics into taxonomy has notably been proposed in several studies^{11,13}. However, we are not aware of any published work that quantifies the contribution of geography across "independent" genomic windows to inform taxonomic decisions, while accounting for genome-wide variation of the genealogical process. We additionally adapted this IBD approach to morphological data, for which we consider that each trait might have undergone complex evolutionary and developmental processes, while being potentially correlated with other traits. Therefore, we do not compute the average IBD using all morphological variables but resample combinations of morphological traits (to account for possible trait correlations), to quantify the strength of the geographic signal.
- The development of simple approaches to estimate taxon-specific threshold values for integrative and spatially informed decision-making in taxonomy: We leverage the power of the NRMSE statistic to assess whether IBD signals (genomic and morphometric) are consistent with models of intra- vs. interspecific divergence. To do so, we systematically compared the distribution of our resampled metrics to the null distribution obtained across populations of the genetically structured *M. tavaratra* (see reply to comment on line 175 for why we think this is appropriate). From this, we propose simple rejection procedures to classify the observed patterns either as "intraspecific", "intermediate" (the gray zone of speciation) or "interspecific". Such an approach mitigates biases arising from uneven or scattered sampling (which are problematic, for instance, when using molecular diagnostic sites or phylogenetic relationships only for species delimitation) and has, to our knowledge, never been formalized before.
- Methodological developments for the evolutionary analysis of morphology and climatic niche: We consider our approach to model morphological and niche evolution as novel because, unlike other studies, we did not limit ourselves to fitting evolutionary models to the data, but rather validated the fitted model using simulations and observed patterns of morphological overlap. The simulation-based approach has been previously used¹⁴⁻¹⁷ but not on morphometric overlap as we did here. Admittedly, this is not a major reason for 'novelty', but still represents a new perspective on this type of data, further showing that it can perform better than traditional model selection approaches based on statistics such as AICc.

Following the reviewer's suggestion, we now refrain from claiming that we propose a novel taxonomic framework and only mention novelty when referring to the specific aspects mentioned above (e.g., lines 115-118, 558), while putting more emphasis on the systematic application of an integrative framework across the entire genus *Microcebus* (e.g., lines 66,118,489).

Third, if I understood the study's framework correctly, taxonomic units that showed insufficient genomic differentiation but had marked phenotypic differentiation would be synonymized. Is this correct? Or does this scenario never happen? In either case, it might be help clarifying the logic.

Reply: We thank the reviewer for pointing out that the logic of our approach was not clear enough. Reviewer #1 commented on a related point (comment on lines 443 and 447; page 5 of this document), and the next comment is also related. We therefore answer briefly on the above questioned aspect here, and further comment on the general framework and on the modifications clarifying the text in the replies to the above-mentioned comments.

To answer the reviewer's specific question, it is important to clarify how insufficient genomic differentiation is defined. We synonymize candidate species on the basis of genetics only (i.e., despite other lines of evidence such as phenotypic differentiation) if we find evidence for non-monophyly, recent/ongoing gene flow, particularly low divergence (i.e., $gdi < 0.2$) or a clear pattern of isolation-by-distance (IBD). We consider such evidence of genetic connectivity sufficient to challenge the original species classifications as most *Microcebus* species were described on the basis of only a few mitochondrial and/or nuclear sequences (which we now detail for each species in the supplementary material). It is true that in such a case of "insufficient genomic differentiation", other lines of evidence such as marked phenotypic differentiation would not be able to overrule the decision according to our framework. Another case of "insufficient genomic differentiation", however, could refer to a situation where the four criteria above were not met (i.e., candidates are reciprocally monophyletic, do not show evidence of gene flow, have $gdi \geq 0.2$ and do not exhibit clear IBD), indicating differentiation/lack of genetic connectivity, but still genetic distances were comparably low. An example of this would be the differentiation between *M. ravelobensis* and *M. bongolavensis* in our study. In that case, marked phenotypic differences would suffice to conclude that the two candidates should be treated as separate species, because the genetic differentiation, albeit low, would be supported by other lines of evidence. As we did not find such clear signals of phenotypic differentiation, we conservatively synonymized the candidates as belonging to the same species until new data may challenge this interpretation. We try to make the logic of our use of additional lines of evidence clear in lines 600-609.

Fourth, I was surprised to see the strict requirement for reciprocal monophyly. One could easily imagine a case of restricted geographical introgression leading to non-monophyly. Should these two hybridizing species be considered the same species based on this alone? More generally, can the study do more to justify why evidence for gene flow is used to argue for synonymization (e.g., no admixed individuals as in L445), given extensive evidence that gene flow can occur between "good" species? Perhaps I am misunderstanding these criteria, but I would appreciate hearing more about the logic. Importantly, as is always going to be the case, sampling is not quite even across species, and some species complexes do not appear to have much sampling close to parapatric species boundaries. Is this a valid concern?

Reply: We thank the reviewer for pointing out the need for clarification of these potentially controversial criteria. As reviewer #1 made a largely similar comment (comment on lines 443 and 447), we would like to point the reviewer to the comprehensive answer given there (page 5 of this document).

We also acknowledge the reviewer's concerns about uneven sampling across species. We would like to emphasize that our work actively addresses this issue. Our geography-focused species delimitation method, based on isolation-by-distance (IBD) across genomic windows, mitigates the impact of sampling gaps or geographically restricted sampling. It assesses whether apparent phylogenetic monophyly reflects true species differentiation or results from restricted sampling within a single, geographically clinal species. In other words, this method helps to distinguish legitimate species boundaries from false positives arising from uneven sampling. We explicitly emphasized the critical role of our IBD framework in mitigating biases arising from uneven and scattered sampling. This emphasis was placed at the outset of the Results and discussion sections (lines 141-145), while a detailed explanation of the framework is provided in the Methods (lines 500-504, 555-599). Finally, following a comment of reviewer #3, we now explicitly state in the Supplementary results and discussion (see species-by-species accounts) for each candidate group how robust our findings are with respect to sampling, i.e., whether additional sampling might challenge the taxonomic conclusion. More specifically, we are confident that our conclusions are robust for at least five of the nine considered candidate groups. For the remaining four groups, additional sampling (i.e., at species boundaries) is recommended to resolve open questions and ultimately refute/support our taxonomic hypotheses, but we do not believe that it would bring major changes.

Fifth, as an extension of my previous comment, many of these criteria are being used as proxies for reproductive isolation - e.g., divergence in mating calls perhaps might influence mate choice. What evidence is there - if any - for reproductive barriers between these species? Some of this is buried in the SI, but I think it would be useful to unearth this discussion and place it in context of the criteria in Figure 1. Such discussion would help this work be in greater dialogue with broader conversations in the speciation community.

Reply: We thank the reviewer for this helpful comment. Several criteria are indeed being used both as proxies for reproductive isolation (RI) and as measures of trait divergence to judge the distinctiveness of a specific taxon from its closest relative. This includes criteria that can likely cause RI (e.g., differentiation in mating calls or reproductive activity) as well as those that might simply be a result of RI without necessarily precluding it (e.g., differentiation in morphometry). Notably, all sister candidate species considered in this study occur allopatrically, impeding the study and confirmation of direct reproductive barriers. This is why our framework is specifically designed for such cases and predominantly focuses on (spatial) genomic analyses (e.g., the genome-wide IBD approach presented here) that are powerful tools to detect signatures of RI even if taxa are no longer in direct contact. Importantly, our framework is designed in a way that differentiation in a line of evidence other than genomics alone can not lead to the confirmation of species status. These proxies (or measures of trait divergence) are only used as additional evidence in case we already find clear genomic differentiation to differentiate population structure from speciation.

It is worthy of note, though, that sympatric mouse lemur species (*M. murinus* - *M. griseorufus*/*M. berthae*/*M. myoxinus*/*M. ravelobensis*; *M. lehilahytsara* - *M. macarthurii*/*M. jonahi*/*M. simmonsii*; *M. arnholdi* - *M. tavaratra*) are more distantly related and are generally assumed to be reproductively isolated as no hybrids have ever been confirmed even though several genetic studies have investigated these systems. Notably, two studies have reported cases of ongoing hybridization among sympatric species (Hapke et al.² for *M. murinus* and *M. griseorufus*, and Teixeira et al.³ for *M. murinus* and *M. ravelobensis*) but their results were systematically refuted in recent years (Poelstra et al.⁴ and Teixeira et al.⁵, respectively), suggesting that strong yet unknown reproductive isolation mechanisms may be effective among sympatric species.

Following the reviewer's suggestion, we made clarify this now, highlighting the importance of allopatry for our study system across the manuscript (e.g., lines 163, 494-497, 600) and that our criteria are used as proxies for RI and/or measures of trait divergence (e.g., lines 114-115, 497-500, 605-609, 1307-1308).

MINOR NOTES

- L71: Does the study actually provide new conservation priorities? Given that the taxonomy has not formally changed, and given that they make some argument for evolutionary significant units (ESUs), it seems like conservation priorities might not have changed at all.

Reply: We thank the reviewer for raising this point. Following reviewer #3's comment on line 704 (see our reply to that comment for more details), we now perform a proper formal reassessment of conservation status according to IUCN guidelines for all species in the updated taxonomy. This shows that the increased sampling presented in our study and the taxonomic revision lead to a different recommended conservation status for several species (and not just for the synonymised ones; see Fig. 2d in main manuscript), which has direct implications for conservation prioritisation. For instance, *M. macarthurii*, *M. rufus* and *M. tanosi*, which represent a single ESU, respectively, no longer fall into a Threatened category due to increased sampling. Another example is given by *M. manitatra*, *M. ganzhorni* and southeastern *M. murinus* populations, which constitute only one single lineage (and ESU) according to our taxonomic investigation, indicating that these three groups should no longer be treated as independent entities for conservation purposes. We mention this now also in the discussion (lines 379-382).

- L95 - 96: Not clear why primates need special care - perhaps briefly expand or strike?

Reply: The reviewer is correct in pointing out this unjustified statement, which we deleted.

- L133 - 136: Accounting for genome-level variation in IBD is a fun idea. I am curious if there are enough data given that this study is using RAD data. How many SNPs were found for a given species complex? Most RAD studies typically report ~20 - 30K, which would be only 20 or 30 windows. Not an issue - more curious. In any case, I recommend that the SI be amended to include the number of SNPs used for inference in each species complex.

Reply: We thank the reviewer for their positive comment on this methodology. For this analysis, we used the SNP set called and filtered for all 214 individuals (across all candidate species) with a maximum of 5% missing data allowed per site, which was subsequently subsampled to include the relevant individuals for each given species complex. The initial genotype set comprised a total of 119,079 SNPs. Since the `vcf_windower` function of the R package 'lostruct' used within our framework trims SNPs at chromosome edges (for window size 1000, 15,079 SNPs were removed), the analysis was finally performed on 104,000 SNPs resulting in a relatively large number of windows ($n = 104$). Accordingly, the number of underlying SNPs is the same for each candidate group and only the amount of missing data varies across groups. Following the reviewer's suggestion, we added this information (i.e., number of sites plus percent missing data for each candidate group) now to Table S17, which already detailed dataset characteristics for the other analyses. Given the high number of SNPs across 104 windows and the relatively low amount of missing data (3.8% on average across candidate groups), we are confident that our analyses are statistically sound.

- L147: I would recommend in this first major data paragraph mentioning the average number of genetic samples (and perhaps unique number of sampling points) per species.

Reply: We thank the reviewer for suggesting this addition to the first major data paragraph. Accordingly, we added the total number of genotyped samples for the phylogeny and the pairwise delimitations, as well as the median number per candidate species (lines 160-164).

- L148: Consider briefly mentioning what data were used to construct this phylogeny - the wording suggested whole-genome data, but it appears it was RAD?

Reply: To address the issue raised by the reviewer, we now explicitly state that the underlying data are RAD-genotyped samples (line 161).

- L150: Was delimitation done only pairwise? Imagine a case where A & B form a clade sister to C. If A & B were compared & synonymized, is AB now compared to C?

Reply: We thank the reviewer for highlighting the need for clarification on this aspect of our framework. Among the nine candidate subsets that we identified, we carried out delimitation tests between all pairs of candidate species. The reviewer is right that, ideally, the procedure should be conducted iteratively in case of a synonymization (i.e., if A & B were synonymised, they should then be compared to C as mentioned above). For our *Microcebus* data set, this applies to the subsets *M. gerpi*/*M. jollyae*/*M. marohita* and *M. bongolavensis*/*M. danfossi*/*M. ravelobensis*, which are the only ones comprised of three candidates with at least one synonymisation (with the exception of the *M. ganzhorni*/*M. manitatra*/*M. murinus* subset, in which all candidates were synonymized anyway). We did not formally carry out all iterative delimitation tests for these cases (i.e., *M. gerpi* vs. *M. jollyae* + *M. marohita*, and *M. danfossi* vs. *M. bongolavensis* + *M. ravelobensis*) for the following reasons:

- *M. gerpi* vs. *M. jollyae* + *M. marohita*: Following our framework (Fig. 1), we can see that *M. gerpi* is reciprocally monophyletic (Fig. S12b), forms distinct clusters (Fig. S12c) and exhibits very high *gdi* (Fig. S12d; although *gdi* are generally high in this candidate group) compared to *M. jollyae* and *M. marohita*. In addition, NRMSE values of the genomic isolation-by-distance (IBD) analysis are high when comparing *M. gerpi* to *M. jollyae*, indicating that between-candidate genetic distances are much larger than within even at similar geographic distances (Fig. S15e), which we take as strong evidence for splitting. Finally, we also detect morphometric differentiation between the two candidates. Data for *M. marohita* were too limited to conduct these tests. The results of the IBD analysis and morphometric overlap estimates might change when performed against a new candidate that comprises individuals of both *M. jollyae* and *M. marohita*. However, given the clear NRMSE pattern in comparison to *M. jollyae* (Fig. S12e, top), the very limited data available for *M. marohita* and the fact that plotted geographic against genetic distances of *M. gerpi* against *M. jollyae* and against *M. marohita* form a single point cloud (Fig. S12e, bottom; blue and orange crosses), we are certain that such analyses would not change the general conclusion of treating *M. gerpi* as a separate species. Data on niches and reproductive activity were mostly too limited to be conclusive for delimitation in this candidate group.
- *M. danfossi* vs. *M. bongolavensis* + *M. ravelobensis*: Similar to the case of *M. gerpi*, we can see that *M. danfossi* is reciprocally monophyletic (Fig. S15b), forms distinct clusters (Fig. S15c) and exhibits very high *gdi* (Fig. S15d) compared to *M. bongolavensis* and *M. ravelobensis*. Moreover, it has particularly high NRMSE values when being compared to each of the two other candidates, indicating that between-candidate genetic distances are much larger than within even at similar geographic distances (Fig. S15e), which we take as strong evidence for splitting. This is further supported by an earlier onset of reproductive activity in *M. danfossi* (Fig. S15h; morphometric and

climatic data are inconclusive in this case). Among these analyses, only the results of the IBD analysis based on the NRMSE would change when performed against a new candidate that comprises individuals of both *M. bongolavensis* and *M. ravelobensis*. However, we are certain that an IBD analysis on such a candidate (i.e., *M. bongolavensis* + *M. ravelobensis*) will yield the same general conclusion (i.e., treating *M. danfossi* as a separate species) given that the NRMSE pattern obtained when comparing *M. danfossi* against *M. bongolavensis* or *M. ravelobensis* are largely congruent (Fig. S15e, top) and that the plot of geographic against genetic distances shows that comparisons of *M. danfossi* against *M. bongolavensis* and against *M. ravelobensis* form a single point cloud (Fig. S15e, bottom; blue and orange crosses).

We made these justifications clear now in the species-by-species accounts in the Supplementary results and discussion. If the reviewer nevertheless thinks that all iterative tests should be formally conducted for the above-mentioned cases, we are happy to include them.

- L164 - 165: Given evidence that levels of within-taxa IBD can vary significantly across species, can the study clarify how a threshold level of IBD was identified?

Reply: The reviewer is right in assuming that within-taxa IBD varies across species, as shown in the IBD plots reported in the Supplementary results and discussion. To overcome this issue, the genome-wide IBD test presented here does not use absolute distances of within- and between-taxa comparisons, but the normalised root mean squared error (NRMSE), a statistical measure that specifically accounts for the variability in within-taxon IBD. This approach and reasoning are explained in detail in the Methods (lines 574-581) and Supplementary methods (lines 1081-1132). We added an additional sentence to the latter to make this point clearer ("The NRMSE normalises genetic distances between and within candidates by the range of observed distances between candidates, thus facilitating comparisons among species complexes with different within-taxon IBD scales." in lines 1083-1086). Finally, to identify a decision threshold associated with this metric that is relevant for the genus *Microcebus*, we leveraged the genetic diversity of the two widely distributed and well-characterized *Microcebus* species *M. lehilahytsara* (including *M. mittermeieri*) and *M. tavaratra*, which is described in detail in the Methods as well (lines 501-508, 582-599). Following the reviewer's suggestions, we also briefly expand on that now in the Results and discussion, albeit not at the exact position the reviewer's question referred to but as part of the first paragraph (lines 151-154), as we believe it is better contextualized there.

- L175: Here, I think it could be useful to explain why *M. tavaratra* is the appropriate comparison point. (See again L483 - 485 - I am curious how it can be so clearly known that something is structure rather than speciation.)

Reply: We thank the reviewer for raising this point. We think that *M. tavaratra* is the appropriate comparison point as it is one of the few species whose distribution is wide (Fig. 1a here, reproduced from Fig. S31a of the Supplementary material) and covers both fragmented and continuous populations, providing thus an empirical case study of strong population structure. In addition, *M. tavaratra* is the species for which we have the most extensive population genomic dataset, thus providing great spatial and genomic resolution (Fig. S32a). This dense population genomic sampling of the species provides the following arguments in support of our claim:

- In Salmona¹⁸, Sgarlata et al.¹⁹ and Aleixo-Pais et al.²⁰, we showed that all sampled individuals belong to *M. tavaratra* based on mtDNA and that no further significant genetic substructuring can be identified within the clade (Fig. 1b here, reproduced from Sgarlata et al.¹⁹).

- Phylogenomic analyses of the present study also show shallow phylogenetic relationships between *M. tavaratra* sampling sites (Fig. 1c here, reproduced from Supplementary Fig. S1) and no clear clustering, inconsistent with expectations of speciation. For comparison, we show the comparably clear phylogenetic relationship of three recognised species *M. mamiratra*, *M. sambiranensis* and *M. margotmarshae* in Fig 1c here.
- Genetic distances plotted against geographic distances (i.e., analysis of isolation-by-distance) are consistent with recent or ongoing gene flow. More specifically, as shown in Fig. 1d here (reproduced from Supplementary Fig. S34), genetic distances tend to increase mostly linearly with the log of geographic distances and exhibit a relatively small range of values (i.e., 0.012-0.014). Two distinct (reproductively isolated) species are not expected to show such a linear relationship between genetic and log of geographic distances among samples, unless speciation has occurred very recently and/or manifests only in few genetic loci. Still, there is a small gap between the point clouds (i.e., genetic distances) when comparing individuals of continuous *M. tavaratra* populations (blue dots/"within" in Fig. 1d here) versus ones from fragmented populations (red dots/"between" in Fig. 1d here), allowing us to use this pattern as a benchmark to identify intraspecific structure (assuming from the other evidence that *M. tavaratra* should be considered a single species).

We have made a few changes to the manuscript for clarifying this point, although not at the particular position the reviewer referred to as we think giving too much detail there would impede readability. In particular, we added:

- Lines 151-154: "Instead of relying on arbitrary cut-offs to distinguish intra- from interspecific divergence, we utilise two genetically structured species encompassing both fragmented and continuous populations as empirical benchmarks to derive genus-specific threshold values."
- Lines 501-508: "To do so, we derived genus-specific thresholds for spatial analyses from the two species *M. tavaratra* and *M. lehilaytsara* (including *M. mittermeieri*) to distinguish intra- and interspecific divergence (detailed for each analysis below). These widely distributed, densely sampled, and genetically structured species encompass both fragmented and continuous populations (Fig. S31a), serving as a "null model" of population structure. Their taxonomic status is well-supported, and their extensive spatial and genomic data (Fig. S31a) provide high resolution for accurate threshold estimation."

It is also to be noted that we were quite cautious in our use of thresholds based on this comparison point. For instance, we only rejected an intraspecific clinal variation model for genomic data if the 0.05 quantiles of both NRMSE distributions of a candidate pair were above the 0.95 quantiles of the reference null NRMSE distributions. Conversely, if the 0.95 quantiles of a NRMSE distribution of a candidate pair was below the 0.95 quantiles of the null NRMSE distributions, we considered genetic distances to be congruent with a model of intraspecific structure. Cases that were neither rejecting nor congruent with the intraspecific model were considered inconclusive and required further evidence.

Fig. 1: Genetic structure of *M. tavaratra*. **a**, RADseq samples of all *Microcebus* (candidate) species used in this study (reproduced from Supplementary Fig. S31); **b**, mtDNA (cyt b + cox2) phylogenetic tree of *Microcebus* samples, reproduced from Sgarlata et al.¹⁹; **c**, Recovered phylogeny of *M. tavaratra* based on RADseq samples of this study (reproduced from Supplementary Fig. S1); **d**, Isolation-by-distance in *M. tavaratra*, including individuals from both connected and fragmented populations. Blue dots indicate individual pairwise comparisons consistent with expectations in connected populations (i.e., continuous increase in genetic distance with log of geographical distance) and have been categorised as ‘within-taxon’ comparisons. Red dots indicate individual pairwise comparisons from fragmented populations and have been categorised as ‘between-taxon’ comparison (reproduced from Supplementary Fig. S34).

- L177 - 179: Based on what the manuscript said earlier, I assume that these to-be synonymized species were originally defined solely on mtDNA data? It might be helpful to remind the reader that the initial evidence for these species being distinct was modest.

Reply: Indeed, *M. bongolavensis*, which we propose to synonymize, was initially defined in Olivieri et al.²¹ based on three mitochondrial loci (i.e., monophyly, pairwise genetic distances, and identification of molecular diagnostic sites) and morphometric differences in snout, lower leg and hindfoot length. We briefly mention this now at the respective position in the main manuscript (lines 193-195). In addition, we

now added detailed original description information for all species in the species-by-species accounts in the Supplementary results and discussion.

- L187: "catheмерal" -> is this a commonly used term by non-primatologists? I had never heard it before! In any case, is there a reason why species with this irregular pattern of activity would be relevant?

Reply: We thank the reviewer for this comment, which raises an interesting point. The term indeed seems to have originated in the (lemur) primatological literature and was formally defined by Tattersall²². He introduced the term to refer to activity patterns that are evenly distributed throughout the entire day (i.e., 24 h), which did not fit the established categories "diurnal" and "nocturnal", yet seemed to be present in several lemur and other primate species. However, we removed the sentence from the manuscript because it was slightly redundant with the previous sentence, and we already mention several other potential applications of our framework to lemurs.

- L217: Perhaps briefly mention where the mutation estimate came from, given how central that is to these claims. (This could be as simple as saying "We used a mutation rate inferred from pedigree data.")

Reply: We thank the reviewer for pointing this out. We now mention here that the mutation rate estimate was based on external evidence from per-generation *de novo* primate mutation rates (line 255-256, 691-692) and provide a detailed justification in the Supplementary material (Supplementary methods: lines 1244-1260).

- L226: It might be helpful here to explain that the authors chose not to use fossils because of their phylogenetic distance. (I assume?) Otherwise, fossils are generally superior to mutation rates ...

Reply: We thank the reviewer for pointing this out. While we agree that external lines of evidence such as fossils are considered the gold standard for calibrating evolutionary distances in substitutions per site to substitutions per absolute time units, such calibrations are not available for mouse lemurs, Lemuriformes, or older primate divergences. Given that only external calibrations are available for the sister group Lorisiformes and the nearest crown group, calibration is not available until Euarchontoglires (see reviews of evidence in Appendix 1 of dos Reis et al.²³ and Table S1 of dos Reis et al.²⁴). In addition, because of the recent divergences among mouse lemur species evident from observations of their genetic distances and previous studies²⁵⁻²⁸, there was considerable risk that conventional analyses with clock models would be compromised by biases towards older calibrations²⁹ on top of the technical biases that could be introduced by combining our RADseq data with published genome assemblies. There is also the reasonable expectation that ignoring the coalescent process for mouse lemurs, where internal branch lengths between speciation events are very short, would overestimate the species split times³⁰. Therefore, we applied a strategy to estimate divergence times that avoids the biases of much older external calibrations and concatenation by accounting for incomplete lineage sorting with the MSC model and transforming branch lengths from substitutions per site to substitutions per year based on external evidence from per-generation *de novo* primate mutation rates and mouse lemur generation times. We provide this justification in the Supplementary material now (Supplementary methods: lines 1226-1243) and briefly refer to it in the main manuscript (lines 254-256, 687-693).

- L262 - 263: This is a fairly non-standard approach so it would be useful to provide a bit more context here until the readers can get the full details in the Methods. Also, what is the null expectation? The authors say the correlation is non-significant, but relative to what evolutionary model?

Reply: We thank the reviewer for pointing out that more explanation would be useful here. We therefore modified the first sentence of this paragraph (lines 299-302), which briefly introduces the whole methodology now, i.e., that we first reconstructed trait overlaps along the phylogeny and then compared observed patterns (quantified as Spearman's correlation coefficient between overlap and node age) with those simulated under different evolutionary models to identify the most appropriate model/evolutionary process.

We also thank the reviewer for pointing out that the results regarding the correlation of node age and morphometric/niche overlap were presented in an unclear manner. Significance here indicates merely whether the estimated correlation coefficient is significantly different from zero (e.g., as estimated through permutation). In other words, it indicates whether we find a correlation or not. We rephrased the respective sentences to make this more intuitive for the reader (lines 302-303, 316-318).

- L282 - 284: I had trouble squaring this idea of a "neutral process of niche diversification" with the discussion in the previous section of niche divergence driving speciation. I am sure these two findings are not mutually exclusive, but it would be helpful to put them in context of each other.

Reply: We thank the reviewer for pointing out that clarification was required regarding the connection between these two ideas. In brief, our results suggest that climatic niches of mouse lemurs evolved according to a random walk model over available habitat across the island, which implies stochastic events of colonization of available climatic niches and extinction, without being necessarily driven by systematic adaptation to specific niches (i.e., "neutral process of niche diversification"). To our understanding, this does not preclude at all the possibility that mouse lemur populations may have become isolated because of limited dispersal and/or the formation of natural barriers (which is a largely random process that does not lead to adaptation towards optimum trait values), promoting genetic, ecological and climatic niche divergence and eventually leading to speciation. In fact, of the three models tested (Brownian Motion, Early-Burst, Stabilising Selection), we believe the neutral model (i.e., Brownian Motion) is easiest to reconcile with the hypothesis of niche divergence driving speciation. We emphasise this now in the manuscript (lines 333-336). If we misunderstood the reviewer or the reviewer still thinks that this is unclear, we will be happy to provide further clarification here and in the manuscript.

- L284 - 5: Under a neutral process of niche divergence we would expect low overlap? Wouldn't that depend on the rate of neutral drift?

Reply: We thank the reviewer for pointing out the lack of clarity in this regard. In general, the expectation under neutral niche divergence is that species of a community have relatively high overlapping niches (where niche can refer to both climatic or ecological niche)^{31,32}, as the neutral model's assumption is that species are ecological equivalent and differences in niches are only driven by stochastic random changes, i.e., there is no "deterministic" directionality towards multiple optima or niche differentiation. As the reviewer pointed out, though, the extent of overlap ultimately depends on the rate of neutral drift. However, our statement refers to the comparison with the single adaptive regime (stabilising selection model; OU), one of the two models to which we compared the neutral model (BM). In fact, under the single optimum OU model, the expectation is that all species of the phylogeny will tend to converge towards the same optimum, thus resulting in high niche overlap among species. Accordingly, if OU and BM models have

similar rates of trait evolution (drift), as is the case here (Fig. 2b here, reproduced from Fig. S30 in the Supplementary material), then species evolving under the BM model should show comparably lower overlap than those evolving under the OU model. We now provide clarification (lines 324-326) and added panel c to Fig. S30 (also shown as Fig. 2c below).

Fig. 2: Comparison of early-burst (EB), Brownian motion (BM) and Ornstein-Uhlenbeck (OU) models of climate niche evolution. a and b, Parameter estimates obtained after fitting the data to an EB and OU model, respectively. The parameters correspond to the net rates of trait evolution and trait pair covariation. The results show that the fitted parameters of the EB and OU models are nearly identical with those of the BM model. c, Distribution of climatic niche overlap along nodes of the *Microcebus* phylogeny, obtained from simulations under the BM and OU models. Blue triangles indicate comparisons where the average overlap for a given node is significantly higher in the OU model compared to the BM model, which would be expected when both models show similar net rates of trait evolution.

- L287 - 8: I can see the logic of neutral niche divergence linking to neutral theory of community assembly, but I think it would be helpful to more explicitly make these connections.

Reply: We agree with the reviewer and have made the connection clearer now (lines 326-333).

- L289 - 291: It seems to me that the morphological stasis & niche conservatism could also explain principles of community assembly, right? For example, in L299 - 302: perhaps communities are phylogenetically overdispersed because of high stasis between closely related species.

Reply: We thank the reviewer for raising this point. We agree that this is a valid hypothesis as well and added a sentence to the main manuscript to raise it (lines 355-358). Fully answering this question is beyond the scope of this study, and it will need to be addressed in future work, ideally with a larger set of (lemur) taxa.

- L289 - 309: This entire section felt superfluous to me, only perhaps because it isn't so strongly grounded in data collected in this paper.

Reply: The reviewer is right that this part of the discussion is not strongly grounded on data collected in this work. However, we consider it relevant to report data and conclusions from previous work to fully integrate our results. In particular, we think that the discussion, in general, is a place where interpretation, hypotheses, and ideas are typically framed in the light of a larger picture, by contextualising it within previous findings present in the literature.

- L311 - 334: How do these different approaches to thinking about species units affect which geographic regions are prioritized for conservation?

Reply: The reviewer is raising an important question. As conservation policies often focus on species as primary targets for conservation, it is usually considered that splitting rather than merging species should help increase conservation and protection of endangered populations. Our proposed taxonomic changes (i.e., synonymisation of several formerly recognized species) might be seen as resulting in a loss of efforts to protect forest areas harboring those lineages that are no longer recognized as distinct species. For instance, *M. ganzhorni* and *M. manitatra* as well as the geographic region they occur in (the Mandena Conservation Zone) might receive less conservation attention because they are now considered mere populations of *M. murinus* which is listed as Least Concern by the IUCN. Similarly, conservation efforts might concentrate on regions that are easier to protect at the cost of regions that are no longer considered to harbor unique microendemic *Microcebus* species diversity (such as the Mandena Conservation Zone). To alleviate these negative and misleading effects of classification, it is crucial for conservation policies to also appreciate the value of maintaining important genetic, morphological and ecological diversity of species, e.g., by using the concept of evolutionarily significant units (i.e., by decoupling conservation to a certain extent from the concept of species units). We tried to make these points as clear as possible in the main manuscript (lines 371-389), although we could not go into too much detail due to the word limit and the fact that the specific implications differ between each of the nine candidate groups assessed in our taxonomic revision. However, to make this point clear, we identified evolutionarily significant units deserving separate conservation attention for each candidate group in the Supplementary material and assessed what their current level of protection is, by mentioning all protected areas that overlap with the distributions of these units (see species-by-species accounts in Supplementary results and discussion). We also removed the mention of “conservation units” in the main manuscript and supplements, as we now focus on “evolutionarily significant units” to be more explicit in our terminology and to avoid confusion. In case we misunderstood the reviewer’s point, or the reviewer thinks we should go into more detail in the main manuscript, we kindly ask for clarification.

- L337: I suggest listing the range of # of inds sampled per species for a given data type.

Reply: The range of individuals/records/assessments per species is now given for each data type (lines 401-438).

- L422: These species were the nominal species, right?

Reply: We thank the reviewer for pointing out that it is not clearly described which taxa are referred to here by “species”. In fact, three described species were split into populations for this assignment due to their non-monophyly, which was clearly indicated by maximum likelihood and quartet-based (with individuals as tips) phylogenetic inference. These are *M. lehilahytsara*, *M. murinus* and *M. simmonsii*, as can be seen in Fig. S6. We clearly state this in main text now (lines 483-486).

- L455: How was g_{di} of 0.25 chosen as a threshold?

Reply: This question was also posed by reviewer #1 (referring to p.13 L455). We would therefore like to refer reviewer #2 to the detailed answer given above.

- L602: I would be pretty wary of a diversification analysis on such a small phylogenetic scale - the difference in rates is likely to be a false positive, a notorious problem of these methods (see Rabosky & Goldberg 2015; 10.1093/sysbio/syu131).

Reply: We thank the reviewer for pointing out this caveat, which we are aware of. The association of humid forest habitat and diversification rates has been hypothesized for a number of taxa in Madagascar already^{33,34}. The reconstruction of ancestral habitat along our phylogeny already suggests intuitively that mouse lemur diversification seemed to have occurred predominantly in humid habitats (given that species separate into a larger, mostly humid habitat- and a smaller dry habitat-associated clade), which would further indicate that such an association is a more general pattern. The formal analysis was then conducted to test this in a quantitative way. We decided to include this analysis despite potential caveats as it could be useful in comparison with earlier studies mentioned above or for future multi-clade meta-analyses as suggested by Rabosky & Goldberg³⁵. In addition, this analysis is not part of any major conclusion of the paper nor is it the basis for any follow-up analyses. We agree, though, that the potential caveats were not highlighted, which we now do in the respective section (lines 290-291; Supplementary results and discussion: lines 898-899).

- L622: I would imagine the correlation across these variables is high - is that an issue for interpretation?

Reply: The reviewer is correct in pointing out the lack of clarity in this regard. Indeed, correlation across variables can be relatively high. Accordingly, these analyses were carried out by modeling the correlation structure across variables through a covariance matrix. We have now added the relevant information (lines 768-769).

- L1205: Consider putting a picture of a mouse lemur in this figure. If you work on something cute, might as well take advantage.

Reply: We thank the reviewer for this suggestion but do not think that a mouse lemur picture would fit well in Fig. 1, which is a technical graph detailing a framework that aims to be generalizable (rather than being specifically designed for mouse lemurs). However, we followed the reviewer’s suggestion by putting a mouse lemur illustration in Fig. 2a (line 1317), which refers only to results obtained for the genus *Microcebus*. We will also be happy to share pictures with the journal in case that they might wish to use them for the cover.

- L1205: Check spelling of Acoustic in figure.

Reply: We thank the reviewer for pointing this out. It has been corrected (Fig. 1; line 1298).

- L1225: I was surprised by how young these species splits are - is this typical of these taxa?

Reply: Indeed, our age estimates are relatively recent, but completely in line with recent coalescent and demographic models²⁵⁻²⁸ (it is important to note though that there are overlaps in the underlying sequence data). Notably, fossil-calibrated estimates inferred from concatenated alignments generally yield older divergence times of about 8-10 million years for this genus^{23,36-38}. This discrepancy between mutation-rate calibrated coalescent and fossil-calibrated clock model estimates is likely caused by several interacting factors (discussed in detail in Tiley et al.³⁰). For instance, no fossil calibrations are available for lemurs and the use of phylogenetically distant, external fossils on a concatenated alignment might inflate divergence times for younger nodes²⁹. Moreover, ignoring the coalescent process in mouse lemurs, where internal branch lengths between speciation events are short, might overestimate species split times³⁰. Of course, mutation rate estimates can be biased or imprecise as well, although this alone can not explain the observed difference in estimates (which we show by modelling mutation rates and generation time through a gamma and lognormal distribution, respectively). We mention this discrepancy and the possible reasons for it in the Results and discussion of the manuscript (lines 263-268) and explore them in more detail in the respective part of the Supplementary material (Supplementary results and discussion: lines 832-841). Following the reviewer's comment regarding line 226 above (please see our answer to this related comment there as well), we additionally explain now in the main manuscript (lines 255-256, 687-693) and in more detail in the Supplementary material (Supplementary methods: lines 1226-1243) why we decided to use mutation rates for calibration (i.e., lack of fossils). Taken together, we believe that this information enables the reader to contextualize our age estimates with respect to the (*Microcebus*) literature and to understand possible caveats.

- L1240: The triangles are hard to see - perhaps color could be used instead, given how shallow some of these node heights are?

Reply: We agree with the reviewer that node heights are shallow and that the triangles are hard to see, e.g., for the clade (*M. boraha*, *M. simmonsii*). We are not in favor of using colour to clarify this, however, because of our use of colour for the tips of the phylogeny. Instead, we changed the x-axis scale to stretch the phylogeny horizontally, improving visibility of the triangles (Fig. 3a; line 1334). If the reviewer still thinks that the triangles are hard to see, we are happy to add additional clarification, e.g., by marking them with asterisks.

- Suppl Info: I really enjoyed how the Suppl Info was formatted, with the extra methods immediately followed by the figures. It made it much more digestible. Thanks!

Reply: We thank the reviewer for this positive feedback.

- SI L152: These species by species accounts are very helpful.

Reply: We thank the reviewer for this positive feedback.

- SI L172: I am not a primatologist, so pardon my ignorance - but let's say a female from Species X has a non-overlapping oestrus with a female from Species Y. Does that necessarily mean that a male from Species Y won't try to mate with a female from Species X? In other words, I am curious how well the non-overlapping oestrus patterns reflect reproduction patterns.

Reply: The reviewer is entirely right that such a temporal shift in oestrus activation in females does not necessarily mean that a male from a species Y may not try to mate with a female of species X, if its testes are well developed during the female's oestrus and there are no other pre-copulatory mechanisms in place that preclude interbreeding. Accordingly, non-overlapping oestrus is no sufficient criterion for reproductive isolation per se. Given that closely related mouse lemur sister taxa do not occur in sympatry (with the one exception of *M. murinus* and *M. griseorufus*), this is not what we would like to highlight there. Rather, our findings indicate that there are stable differences in female seasonal reproductive activation between *M. berthae* and the other two candidate species, which give further support for their taxonomic distinctiveness (even if they may not be sufficient to secure reproductive isolation). We edited the sentence to clarify this point (Supplementary results and discussion: lines 182-184).

- SI: On the Figures showing climatic niche (e.g., Fig 7) consider putting the occurrence points on the ENMs so that readers can assess under- and over-fitting.

Reply: We agree with the reviewer that the occurrence points used for niche modeling are crucial information to interpret the resulting models. However, as these are already given in the sampling maps in panel a of these summary figures, we do not think it is necessary to plot them on the ENMs as well, which leaves these already very busy (and small) figures a bit cleaner. If the reviewer disagrees with this, we will be happy to include them.

Reviewer #3 (Remarks to the Author):

In this manuscript the authors re-assess the species diversity of the genus *Microcebus* using a new and integrative framework incorporating multiple types of genomic data alongside morphological, bioclimatic, reproductive and acoustic data. Their stepwise approach and accompanying flowchart seem to be a well-thought-out and appropriate approach that will recognize good species while being conservative and avoiding over-inflation. They find that only 19 of the 26 candidate species are supported and recommend taxonomic deflation. Further, the manuscript yields important insights on phylogeny/paleobiogeography, divergence times and speciation rates, geographic structure of genomic and morphological variables, and the pace of evolution (supporting stabilizing selection and a neutral process of niche diversification).

I feel this is a critically important study that has been executed well and presented clearly. Many of us are tired of the arguments between splitters and lumpers, and this paper is a tangible and important step towards finding meaningful middle ground, and providing a model that could be profitably applied to other taxonomic groups. I was impressed and the comments I offer below are relatively minor.

Reply: We thank reviewer #3 for the very positive and constructive feedback, as well as the encouraging remarks regarding our manuscript.

One larger point I thought could be addressed is the potential stability of the findings in the face of future evidence yet to be gathered. For example, what if currently an assessment of a pair of candidate species is based on 2 local populations for one species and 3 for another. Will future sampling raising this to 3 and 5 be likely to affect the results? Similarly, how about increased morphometric sampling from those new sites and new accumulation of other types of data. I know there are various aspects of the methods where randomization approaches are used, I think in order to address this issue. But, when a geographic

range is relatively large, and you're able to do those randomizations only among the individuals you've sampled from 2 or 3 localities, it wouldn't accomplish the same thing as truly randomly selecting individuals from across the entire geographic range. I wonder if this could be addressed or acknowledged.

Reply: The reviewer is raising a very important point. The stability of the findings in light of additional sampling/evidence differs between the nine candidate groups. We already explicitly addressed this point in the Supplementary results and discussion for some candidate groups (i.e., *M. boraha*/*M. simmons*, *M. M.gerpi*/*M. jollyae*/*M. marohita*, *M. jonahi*/*M. macarthurii*) for which more data is urgently required. In addition, we also already mentioned for most groups how well our sampling covers the species' distributions to inform readers how well our findings are supported by data availability. In fact, for most candidates our sampling (genetics, morphometry and occurrence records) covers large parts of their known distributions, making our findings relatively robust. However, following the reviewer's suggestion, we now mention across all groups how robust our findings are and whether additional sampling is necessary to resolve open questions or could potentially challenge the results (see species-by-species accounts in the Supplementary results and discussion). We also highlight in the main manuscript now that taxonomy is a dynamic process that is constantly re-evaluated upon the acquisition of new data (lines 131-132) and that additional data/sampling will be necessary to clarify open questions and test the robustness of our conclusions (lines 217-221).

Line-by-line comments:

60: I rather disagree with the term "behavioural" (see further comments below) and I think it's clearer if split here into reproductive and acoustic data.

Reply: We agree with the reviewer that some included data (e.g. testis size) are not behavioural and that the suggested categories ('reproductive activity data' and 'acoustic data') are clearer. Because we modified the abstract slightly (lines 62-70), these terms are no longer mentioned there, but we changed "behaviour" to "reproductive activity and acoustic communication" throughout the manuscript (lines 114, 149, 170, 498, 605; see also replies to comments below).

187: lower case for mittermeieri.

Reply: We thank the reviewer for pointing this out. It has been corrected (lines 205).

251: Replace dryer with drier.

Reply: We thank the reviewer for pointing this out. It has been corrected (line 289).

303: Is it not a bit contradictory to say "the trait diversity of lemur communities may have relatively ancient origins" just after showing that the entire *Microcebus* genus is only 1.5 million years old?

Reply: We thank the reviewer for pointing out this seemingly contradictory statement. We modified it to a sentence highlighting the point we originally wanted to make, i.e., that the trait diversity and niche partitioning of lemur communities might predate the actual species diversification of various lemur genera (lines 349-351).

315: I feel "mitigation" is not the right word choice here. Wouldn't it be better to express that the lumping results in a species that is at the same level of endangerment, or a lower level of endangerment, than the

least endangered of the two candidate species that were lumped? That, to me, is the only logical and consistent thing you can say.

Reply: We agree that the wording mentioned above is more appropriate. Accordingly, we rephrased the section to be more precise in our conclusion (361-368) and provide a detailed explanation in the Supplementary material (Supplementary results and discussion: lines 1019-1040).

323: The authors refer to “supplementary results and discussion”, yet I only see two supplementary files: 1) tables and 2) text and figures.

Reply: We thank the reviewer for pointing this out. We renamed the second supplementary file mentioned above to “Supplementary methods, results and discussion, including supplementary figures S1–S38” (Supplementary information: lines 5-6).

373: It is an odd categorization to call everything on this list “Behaviour”. Male testis size is an anatomical characteristic, and the others seem more like life history characteristics. Nothing on the list feels very behavioural. Fig S32D refers to them as “data on reproductive activity” and Table S15 refer to them as “reproductive activity data. In addition, the description in the main text is a bit misleading. For example, I would naturally think of “pregnancy” as some kind of length of time, or timing throughout the year, but what I see in the supplemental table is simply pregnant or not on a specific day of capture. The main text should be clearer.

Reply: As mentioned before, we agree that the term behavioural is misleading. To address this, we split this section into two, called “Reproductive activity” and “Acoustic communication” (lines 429, 435). We also improved the description of reproductive activity data in this section (and the species delimitation part of the methods section) by pointing out that these are presence/absence data of oestrus, pregnancy or lactation in females and of enlarged testes in males at the time of capture (lines 430-433). We also clarified these points in the Supplementary material (Supplementary methods: lines 1136-1139).

486: Lower case for mittermeieri.

Reply: We thank the reviewer for pointing this out. It has been corrected (line 587).

489: lehilahytsara.

Reply: We thank the reviewer for pointing this out. It has been corrected (line 590).

540: NRMSE?

Reply: We thank the reviewer for pointing this out. This entire section has been reworded to improve clarity (lines 625-640).

565: As with previous comment, I feel “behaviour” is not the right heading to use.

Reply: As mentioned before, we agree that the term behavioural was not fully accurate. To address this, we split this section into two, called “Reproductive activity” and “Acoustic communication” (lines 673, 681).

567: oestrus?

Reply: We thank the reviewer for pointing this out. It has been corrected (line 675).

704: I disagree with the tone taken here in the “Conservation reassessment” section and the way it’s explained. First, a published article can’t officially “update” an IUCN endangerment status – only an IUCN-published assessment can do that. The tone here implies that the authors are doing this. I would suggest a re-wording using the wording “new suggested conservation statuses”, or “likely new conservation status”. Secondly, looking at Table S10, there is quite a mix of things going on. For *M. lehilahytsara*, Dolch et al already did the reassessment officially; this should be mentioned somewhere as a counterpoint to the others, which have not been reassessed. There is also quite a wide range of cases. Sometimes (e.g. *murinus+ganzhorni+manitatra*) it is fairly straightforward: a widespread LC species that now has two relatively tiny chunks of habitat added will certainly come out as a LC species. However, for *simmonsii+boraha*, you have an EN species plus a DD species, respectively. Table S10 tells us these authors have decided to make it a VU species, with no explanation. Given the (not perfect but) quantitative and repeatable IUCN endangerment criteria, this comes across as relatively sloppy/arbitrary. I would say there either needs to be justification given for things like this, or the authors back off on the proclamations.

Reply: We agree with the reviewer that there were several issues as mentioned above with this section of the manuscript, which we now address: First, we emphasize in the methods section that we only provide new conservation status recommendations for revised species instead of actual updated IUCN statuses (lines 817-819). Second, we clarify what these recommendations are based on by using the area of occupancy (AOO) and extent of occurrence (EOO) of each species, following guidelines in IUCN (2012). These reassessments were done in collaboration with Daniel Hending, who is now also listed as co-author on the manuscript. We describe the procedure in the methods section (lines 819-827) and detail the underlying criteria for each status recommendation in Supplementary Table S10. In addition, we provide a section in the Supplementary material that details and explains the suggested changes made for each species, which we believe would be too much detail and too long for the main manuscript (Supplementary results and discussion: lines 1019-1040). There, we explicitly stated that the formal reassessment for *M. lehilahytsara* had already been done by Dolch et al.³⁹, and that we confirm it here.

1203: The wording “largely overlapping” in the figure feels a bit arbitrary/qualitative compared to the other, more quantitative criteria. Could we be more objective/quantitative?

Reply: The reviewer is correct, and we agree that the wording was imprecise. Unfortunately, we do not think that the nature of our reproductive activity data allows for an entirely quantitative comparison of reproductive schedules (e.g., concluding an overlap of X percent or months) because sampling effort is very heterogeneous across species and/or reproductive indicators. We therefore analyzed differentiation in reproductive activity on a case-by-case manner, checking in detail what conclusions the data for each of the candidate subsets would allow us to draw, with a particular interest in the timing of activation of reproductive activity (the results of which are given in the species-by-species accounts in the Supplementary results and discussion). The two candidates *M. bongolavensis* and *M. ravelobensis*, which constitute the example given in Figure 1, are actually relatively well sampled with respect to reproductive activity data (Fig. S15h). Looking at the bottommost graph of Figure S15h, we can see that enlarged testes can be observed in both species starting around July. Although our sampling does not cover months prior to this for *M. bongolavensis* (in contrast to *M. ravelobensis*), the absence of regressed testes in both species starting approximately in September further supports the hypothesis that reproductive activity in males begins around the same time. Similarly, the appearance of female oestrus and pregnancy seems to be synchronous for the two candidates. Taken together, our data provide no evidence for differentiation in reproductive activity but indicate synchronous activation of reproductive activity in both species.

Accordingly, we replaced the wording “Largely overlapping” with “Synchronous seasonal activation” in Figure 1 (line 1298), which we believe is more of an objective statement and well supported by our data. We also improved clarity of this in the main text (lines 190-193) as well as in the Supplementary results and discussion for this case (*M. bongolavensis*-*M. ravelobensis*) and the two other cases *M. macarthurii*-*M. jonahi* and *M. lehilahytsara*-*M. mittermeieri* (Supplementary results and discussion: lines 243-244, 554-555, 706-715). Finally, we also replaced “Overlap = 0.391” with “ $D = 0.391$ ” and “Overlap = 0.424” with “Hypervolume overlap = 0.424” in the rightmost column of Figure 1 (line 1298) to be more precise for these lines of evidence as well.

1203: Correct “Acoustic” in the figure

Reply: We thank the reviewer for pointing this out. It has been corrected (Fig. 1; line 1298).

Additional changes:

-We added repository names in lines 830-833.

-We added BioSample accessions for RADseq samples in Supplementary Table S11.

-We changed the figure panel labeling from majuscule to minuscule letters.

-We removed the former Table S11 and Fig. S31.

References

1. Jackson, N. D., Carstens, B. C., Morales, A. E. & O’Meara, B. C. Species delimitation with gene flow. *Syst. Biol.* **66**, 799–812 (2017).
2. Hapke, A., Gligor, M., Rakotondrany, S. J., Rosenkranz, D. & Zupke, O. Hybridization of mouse lemurs: Different patterns under different ecological conditions. *BMC Evol. Biol.* **11**, 1–17 (2011).
3. Teixeira, H. *et al.* RADseq data suggest occasional hybridization between *Microcebus murinus* and *M. ravelobensis* in northwestern Madagascar. *Genes* **13**, 913 (2022).
4. Poelstra, J. W. *et al.* RADseq data reveal a lack of admixture in a mouse lemur contact zone contrary to previous microsatellite results. *Proc. R. Soc. B* **289**, 20220596 (2022).
5. Teixeira, H. *et al.* Retraction: Teixeira *et al.* RADseq data suggest occasional hybridization between *Microcebus murinus* and *M. ravelobensis* in northwestern Madagascar. *Genes* 2022, 13, 913. *Genes* **13**, 2146 (2022).
6. Li, H. & Ralph, P. Local PCA shows how the effect of population structure differs along the genome. *Genetics* **211**, 289–304 (2019).
7. Nei, M. & Li, W. H. Mathematical model for studying genetic variation in terms of restriction endonucleases. *Proc. Natl. Acad. Sci.* **76**, 5269–5273 (1979).
8. Junker, R. R., Kuppler, J., Bathke, A. C., Schreyer, M. L. & Trutschnig, W. Dynamic range boxes – a robust nonparametric approach to quantify size and overlap of *n*-dimensional hypervolumes. *Methods Ecol. Evol.* **7**, 1503–1513 (2016).
9. Padial, J. M., Miralles, A., De la Riva, I. & Vences, M. The integrative future of taxonomy. *Front. Zool.* **7**, 16 (2010).
10. Mayr, E. *Systematics and the Origin of Species from the Viewpoint of a Zoologist*. (Columbia University Press, 1942).
11. Hausdorf, B. & Hennig, C. Species delimitation and geography. *Mol. Ecol. Resour.* **20**, 950–960 (2020).
12. Bradburd, G. S., Coop, G. M. & Ralph, P. L. Inferring continuous and discrete population genetic structure across space. *Genetics* **210**, 33–52 (2018).
13. Guillot, G., Renaud, S., Ledevin, R., Michaux, J. & Claude, J. A unifying model for the analysis of phenotypic, genetic, and geographic data. *Syst. Biol.* **61**, 897–911 (2012).
14. Boettiger, C., Coop, G. & Ralph, P. Is your phylogeny informative? Measuring the power of

- comparative methods. *Evolution (N. Y.)*. **66**, 2240–2251 (2012).
15. Slater, G. J. & Pennell, M. W. Robust regression and posterior predictive simulation increase power to detect early bursts of trait evolution. *Syst. Biol.* **63**, 293–308 (2014).
 16. Pennell, M. W., Fitzjohn, R. G., Cornwell, W. K. & Harmon, L. J. Model adequacy and the macroevolution of angiosperm functional traits. *Am. Nat.* **186**, E33–E50 (2015).
 17. Voje, K. L., Starrfelt, J. & Liow, L. H. Model adequacy and microevolutionary explanations for stasis in the fossil record. *Am. Nat.* **191**, 509–523 (2018).
 18. Salmona, J. Comparative conservation genetics of several threatened lemur species living in fragmented environments (PhD thesis). (Instituto Gulbenkian de Ciência - ITQB - UNL Oeiras ITQB, 2015).
 19. Sgarlata, G. M. *et al.* Genetic and morphological diversity of mouse lemurs (*Microcebus* spp.) in northern Madagascar: The discovery of a putative new species? *Am. J. Primatol.* **81**, e23070 (2019).
 20. Aleixo-Pais, I. *et al.* The genetic structure of a mouse lemur living in a fragmented habitat in Northern Madagascar. *Conserv. Genet.* **20**, 229–243 (2019).
 21. Olivieri, G. *et al.* The ever-increasing diversity in mouse lemurs: Three new species in north and northwestern Madagascar. *Mol. Phylogenet. Evol.* **43**, 309–327 (2007).
 22. Tattersall, I. Cathemeral activity in primates: A definition. *Folia Primatol.* **49**, 200–202 (1987).
 23. Dos Reis, M. *et al.* Using phylogenomic data to explore the effects of relaxed clocks and calibration strategies on divergence time estimation: Primates as a test case. *Syst. Biol.* **67**, 594–615 (2018).
 24. Dos Reis, M. *et al.* Uncertainty in the timing of origin of animals and the limits of precision in molecular timescales. *Curr. Biol.* **25**, 2939–2950 (2015).
 25. Poelstra, J. W. *et al.* Cryptic patterns of speciation in cryptic primates: Microendemic mouse lemurs and the multispecies coalescent. *Syst. Biol.* **70**, 203–218 (2021).
 26. Yoder, A. D. *et al.* Geogenetic patterns in mouse lemurs (genus *Microcebus*) reveal the ghosts of Madagascar's forests past. *Proc. Natl. Acad. Sci.* **113**, 8049–8056 (2016).
 27. Tiley, G. P. *et al.* Population genomic structure in Goodman's mouse lemur reveals long-standing separation of Madagascar's Central Highlands and eastern rainforests. *Mol. Ecol.* **31**, 4901–4918 (2022).
 28. van Elst, T. *et al.* Diversification processes in Gersp's mouse lemur demonstrate the importance of rivers and altitude as biogeographic barriers in Madagascar's humid rainforests. *Ecol. Evol.* **13**, e10254 (2023).
 29. Angelis, K. & Dos Reis, M. The impact of ancestral population size and incomplete lineage sorting on Bayesian estimation of species divergence times. *Curr. Zool.* **61**, 874–885 (2015).
 30. Tiley, G. P., Poelstra, J. W., dos Reis, M., Yang, Z. & Yoder, A. D. Molecular clocks without rocks: New solutions for old problems. *Trends Genet.* **36**, 845–856 (2020).
 31. Chave, J. Neutral theory and community ecology. *Ecol. Lett.* **7**, 241–253 (2004).
 32. Gravel, D., Canham, C. D., Beaudet, M. & Messier, C. Reconciling niche and neutrality: The continuum hypothesis. *Ecol. Lett.* **9**, 399–409 (2006).
 33. Crottini, A. *et al.* Vertebrate time-tree elucidates the biogeographic pattern of a major biotic change around the K-T boundary in Madagascar. *Proc. Natl. Acad. Sci.* **109**, 5358–5363 (2012).
 34. Everson, K. M., Soarimalala, V., Goodman, S. M. & Olson, L. E. Multiple loci and complete taxonomic sampling resolve the phylogeny and biogeographic history of tenrecs (Mammalia: Tenrecidae) and reveal higher speciation rates in Madagascar's humid forests. *Syst. Biol.* **65**, 890–909 (2016).
 35. Rabosky, D. L. & Goldberg, E. E. Model inadequacy and mistaken inferences of trait-dependent speciation. *Syst. Biol.* **64**, 340–355 (2015).
 36. Herrera, J. P. & Dávalos, L. M. Phylogeny and divergence times of lemurs inferred with recent and ancient fossils in the tree. *Syst. Biol.* **65**, 772–791 (2016).
 37. Louis, E. E. & Lei, R. Mitogenomics of the family Cheirogaleidae and relationships to taxonomy and biogeography in Madagascar. in *The dwarf and mouse lemurs of Madagascar: biology, behavior and conservation biogeography of the Cheirogaleidae* (eds. Lehman, S. M., Radespiel, U. &

- Zimmermann, E.) 54–93 (Cambridge University Press, 2016).
38. Everson, K. M. *et al.* Not one, but multiple radiations underlie the biodiversity of Madagascar's endangered lemurs. *bioRxiv* 2023.04.26.537867 (2023) doi:10.1101/2023.04.26.537867.
 39. Dolch, R., Schübler, D., Radespiel, U. & M, B. *Microcebus lehilahytsara*. *The IUCN Red List of Threatened Species*. (2022).Decision Letter, second revision:

16th April 2024

Dear Dr Salmons,

Your manuscript entitled "An integrative and generalizable approach to elucidate cryptic diversifications sheds light on mouse lemur taxonomy and evolution" has now been seen by the same three reviewers, whose comments are attached. It's clear that the manuscript is progressing well, and reviewer 1 signs off. Reviewers 2 and 3 are pleased with the revisions but have some more comments that suggest that the paper would benefit from a further round of revision before it's accepted for publication. We will therefore need to see your responses to the criticisms raised and to some editorial concerns, along with a revised manuscript, before we can reach a final decision regarding publication.

We therefore invite you to revise your manuscript taking into account all reviewer and editor comments. Please highlight all changes in the manuscript text file.

* If you have not done so already please begin to revise your manuscript so that it conforms to our Article format instructions at <http://www.nature.com/natecolevol/info/final-submission>. Refer also to any guidelines provided in this letter.

* Include a revised version of any required reporting checklist. It will be available to referees (and,

40potentially, statisticians) to aid in their evaluation if the manuscript goes back for peer review. A revised checklist is essential for re-review of the paper.

[REDACTED]

Nature Ecology & Evolution is committed to improving transparency in authorship. As part of our efforts in this direction, we are now requesting that all authors identified as 'corresponding author' on published papers create and link their Open Researcher and Contributor Identifier (ORCID) with their account on the Manuscript Tracking System (MTS), prior to acceptance. ORCID helps the scientific community achieve unambiguous attribution of all scholarly contributions. You can create and link your ORCID from the home page of the MTS by clicking on 'Modify my Springer Nature account'. For more information please visit www.springernature.com/orcid.

Yours sincerely,

[REDACTED]

Reviewer expertise:

as before

Reviewers' comments:

Reviewer #1 (Remarks to the Author):

I have looked at the revised version of the ms. and the authors' response to my comments. I think they have adequately dealt with my concerns, and I support the publication of the paper in Nature Ecology and Evolution.

Reviewer #2 (Remarks to the Author):

In this manuscript, van Elst and colleagues define species boundaries in the mouse lemur genus *Microcebus* and then use the revised taxonomy to infer the genus's evolutionary history. This manuscript is a revision of a previous submission; I served as a reviewer on the original submission as well. In general, I thank the authors for their careful consideration of all points made by me and the other reviewers. I think the manuscript is much clearer. I have a few remaining concerns.

1. The use of taxonomically-informed cutoffs needs more explanation earlier in the paper. L151 - L154 is a great start, but I wanted to see a clearer explanation earlier in the paper so that readers can interpret the findings in context of the authors' approach. (I realize this is a challenge of manuscripts where Results come before Methods.) Why *M. tavaratra*, *M. mittermeieri*, and *M. lehilahytsara* are good baselines never became that clear to me - especially because the *M. tavaratra* patterns of IBD do show a clear discontinuity. How do the authors know that doesn't reflect an incipient lineage? Is it as simple that the authors have the best data for these species? (Note - the authors' argument is very clear in their rebuttal. While I don't agree with them, I understand their reasoning. I would have liked to see these same arguments made in text, particularly early enough so that the readers understand the logic.)

2. Similarly, the focus on genomic independence (as measured by *gdi*, IBD, reciprocal monophyly, and no admixture) is now better explained in the Methods. But, I think this needs to come earlier. While this focus on genomic independence might be sensible for mouse lemurs, it does not make as much sense in many other taxonomic contexts. I think justifying this choice early would help the reader. That said, I am not sure I understand the authors' argument that we don't expect gene flow in mouse lemurs. Relying on patterns in sympatry doesn't quite work, because we cannot see all the attempts at secondary

42sympatry that failed because the species hybridized to extinction. Similarly, the idea that gene flow suggests species aren't distinct seems to assume that gene flow will invariably lead to species homogenization. I'm not so sure about that argument -- increasingly, we are seeing that lineages can remain distinct under even high levels of gene flow.

3. The authors make a great point that which traits are considered for species delimitation should vary based on the taxonomic group being studied. I agree and I would encourage them to add more information on why they chose to focus on morphometric measurements, climatic niche overlap, reproductive behavior, and acoustic calls for the mouse lemurs.

4. Now that I understand the species delimitation approach, I was able to better parse all the species-by-species decisions. There are 3 cases that confuse me. First, *M. boraha* was synonymized even though there were no ecological / reproductive / morphological data to identify how differentiated it was from *M. simmons*. I can see the authors' arguments that this is the more conservative approach, but one could argue that any change from an existing taxonomy is the less conservative solution. It strikes me as premature to collapse the lineages. (It was equally premature to elevate the lineages, to be fair.) Second, I would make a similar argument for *M. marohita*. Third, I wasn't sure why *M. murinus* North wasn't retained as a distinct lineage - it is monophyletic, has high GDI scores, shows no evidence of admixture. IBD is inconclusive but it appears to be distinct in climatic space & reproductive behavior. So what ultimately led to its collapse? I thought the species only had to be distinct in one trait (as shown by the decision tree in Figure 1) - but perhaps I am misunderstanding?

MINOR NOTES

- L217: I appreciate that a few genes is not ideal, but is it truly fair to call it bad taxonomic practice? Given the cost of sequencing - and the inaccessibility of it still to many folks around the globe - requiring genomic data might make taxonomy inaccessible for many researchers worldwide.
- L293 - 296: Do the revised geographic boundaries (for the revised species boundaries) map to steep environmental gradients and rivers, as suggested here?
- L298: Consider briefly discussing here what morphometric traits were analyzed.
- L352: Were mouse lemurs last to arrive in the communities to which they belong?
- L538: did the authors test for best-fitting K? If so, I would recommend including that in figures.

Reviewer #3 (Remarks to the Author):

This is my second review of this manuscript.

43In this manuscript the authors re-assess the species diversity of the genus *Microcebus* using a new and integrative framework incorporating multiple types of genomic data alongside morphological, bioclimatic, reproductive and acoustic data. As before, I'm impressed by the well-thought-out stepwise approach and accompanying flowchart, execution and communication of analyses, and the concise, clear writing. I feel this will be an important and well-cited contribution to lemur taxonomy, and is a great step in finding the right compromise between lumpers and splitters.

I am also very impressed with the great care taken by the authors to respond to my comments, and those of the other reviewers. I appreciate both the written response, and the careful revisions made. I'm therefore left with little in terms of major comments; below I offer just one conceptual question/concern and a short list of minor comments that would improve the manuscript presentation a little.

Conceptual Question (and I apologize for not raising this last time):

In considering the nicely-presented Figure 1, I'm just struck by the fact that of the 8 criteria for testing, seven are really quite intrinsic to the species (pair) – four genomic criteria plus morphometry, reproductive activity and acoustic communication. The final one, climatic niche, measures something external (extrinsic) to the species' biology, even if you could argue that it tends to cause biological adaptations to those conditions.

I acknowledge that ENM (Ecological Niche Modelling) is a really useful tool in understanding species' ecologies, and in looking at divergence among closely related species. However, Figure 1 tells us that in a situation where you have sister populations that are reciprocally monophyletic and show genetic divergence and clustering, then even when those two populations don't diverge in morphometry, reproductive activity or acoustic communication, their divergence in ecological niche would lead to confirmation as distinct species.

To perhaps be more clear: consider two test cases A and B in which the pairs had the same degree of genomic divergence (high enough to pass the four tests) but were similar in morphometry, reproductive schedule and acoustic communication. You'd determine there was a single species in pair A, where the two populations occupied similar habitats, but you'd determine there were two species in B, where the two populations occupied climatically different habitats.

It occurs to me this is a way that one might erroneously recognize species where you in reality have one ecologically flexible species that has colonized multiple habitats in the relatively distant past, remained somewhat reproductively isolated, and diverged genetically. This is also a case where sampling would matter – especially if sampling didn't happen in intermediate climatic zones. And, given that species differ considerably along the generalist-specialist continuum, this seems like a concern, especially in applying this framework to other genera and orders.

The examples I'm thinking of are the aye-aye (which occupies east and west forests in Madagascar), as

well as humans (perhaps an unfair example, but also a primate).

I feel the authors have thought through these issues more than me. Is the answer here that “well, if the gdi is high and other genomic separation is there, then it’s valid”? Or perhaps that *Microcebus* species are more specialized than my two funny examples, so it’s not a fair comparison? I just wanted to offer this comment to the authors and the editor; if I’m wrong, perhaps a brief mention of why it’s valid, or the errors it could potentially lead to, might be nice.

Line-by-line comments

97: “increasing unanimity” is a paradoxical term, rather like “more unique” or “less unique”. Suggest replacing with “increasing agreement”.

112: “morphologically static” suggests all species are morphologically identical. Suggest “relatively morphologically static? Or “while showing relatively little morphological divergence”?

124: Replace “Our work does not only shed light on... provides” with “Our work sheds light on the taxonomy and diversification of the genus *Microcebus*, while also providing ...” (edit for clarity).

131: I question whether this first paragraph of the results is best placed here. After coming through the introduction, when the reader sees “Results” she/he is ready for results. This is a rather long paragraph that feels like discussion and repeats much of what came in the introduction.

221: I think it’s quite a useful product of this study that it “identified key areas where further sampling is necessary”. However, I don’t see it (I think) in the main text. I would prefer this to be more available in the main text – not a long treatment, but if there are 2 or 3 places (species pairs) that really could benefit from more sampling, I’d mention them right here.

272: should be “contrasts with”?

354: I feel “also occupied” would work better than “occupied also”.

356: Awkward; perhaps “may have served to intensify”? or just “intensified”?

357: Similarly, suggest replacing “leading to rare co-occurrence” with “resulting in low levels of co-occurrence...”

423: Remove “i.e.,” as this is a complete list.

503: correct spelling, “lehilahytsara”

616: There is a word missing after “intraspecific”, possibly “variation” – even so, “allows predicting” is an awkward phrase too.

673: Compared with the rest of this section, this feels “rough”. We are given two sentences; the first says ‘we checked for differences’ and the second is rather unclear (should be cleaned up) but I believe it tells us that lactation and pregnancy were used to count backwards to a putative estrus date (or range of dates). I feel this is not the most important piece of the methods, but it should be buttressed a little so that we at least understand what goes into the decision of “different” or “same”.

1326: English readers may not be universally familiar with the term “minuscule”... perhaps just “lower

case letters"? If you change, change within supplemental material as well.

1353: For Fig 4c, am I wrong, or is there a discrepancy between the order of colors in the plot (red, green, blue) and the order shown in the legend (blue, red, green)? Better if they match, no?

*****END*****

Author Rebuttal, first revision:

Reviewers' comments:

Reviewer #1 (Remarks to the Author):

I have looked at the revised version of the ms. and the authors' response to my comments. I think they have adequately dealt with my concerns, and I support the publication of the paper in Nature Ecology and Evolution.

Reply: We sincerely thank Reviewer #1 for the positive review of our revised manuscript. We appreciate the recognition of the effort we put into addressing the comments. The detailed feedback significantly improved the manuscript's flow and clarity.

Reviewer #2 (Remarks to the Author):

In this manuscript, van Elst and colleagues define species boundaries in the mouse lemur genus *Microcebus* and then use the revised taxonomy to infer the genus's evolutionary history. This manuscript is a revision of a previous submission; I served as a reviewer on the original submission as well. In general, I thank the authors for their careful consideration of all points made by me and the other reviewers. I think the manuscript is much clearer. I have a few remaining concerns.

46Reply: We thank Reviewer #2 for the second thorough and positive review. The insightful comments have significantly improved the manuscript, and we address below the remaining points which further enhance readability and results presentation.

1. The use of taxonomically-informed cutoffs needs more explanation earlier in the paper. L151 - L154 is a great start, but I wanted to see a clearer explanation earlier in the paper so that readers can interpret the findings in context of the authors' approach. (I realize this is a challenge of manuscripts where Results come before Methods.) Why *M. tavaratra*, *M. mittermeieri*, and *M. lehilahytsara* are good baselines never became that clear to me - especially because the *M. tavaratra* patterns of IBD do show a clear discontinuity. How do the authors know that doesn't reflect an incipient lineage? Is it as simple that the authors have the best data for these species? (Note - the authors' argument is very clear in their rebuttal. While I don't agree with them, I understand their reasoning. I would have liked to see these same arguments made in text, particularly early enough so that the readers understand the logic.)

Reply: We thank the reviewer for pointing out that more explanation was still required regarding this methodology. Following Reviewer #3's comment on line 131 and Reviewer #1's first comment in the previous revision round, we now introduce our framework already in the section Main instead of the Results and discussion (lines 121-137). There, we also briefly provide the main arguments for our use of variation among *M. tavaratra*/*M. lehilahytsara* populations to obtain taxonomically-informed cutoffs, i.e., that these species comprise both fragmented and continuous populations, that sampling is good, and that patterns of gene flow and IBD are well-characterised. We believe that this is an appropriate compromise between conveying the main ideas of the approach while not getting lost in detail at this point of the manuscript. Nevertheless, we also added a more detailed explanation to the Methods (lines 554-571, 611-614) as well as the Supplementary methods (lines 1206-1256; Fig. S32).

With respect to the reviewer's question regarding potential incipient speciation in *M. tavaratra*, we are aware that the plot of geographic against genetic distances shows a relatively clear discontinuity. This can result, for instance, as a consequence of stochastic processes, smaller-scale fragmentation effects and/or, as the reviewer suggested, incipient speciation (due to geographic isolation). To understand which of these processes most likely explains the discontinuity observed in *M. tavaratra*, let us first clarify that comparisons were labelled as "within" or "between" on the basis of the discontinuity in the first place, regardless of associated sampling locality (i.e., k-means clustering was used to identify the two groups of comparisons which we labelled as "between" and "within"; we make this clearer now in the Supplementary methods; 1207-1215). This was done to obtain a reference for levels of discontinuity in IBD that could be observed in a single spatially structured species where we would not expect incipient speciation. If allopatric speciation explained the discontinuity, we would expect

47comparisons labelled as “within” to be geographically clustered, while those labelled as “between” would refer to comparisons between such clusters. If we now have a look at the associated sampling localities (Fig. 1 herein), we see that “between” comparisons are not restricted to two specific sites but rather represent comparisons between several forest patches at varying distances to each other (Fig. 1a), even though we acknowledge that comparisons involving one of three sampling sites (named X, Y and Z in Fig. 1) account for many such data points. In addition, comparisons among these forest patches provide several data points labelled as “within” (i.e., “within” comparisons are not only found within these forest patches), and this also includes sites X, Y and Z (Fig. 1b). A detailed characterisation of this spatial structure is given in Salmons (2015), Aleixo-Pais et al. (2019) and Sgarlata et al. (2019). Taken together, the observed genetic structure in *M. tavaratra* is more complex than what we would expect if there was incipient speciation between geographically isolated lineages. Instead, we argue that it can best be explained by stochastic processes and (recent) habitat fragmentation across the entire distribution of the species.

Fig. 1: Geographic representation of pairwise comparisons of *M. tavaratra* individuals used for the analysis of IBD. Edges connect pairs of individuals for which genetic and geographic distances were estimated (a: comparisons labelled as “between” and red in Fig. S32c; b: comparisons labelled as “within” and blue in Fig. S32c). Although three sampling sites (named X, Y and Z) account for the majority of comparisons labelled as “between”, the latter are not restricted to these sites but involve several additional forest patches (a). Conversely, comparisons among these forest patches also provide several data points labelled as “within”, including the sites X, Y and Z (b).

2. Similarly, the focus on genomic independence (as measured by gdi, IBD, reciprocal monophyly, and no admixture) is now better explained in the Methods. But, I think this needs to come earlier. While this focus on genomic independence might be sensible for mouse lemurs, it does not make as much sense in many other taxonomic contexts. I think justifying this choice early would help the reader. That said, I am not sure I understand the authors' argument that we don't expect gene flow in mouse lemurs. Relying on patterns in sympatry doesn't quite work, because we cannot see all the attempts at secondary sympatry that failed because the species hybridized to extinction. Similarly, the idea that gene flow suggests species aren't distinct seems to assume that gene flow will invariably lead to species homogenization. I'm not so sure about that argument -- increasingly, we are seeing that lineages can remain distinct under even high levels of gene flow.

Reply: We thank the reviewer for raising this point again, which led us to reconsider how to account for hybridization/introgression in our framework. As a result, we have now changed the order of genomic tests. More specifically, the IBD-based statistical test, which essentially explores whether there is a detectable “gap” in the plot of geographic vs genetic distances across genomic loci when considering individual comparisons between vs. within candidate lineages, is now the first step in our framework (Fig. 1 in main manuscript). In other words, it has priority over the remaining genomic analyses and can readily lead to the confirmation of candidates or their synonymisation. This is because we expect on average measurably higher genetic distances (at similar geographic distances) between candidate individuals than within candidate individuals if these truly represent separately evolving metapopulations. Moreover, this test allows for occasional introgression/hybridization as the NRMSE distributions integrate data over all genomic windows and individuals (unlike clustering analysis and maximum likelihood phylogenetic inference, where a single individual pattern can lead to a strong conclusion, i.e., the presence of gene flow and absence of reciprocal monophyly, respectively).

Let us now consider the three possible outcomes of this test:

- **Confirmation of candidates as distinct species:** When the 0.05 quantiles of the estimated NRMSE distributions exceed the threshold (0.95 quantile) of the reference species (*M. tavaratra*/*M. lehilahytsara*), we confirm the two candidates as distinct species. In other words, the genetic distances between candidates are much higher than within candidates, and this can not be explained by geography and fragmentation alone. The reframed approach allows for gene flow and hybridization even between confirmed species candidates. This is possible because the associated analyses of admixed ancestry proportions, or non-reciprocal monophyly, no longer precede the IBD test. The genetic distance (or “gap”) required to confirm candidate species and the amount of hybridization allowed is eventually modulated by the NRMSE threshold, which is study system-specific.
- **Synonymisation of candidates:** When individuals of different candidates have similar genetic distances as those within the same candidate (majority of one NRMSE distribution is below threshold), we consider this strong evidence for a genetic cline (simple model of IBD) and therefore an intraspecific model. In such cases, we most likely also observe admixture, no reciprocal monophyly and low to intermediate *gdi*.
- **Inconclusiveness:** If none of the two above-outcomes are supported, the IBD test is considered inconclusive. We therefore follow the flowchart framework downstream (Fig. 1 in main manuscript), using heuristics such as a lack of reciprocal monophyly, admixture and low *gdi* to synonymise candidates, and differences in morphometry, climatic niches, acoustic communication and reproductive activity to confirm candidates.

In summary, our framework allows for the presence of gene flow between confirmed candidates if there is a clear gap in genetic distances between vs. within candidates (at similar geographic distances). Heuristics of genomic independence and additional lines of evidence are only employed (in the way they were before) when the test fails to come to a conclusion. This update allows our framework to be applicable to study systems with recent or ongoing hybridization. It also no longer precludes the possibility of gene flow between *Microcebus* species. Notably, this change in our framework does not change the taxonomic conclusions drawn in this study. We adapted all respective sections and figures in the manuscript (lines 201-212, 574-583, 628-648, 1365-1374, 1405-1419; see also species-by-species accounts in the Supplementary results and discussion). We also added an illustration of the logic of the IBD-based test to the Supplementary methods (lines 1259-1277; Fig. S33).

3. The authors make a great point that which traits are considered for species delimitation should vary based on the taxonomic group being studied. I agree, and I would encourage them to add more

information on why they chose to focus on morphometric measurements, climatic niche overlap, reproductive behavior, and acoustic calls for the mouse lemurs.

Reply: We thank the reviewer for this suggestion, and we agree that it is an important point that was not yet properly covered in our manuscript. Because of the cryptic nature of mouse lemurs and of our integrative objectives, we used a holistic approach, considering all available data. However, while there are obvious practical reasons to include certain lines of evidence (e.g., coordinates and data on morphometry and/or reproductive activity are routinely collected by most researchers during sampling), we also believe that consistent differences in these traits are taxonomically informative in mouse lemurs for the following reasons:

- **Morphometry:** Even though the genus is considered cryptic, quantitative analyses might reveal consistent morphometric differences between *Microcebus* lineages (e.g., Schüßler et al., 2023). Due to the apparent lack of morphological variation in *Microcebus* species, we consider such differences (accounting for geographic variation), if accompanied by genomic differentiation, evidence to confirm candidate species.
- **Climatic niche:** Most described *Microcebus* species are confined to relatively small geographic areas (i.e., micro-endemism, but see *M. murinus* and *M. lehilahytsara*). While most allopatric sister lineages occupy neighbouring regions and are therefore expected to share most of their climatic niche, sister lineages using drastically different bioclimatic niches may be adapted to the latter. We therefore consider pronounced differences in climatic niche space, if accompanied by genomic differentiation, evidence to confirm candidate species. While climatic niche may be informative, we now acknowledge in the manuscript that it has to be integrated carefully (lines 656-658, 718; see also conceptual question of reviewer #3).
- **Reproductive activity/acoustic (mating) calls:** Such traits are probably the most obvious proxies of reproductive isolation because their differentiation can preclude interbreeding and lead to speciation. However, we also consider them as potential consequence of divergence. Therefore, we treat consistent and pronounced differences in these traits as evidence to confirm candidate species, if accompanied by genomic differentiation. Again, conclusions have to be drawn carefully, e.g., to avoid over-interpreting slight shifts in reproductive activity, that are due to variation in climatic conditions (see, for example, delimitation of *M. murinus* in Supplementary results and discussion; lines 705-706).

We now mention these considerations in the Methods (lines 664-667, 712-718, 737-741) as we do not want to go into these details in the introduction (“Main”) or the Results and discussion section.

4. Now that I understand the species delimitation approach, I was able to better parse all the species-by-species decisions. There are 3 cases that confuse me. First, *M. boraha* was synonymized even though there were no ecological / reproductive / morphological data to identify how differentiated it was from *M. simmons*. I can see the authors' arguments that this is the more conservative approach, but one could argue that any change from an existing taxonomy is the less conservative solution. It strikes me as premature to collapse the lineages. (It was equally premature to elevate the lineages, to be fair.) Second, I would make a similar argument for *M. marohita*. Third, I wasn't sure why *M. murinus* North wasn't retained as a distinct lineage - it is monophyletic, has high GDI scores, shows no evidence of admixture. IBD is inconclusive but it appears to be distinct in climatic space & reproductive behavior. So what ultimately led to its collapse? I thought the species only had to be distinct in one trait (as shown by the decision tree in Figure 1) - but perhaps I am misunderstanding?

Reply: We thank the reviewer for the detailed examination of all the species-by-species decisions. Indeed, the three cases mentioned here are challenging, and we present our considerations and reasoning for each of those in the following:

- ***M. boraha* - *M. simmons*:** The delimitation of *M. simmons* and *M. boraha* is indeed difficult because of the very limited amount of available data that limit genetic analyses and do not allow quantification of differences in morphometry, climatic space, reproductive activity and communication. If we naively call individuals at Ambodiriana *M. simmons* (as in Poelstra et al. 2021), admixture analysis and a lack of reciprocal monophyly suggest synonymisation of *M. boraha* (Fig. S11bcd). This is also supported by a continuous pattern of IBD when considering comparisons within *M. simmons* and between *M. boraha* and *M. simmons* (Fig. S11d, bottom), indicating that genetic distances can be explained by geographic distribution rather than a speciation event. It is true that we no longer observe reciprocal monophyly and mixed clusters if we call Ambodiriana individuals *M. boraha* instead (as indicated by phylogenetic inference; Fig. S11bc). However, we still find a relatively continuous IBD pattern (Fig. 2 herein), there is only a comparably small number of substitutions separating the two candidates in the phylogeny (Fig. S11b), and we lack sampling to test for clinal variation. As we present in the manuscript, our approach aims to be conservative in the sense that we only split species that do not show significant discontinuity in patterns of IBD if we observe differentiation in genomic data and at least one additional line of evidence (i.e., if we have convincing evidence to reject the null hypothesis of a single-species model). Following this logic, we indeed consider the description of *M. boraha* as a distinct species (based on few individuals and six loci) premature. We also suggest synonymising *M. boraha* in the second scenario (i.e., when treating Ambodiriana individuals as *M. boraha*) until more evidence becomes available because the genomic differentiation, although detectable, is low and we do not have supporting evidence for

53differentiation in any other trait (regardless of whether this is due to lack of data). We agree with the reviewer that the point can also be made “that any change from an existing taxonomy is the less conservative solution”. However, it is arguable whether a species description not strongly grounded in data should be considered more valid just because it is associated with a publication. Moreover, we aim to consistently delimit species across the genus by applying the same framework and considering the same null hypothesis for all candidate groups to overcome previous biases and avoid oversplitting. These are the points we would like to make in this publication and the reasons for our taxonomic decision regarding these two candidates.

Fig. 2: Effects of different labelling of Ambodiriana individuals on patterns of IBD (a: labelled as *M. simmonsii*; b: labelled as *M. boraha*). Top: Normalised root mean square error (NRMSE) distributions of within and between candidate IBD across genomic windows (colour indicates focal taxon for within candidate IBD; vertical lines indicate different thresholds for species delimitation); bottom: genome-wide patterns of IBD in the candidate group.

- M. marohita* - *M. jollyae*:** Similar to the case of *M. simmonsii* and *M. boraha*, the delimitation of *M. marohita* and *M. jollyae* is impeded by the very limited amount of available data that do not allow quantification of differences in traits other than genetics. In addition, genetic data are restricted to two individuals per candidate species. Our decision to synonymise is based on the facts that structure within *M. gerpi* appears earlier than between *M. marohita* and *M. jollyae* in admixture analysis (Fig. S12c) and that genetic distances between these two candidates are similar to those we observe among *M. gerpi* individuals (Fig. S12d, bottom). This is also indicated by the fact that *M. jollyae* and *M. marohita* are separated by a similar number of substitutions in the phylogeny as the two *M. gerpi* lineages (Fig. S12b). As mentioned before, our approach aims to be conservative in the sense that we only split species when there is convincing evidence (null hypothesis of a single-species model). In the case of *M. jollyae* and *M. marohita*, the low sample size does not allow us to identify clear genomic differentiation (instead, we observe low genetic distances between the two candidates, when compared to those found among *M. gerpi* individuals), and we currently have no evidence for differentiation in any other trait (albeit this is due to lack of data). Accordingly, we also consider the description of *M. marohita* as a distinct species (based on three individuals and six loci) premature and propose its synonymisation until more evidence becomes available.
- M. murinus* (north) - *M. murinus* (central/south):** Our decision to collapse these two lineages is based on the following findings. First, although our isolation-by-distance (IBD) test is inconclusive, the plot of geographic against genetic distances reveals a relatively continuous pattern of IBD when considering comparisons within *M. murinus* (central), within *M. murinus* (north) and between the central and northern lineage (Fig. S14d), indicating that genetic distances can be explained by geographic distribution rather than a speciation event. Second, individuals at Bombetoka, which cluster phylogenetically with northern *M. murinus* (Fig. S14b), do show admixed ancestry at a variety of different K (Fig. S14c). In addition, when assuming $K=2$ (representing the hypothesis of two distinct species) we also see admixed ancestry in the central *M. murinus* individuals. Following our framework, this would already be considered sufficient

evidence for synonymization (considering that the IBD test is inconclusive) regardless of differences in climatic space or reproductive behaviour. As a side note, the mean *gdi* between the two lineages is not particularly high and not much different from mean values between *M. murinus* (central) and the southern lineages (Fig. S14e). Third, there is potentially a large sampling gap between the northern and central lineages (Fig. S14a), which needs to be addressed to exclude the possibility that they present opposite ends of a cline in character variation. Branch lengths separating the central from the northern lineage appear comparably small when considering the number of substitutions present within the central/southern clade (Fig. S14b). Finally, the observed differences in climatic space and reproductive activity can be explained by the large distribution of this species, covering almost the entire north-south axis of Madagascar. Comparing individuals from the northern end and the more southern part of this distribution inevitably results in the detection of differences in climatic space and potentially reproductive activity (which can be affected by climate). Again, this highlights the need to address sampling gaps, and it is connected to the conceptual question of reviewer #3 (see below). That is, additional lines of evidence need to be thoughtfully integrated in a case-by-case manner (e.g., by discussing the role of plasticity in explaining variation). We make this point clearer now in the manuscript (lines 655-656). For these reasons, we consider *M. murinus* a single species until more evidence becomes available to challenge this conclusion.

We made adjustments to the species-by-species accounts in the Supplementary results and discussion to make our arguments clearer (lines 458-473, 547-556, 720-738).

MINOR NOTES

- L217: I appreciate that a few genes is not ideal, but is it truly fair to call it bad taxonomic practice? Given the cost of sequencing - and the inaccessibility of it still to many folks around the globe - requiring genomic data might make taxonomy inaccessible for many researchers worldwide.

Reply: We thank the reviewer for pointing this out. We agree that keeping taxonomy accessible to researchers worldwide is of crucial importance, particularly in light of the taxonomic impediment. We also agree that using the term “bad taxonomic practice” might not be appropriate, as species descriptions or delimitation decisions can be based on genetics (rather than genomics) if the underlying geographic sampling is good and if additional lines of evidence are considered. Following the reviewer’s suggestion, we modified the respective sentence (lines 245-247).

- L293 - 296: Do the revised geographic boundaries (for the revised species boundaries) map to steep environmental gradients and rivers, as suggested here?

Reply: Indeed, environmental gradients and/or rivers seem to explain the diversification and distribution of a number of *Microcebus* species in Madagascar's humid rainforests. For example, the distribution of *M. simmonsii* (incl. *M. boraha*) seems to be limited by the Anove River in the north and the Ivondro River in the south (Fig. 11a). Similarly, the Antanaimbala likely is the northern distributional limit for *M. macarthurii* while the Anove River is the southern limit for *M. jonahi* (Fig. S13a; Schüßler et al. 2020; Poelstra et al. 2021). The exact distributional boundary between the two species is not known. Finally, the distribution of *M. gerpi* is separated from that of *M. jollyae* (incl. *M. marohita*) by the Mangoro River and an elevational gradient towards the Central Highlands (Fig. S12a; van Elst et al., 2023). However, environmental gradients and rivers also account for structure within species (van Elst et al., 2023; unpublished data). On the other hand, the distribution of *M. lehilahytsara* does not seem to be limited by riverine barriers or elevational gradients (Tiley et al., 2022). Assessing which factors represent barriers to gene flow, for which species, and why will require further investigation. In any case, while answering this comment, we reconsidered a comment made by the reviewer in the first round of revisions concerning the unreliability of diversification rate analyses on small phylogenetic scales. We went back to the modeling results of our diversification rate analysis on humid and dry habitats and found that support for an equal rates model was almost similarly high ($\Delta AIC < 2$). For this reason, we decided to remove this part from the main manuscript (lines 329-337) and, instead, mention in the Supplementary results and discussion now (lines 987-1002) that we conducted the analysis but obtain inconclusive results.

- L298: Consider briefly discussing here what morphometric traits were analyzed.

Reply: We thank the reviewer for this suggestion. We agree that this information is of potential interest to readers and added this information (lines 346-348). We added a similar mention for the climatic niche analysis (lines 363-364). We do not want to go into more detail at these positions as we believe it would inflate the section and disrupt the thread of argument.

While working on this comment, we also noticed that the Methods section still contained a remnant of an earlier version of the manuscript. More specifically, we initially conducted the modeling of morphometric diversification on two alternative subsets of morphometric variables. This was done to explore whether we would obtain congruent results when sampling different morphometric traits, which we did. We decided to not include this data exploration in the final version to keep the manuscript more concise. However, the Methods section still contained mention of this (while the

associated results were no longer part of the manuscript). We removed this mention now (lines 812-814).

- L352: Were mouse lemurs last to arrive in the communities to which they belong?

Reply: We appreciate the reviewer's note about our sentence implying mouse lemurs arrived last in their communities. As this was not our intended message, we modified the sentence (lines 401-402). Indeed, while lemurs underwent successive radiations, the diversification of distantly related lemur lineages seems to have largely overlapped in time (Everson et al., 2023). The new sentence now reads "as they radiated alongside distantly-related and larger-bodied lemur species".

- L538: did the authors test for best-fitting K ? If so, I would recommend including that in figures.

Reply: We acknowledge that assessing the best-fitting K is a standard practice in clustering analyses (Evanno et al. 2005). However, we opted not to include these analyses in the manuscript for several reasons. First, while best-fitting K metrics are valuable for population structure assessment, they do not directly allow distinguishing species from populations. Our primary focus in this study is species delimitation. Second, the ΔK metric of Evanno et al. does not account for $K=1$, the scenario where all individuals belong to a single population. In fact, the estimation of best-fitting K supports our conclusion for species delimitation in all cases except those where we synonymise candidates to a single species. Given these considerations, we prioritized presenting the core findings related to species delimitation and avoided cluttering the Supplementary Material with potentially misleading information.

Reviewer #3 (Remarks to the Author):

This is my second review of this manuscript.

In this manuscript the authors re-assess the species diversity of the genus *Microcebus* using a new and integrative framework incorporating multiple types of genomic data alongside morphological, bioclimatic, reproductive and acoustic data. As before, I'm impressed by the well-thought-out stepwise approach and accompanying flowchart, execution and communication of analyses, and the concise, clear writing. I feel this will be an important and well-cited contribution to lemur taxonomy, and is a great step in finding the right compromise between lumpers and splitters.

59I am also very impressed with the great care taken by the authors to respond to my comments, and those of the other reviewers. I appreciate both the written response, and the careful revisions made. I'm therefore left with little in terms of major comments; below I offer just one conceptual question/concern and a short list of minor comments that would improve the manuscript presentation a little.

Reply: We thank Reviewer #3 for the second careful and detailed review of our work. We think the comments so far have already improved the manuscript significantly, and addressing the points below further improved readability and the presentation of our results.

Conceptual Question (and I apologize for not raising this last time):

In considering the nicely-presented Figure 1, I'm just struck by the fact that of the 8 criteria for testing, seven are really quite intrinsic to the species (pair) – four genomic criteria plus morphometry, reproductive activity and acoustic communication. The final one, climatic niche, measures something external (extrinsic) to the species' biology, even if you could argue that it tends to cause biological adaptations to those conditions.

I acknowledge that ENM (Ecological Niche Modelling) is a really useful tool in understanding species' ecologies, and in looking at divergence among closely related species. However, Figure 1 tells us that in a situation where you have sister populations that are reciprocally monophyletic and show genetic divergence and clustering, then even when those two populations don't diverge in morphometry, reproductive activity or acoustic communication, their divergence in ecological niche would lead to confirmation as distinct species.

To perhaps be more clear: consider two test cases A and B in which the pairs had the same degree of genomic divergence (high enough to pass the four tests) but were similar in morphometry, reproductive schedule and acoustic communication. You'd determine there was a single species in pair A, where the two populations occupied similar habitats, but you'd determine there were two species in B, where the two population occupied climatically different habitats.

It occurs to me this is a way that one might erroneously recognize species where you in reality have one ecologically flexible species that has colonized multiple habitats in the relatively distant past, remained somewhat reproductive isolated, and diverged genetically. This is also a case where sampling would matter – especially if sampling didn't happen in intermediate climatic zones. And, given that species differ considerably along the generalist-specialist continuum, this seems like a concern, especially in applying this framework to other genera and orders.

The examples I'm thinking of are the aye-aye (which occupies east and west forests in Madagascar), as well as humans (perhaps an unfair example, but also a primate).

I feel the authors have thought through these issues more than me. Is the answer here that "well, if the gdi is high and other genomic separation is there, then it's valid"? Or perhaps that *Microcebus* species are more specialized than my two funny examples, so it's not a fair comparison? I just wanted to offer this comment to the authors and the editor; if I'm wrong, perhaps a brief mention of why it's valid, or the errors it could potentially lead to, might be nice.

Reply: We thank the reviewer for bringing up this important point. We agree that differentiation in ecological niche space as a criterion for species delimitation has limitations. In the following, we first comment on the two examples mentioned and then come back to the use of differentiation in ENMs for species delimitation.

Concerning the aye-aye, the east-west connectivity of its distribution seems to have been broken only very recently (Petter, 1977) and only in certain areas (Rambeloson et al., 2021), from forest loss and fragmentation. Furthermore, aye-ayes seem to have very large home ranges and can travel unexpectedly long distances per day (Sterling, 1993; Louis, Jr., unpublished data), suggesting that the broad distribution of the species has been maintained over time due to a good dispersal ability. We therefore expect the aye-aye to exhibit a neat pattern of isolation-by-distance (IBD) potentially combined with the effects of past landscape components (e.g., rivers and forest cover; these aspects were not treated in the manuscript). Unfortunately, comprehensive sampling and genomic data are lacking for the aye-aye. Notably, the niche of the aye-aye does not seem to be climatic (it was almost present everywhere in Madagascar), but rather a feeding behaviour niche, similar to that of the woodpecker (Boyer et al., 2021), absent in Madagascar (Yamagishi and Eguchi, 1996). While humans are a difficult example, bringing up a cosmopolitan species is a valid point. In humans, high dispersal abilities have been shown to translate into patterns of IBD (e.g., Relethford 2004), which would likely suggest synonymisation of "candidate species" and preclude the use of ENM.

We integrated ecological niche modelling in our framework because it can potentially serve as a proxy for biological/physiological adaptation, as the reviewer pointed out (we mention this now in lines 712-718). It is increasingly considered for that purpose (e.g., Minoli et al., 2010; Zhang et al., 2014; Dagnino et al., 2017). Many described *Microcebus* species are confined to relatively small geographic areas (i.e., micro-endemism, but see *M. murinus* and *M. lehilahytsara*), and their climatic niche may reveal how much they are adapted to certain climatic conditions. Notably, in the end, ecological niche models did not act as a decision maker for delimitation in any of the nine candidate groups but provided additional evidence to confirm/interpret decisions based on other/earlier criteria. We decided to retain it in the paper for the following reasons. First, we still consider it taxonomically informative (if integrated

carefully) and a good illustration of how to include and analyse additional lines of evidence. Second, occurrence and bioclimatic data are readily available for most study systems, which is not the case for many other lines of evidence. Finally, the problem of intraspecific variation is not confined to ecological niche but can also play a role in other traits such as morphology or reproductive activity (e.g., plasticity), but if we would like to delimit cryptic species, we eventually need to base our decision on some heuristics. These caveats certainly illustrate that it is important to carefully consider which additional lines of evidence are appropriate for species delimitation in a given system and to thoughtfully integrate them in a case-by-case manner (e.g., by discussing the role of plasticity). To emphasise this, we followed Padiál et al. (2010) and added that additional lines of evidence can be used to distinguish species if other explanations for their differentiation (e.g., plasticity or local adaptation) can be excluded (lines 656-657, 1376-1377). We also added mention of plasticity and ecological flexibility in the last paragraph of the section “An integrative framework for taxonomic re-evaluation”, where we already touched on it before (lines 259-260).

Line-by-line comments

97: “increasing unanimity” is a paradoxical term, rather like “more unique” or “less unique”. Suggest replacing with “increasing agreement”.

Reply: We thank the reviewer for pointing this out. We changed the respective wording (line 104).

112: “morphologically static” suggests all species are morphologically identical. Suggest “relatively morphologically static? Or “while showing relatively little morphological divergence”?

Reply: We thank the reviewer for this suggestion. We changed the respective wording (lines 119-120).

124: Replace “Our work does not only shed light on... provides” with “Our work sheds light on the taxonomy and diversification of the genus *Microcebus*, while also providing ...” (edit for clarity).

Reply: We thank the reviewer for this suggestion. We have changed the respective wording (lines 148-149).

131: I question whether this first paragraph of the results is best placed here. After coming through the introduction, when the reader sees “Results” she/he is ready for results. This is a rather long paragraph that feels like discussion and repeats much of what came in the introduction.

Reply: We agree with the reviewer that the paragraph felt misplaced. Our intention was to give the reader an understanding of our framework and its main concepts before diving into the Results and discussion given that the Methods are placed at the end of *Nature Ecology & Evolution* publications. As the reviewer pointed out, the information given was partially redundant to what was already described in the introduction (“Main”). Therefore, we extended the last paragraph of the introduction to present the most important ideas and concepts of our framework while not going into too much detail (reviewer #2 suggested to convey these ideas relatively early in the paper; lines 121-141) and removed the first paragraph of the Results and discussion (lines 155-183; its last sentence was added to lines 267-270).

221: I think it’s quite a useful product of this study that it “identified key areas where further sampling is necessary”. However, I don’t see it (I think) in the main text. I would prefer this to be more available in the main text – not a long treatment, but if there are 2 or 3 places (species pairs) that really could benefit from more sampling, I’d mention them right here.

Reply: We thank the reviewer for this suggestion, which we also consider an important point to make. We have added a mention of this to the main manuscript (lines 252-255).

272: should be “contrasts with”?

Reply: We thank the reviewer for pointing this out. It has been corrected (line 314).

354: I feel “also occupied” would work better than “occupied also”.

Reply: We thank the reviewer for this suggestion. The sentence has been modified following the comment on line 352 by Reviewer #2 (line 401).

356: Awkward; perhaps “may have served to intensify”? or just “intensified”?

Reply: We thank the reviewer for this suggestion. We have changed the respective wording (line 403).

357: Similarly, suggest replacing “leading to rare co-occurrence” with “resulting in low levels of co-occurrence...”

Reply: We thank the reviewer for this suggestion. We have changed the respective wording (line 404).

423: Remove “i.e.,” as this is a complete list.

Reply: We thank the reviewer for pointing this out. It has been corrected (line 462).

503: correct spelling, “lehilahytsara”

Reply: We thank the reviewer for pointing this out. The entire section has been rephrased following the first comment of Reviewer #2 (lines 552-571).

616: There is a word missing after “intraspecific”, possibly “variation” – even so, “allows predicting” is an awkward phrase too.

Reply: We thank the reviewer for pointing this out. We removed the entire sentence to make this part of the Methods more concise as the sentence did not contain much information (lines 671-673).

673: Compared with the rest of this section, this feels “rough”. We are given two sentences; the first says ‘we checked for differences’ and the second is rather unclear (should be cleaned up) but I believe it tells us that lactation and pregnancy were used to count backwards to a putative estrus date (or range of dates). I feel this is not the most important piece of the methods, but it should be buttressed a little so that we at least understand what goes into the decision of “different” or “same”.

Reply: We thank the reviewer for pointing this out and agree that this section contained insufficient information. Indeed, pregnancy and lactation were used to obtain a putative timing of oestrus. We reworded the entire section, aiming to be more comprehensive as well as precise (lines 737-751; corresponding changes were also made to the Supplementary methods at lines 1316-1330). Notably, checking for differences in reproductive activity was done qualitatively because the large variation in sampling effort/size and timing across candidates made a quantitative assessment difficult. Accordingly, in each candidate species comparison, we compare various aspects of reproductive activity (e.g., its onset or total duration) while accounting for the quality of the underlying data to conclude whether

there is evidence of differentiation or not (see species-by-species accounts in the supplementary material). For this reason, we find it difficult to present a general statement of what is considered “different” or “same”. However, we added a mention that the comparison was done in a qualitative (rather than a quantitative) manner (lines 748-751).

1326: English readers may not be universally familiar with the term “minuscule”... perhaps just “lower case letters”? If you change, change within supplemental material as well.

Reply: We thank the reviewer for this suggestion. We changed the respective wording there and in the supplementary material (lines 1395-1397; lines 967, 974 in Supplementary results and discussion).

1353: For Fig 4c, am I wrong, or is there a discrepancy between the order of colors in the plot (red, green, blue) and the order shown in the legend (blue, red, green)? Better if they match, no?

Reply: We thank the reviewer for pointing out this detail. We corrected it (line 1426).

References

Aleixo-Pais, Isa, et al. "The genetic structure of a mouse lemur living in a fragmented habitat in Northern Madagascar." *Conservation Genetics* 20 (2019): 229-243.

Boyer, Doug M., L. M. Schaeffer, and K. C. Beard. "New dentaries of Chiromyoides (Primate, Platyrrhini) and a reassessment of the “mammalian woodpecker” ecological niche." *Geobios* 66 (2021): 77-102.

Dagnino, Davide, L. Minuto, and G. Casazza. "Divergence is not enough: the use of ecological niche models for the validation of taxon boundaries." *Plant Biology* 19.6 (2017): 1003- 1011.

Evanno, Guillaume, Sebastien Regnaut, and Jérôme Goudet. "Detecting the number of clusters of individuals using the software STRUCTURE: a simulation study." *Molecular ecology* 14.8 (2005): 2611-2620.

Everson, Kathryn M., et al. "Not one, but multiple radiations underlie the biodiversity of Madagascar's endangered lemurs." *bioRxiv* (2023): 2023-04.

- Minoli, Ignacio, Mariana Morando, and Luciano Javier Avila. "Integrative taxonomy in the *Liolaemus fitzingerii* complex (Squamata: Liolaemini) based on morphological analyses and niche modeling." (2014).
- Padial, José M., et al. "The integrative future of taxonomy." *Frontiers in zoology*, 7 (2010), 1-14.
- Petter, Jean-Jacques. "The aye-aye." *Primate conservation* (1977): 37-57.
- Poelstra, Jelmer W., et al. "Cryptic patterns of speciation in cryptic primates: microendemic mouse lemurs and the multispecies coalescent." *Systematic Biology* 70.2 (2021): 203-218.
- Rambelison, Elodi, et al. "Initial Reintroduction of the Aye-Aye (*Daubentonia madagascariensis*) in Anjajavy Reserve, Northwestern Madagascar." *Folia Primatologica* 92.5-6 (2022): 284- 295.
- Relethford, John H. "Global patterns of isolation by distance based on genetic and morphological data." *Human biology* (2004): 499-513.
- Salmona, Jordi. *Comparative conservation genetics of several threatened lemur species living in fragmented environments. A glimpse through natural history of northern Madagascar lemurs*. Diss. Ph. D Thesis, 2015.
- Schüßler, Dominik, et al. "Morphological variability or inter-observer bias? A methodological toolkit to improve data quality of multi-researcher datasets for the analysis of morphological variation." *American journal of biological anthropology* 183.1 (2024): 60-78.
- Sgarlata, Gabriele Maria, et al. "Genetic and morphological diversity of mouse lemurs (*Microcebus* spp.) in northern Madagascar: The discovery of a putative new species?." *American Journal of Primatology* 81.12 (2019): e23070.
- Sterling, Eleanor J. "Patterns of range use and social organization in aye-ayes (*Daubentonia madagascariensis*) on Nosy Mangabe." *Lemur social systems and their ecological basis*. Boston, MA: Springer US, 1993. 1-10.
- Yamagishi, Satoshi, and Kazuhiro Eguchi. "Comparative foraging ecology of Madagascar vangids (Vangidae)." *Ibis* 138.2 (1996): 283-290.
- Zhang, Yueyun, et al. "Insights from ecological niche modeling on the taxonomic distinction and niche differentiation between the black-spotted and red-spotted tokay geckoes (*Gekko gecko*)." *Ecology and Evolution* 4.17 (2014): 3383-3394.

Decision Letter, second revision:

Our ref: NATECOLEVOL-23030664C

5th July 2024

Dear Dr. van Elst,

Thank you for your patience as we've prepared the guidelines for final submission of your Nature Ecology & Evolution manuscript, "An integrative and generalizable approach to elucidate cryptic diversifications sheds light on mouse lemur taxonomy and evolution" (NATECOLEVOL-23030664C). Please carefully follow the step-by-step instructions provided in the attached file, and add a response in each row of the table to indicate the changes that you have made. Please also check and comment on any additional marked-up edits we have proposed within the text. Ensuring that each point is addressed will help to ensure that your revised manuscript can be swiftly handed over to our production team.

****We would like to start working on your revised paper, with all of the requested files and forms, as soon as possible (preferably within two weeks). Please get in contact with us immediately if you anticipate it taking more than two weeks to submit these revised files.****

67In recognition of the time and expertise our reviewers provide to Nature Ecology & Evolution's editorial process, we would like to formally acknowledge their contribution to the external peer review of your manuscript entitled "An integrative and generalizable approach to elucidate cryptic diversifications sheds light on mouse lemur taxonomy and evolution". For those reviewers who give their assent, we will be publishing their names alongside the published article.

Nature Ecology & Evolution offers a Transparent Peer Review option for new original research manuscripts submitted after December 1st, 2019. As part of this initiative, we encourage our authors to support increased transparency into the peer review process by agreeing to have the reviewer comments, author rebuttal letters, and editorial decision letters published as a Supplementary item. When you submit your final files please clearly state in your cover letter whether or not you would like to participate in this initiative. Please note that failure to state your preference will result in delays in accepting your manuscript for publication.

Cover suggestions

We welcome submissions of artwork for consideration for our cover. For more information, please see our guide for cover artwork.

Nature Ecology & Evolution has now transitioned to a unified Rights Collection system which will allow our Author Services team to quickly and easily collect the rights and permissions required to publish your work. Approximately 10 days after your paper is formally accepted, you will receive an email in providing you with a link to complete the grant of rights. If your paper is eligible for Open Access, our Author Services team will also be in touch regarding any additional information that may be required to arrange payment for your article.

Please note that *Nature Ecology & Evolution* is a Transformative Journal (TJ). Authors may publish their research with us through the traditional subscription access route or make their paper immediately open access through payment of an article-processing charge (APC). Authors will not be required to make a final decision about access to their article until it has been accepted. Find out more about Transformative Journals

Authors may need to take specific actions to achieve compliance with funder and institutional open access mandates. If your research is supported by a funder that requires immediate open access (e.g. according to Plan S principles) then you should select the gold OA route, and we will direct you to the compliant route where possible. For authors selecting the subscription publication route, the journal's standard licensing terms will need to be accepted, including <https://www.nature.com/nature-portfolio/editorial-policies/self-archiving-and-license-to-publish>. Those licensing terms will supersede any other terms that the author or any third party may assert apply to any version of the manuscript.

[REDACTED]

[REDACTED]

Reviewer #2:

Remarks to the Author:

This manuscript proposes a framework for delimiting species by integrating genetic and phenotypic data. The approach emphasizes identifying geographic discontinuities in differentiation; these discontinuities likely indicate species boundaries. The authors apply this framework to revise the taxonomy of the mouse lemurs of Madagascar and then use this new taxonomy to understand the diversification of the mouse lemurs.

I served as a reviewer of the original submission and the revision, and I find this second revision has addressed all my concerns. In particular, even though I disagree with the authors on some of their points, their writing clearly presents their logic, and their logic is consistent throughout. I commend the authors for presenting such well-considered arguments and for being so amenable to discussion and revision through the review process.

My only lingering comment: the authors discuss in their rebuttal that they are aiming to infer a "conservative" taxonomy. I understand their rationale. I think it would be helpful to briefly state this in

69the opening paragraphs of the paper so that readers can understand the decision to synonymize in some of the more complicated cases. The language used in the rebuttal to this point was clear -- and perhaps can be simply copied-and-pasted into the main text?

I am glad to have reviewed this paper - it challenged me and I learned a lot!

Reviewer #3:

Remarks to the Author:

This is my third review of this manuscript. I continue to be impressed with the quality of the manuscript, and the thoughtful responses to reviewers.

The only substantial issue I raised in my second review was the fact that climatic niche is essentially an extrinsic, rather than an intrinsic, characteristic of a species candidate – and the issue of whether this could lead to “confirmation” for a species pair that is really a more generalist species able to live in differing climates. I am satisfied with the authors’ response to this. It would seem odd to leave it out altogether, and the fact that it’s embedded with 7 other types of tests means it’s unlikely to pose a problem – and in the case where that variable is the one that makes the difference, researchers can examine that issue separately if desired.

Beyond that, I struggled to find any other faults, and I offer only two suggestions to improve clarity. The authors have produced a well-justified and well-written manuscript, which is remarkable given how much ground it covers, and I think it will have a meaningful impact on the field.

498: replace “the species” with “this species” to make more clear that you’re referring to *M. tavaratra*?

511: replace “species are often structured in space...” with “species often show spatial patterning of variation, which can confound species delimitation if ignored or not represented adequately in the sampling, we first tested... (otherwise the second part of the sentence refers to a noun - the spatial patterning - which wasn’t introduced as such in the first part).

Author Rebuttal, second revision:

Reviewer #2:

Remarks to the Author:

This manuscript proposes a framework for delimiting species by integrating genetic and phenotypic data. The approach emphasizes identifying geographic discontinuities in differentiation; these discontinuities likely indicate species boundaries. The authors apply this framework to revise the taxonomy of the mouse lemurs of Madagascar and then use this new taxonomy to understand the diversification of the mouse lemurs.

I served as a reviewer of the original submission and the revision, and I find this second revision has addressed all my concerns. In particular, even though I disagree with the authors on some of their points, their writing clearly presents their logic, and their logic is consistent throughout. I commend the authors for presenting such well-considered arguments and for being so amenable to discussion and revision through the review process.

My only lingering comment: the authors discuss in their rebuttal that they are aiming to infer a "conservative" taxonomy. I understand their rationale. I think it would be helpful to briefly state this in the opening paragraphs of the paper so that readers can understand the decision to synonymize in some of the more complicated cases. The language used in the rebuttal to this point was clear -- and perhaps can be simply copied-and-pasted into the main text?

Reply: We thank the reviewer for this suggestion and added mention of this to the end of the Introduction (lines 140-142).

Reviewer #3:

Remarks to the Author:

This is my third review of this manuscript. I continue to be impressed with the quality of the manuscript, and the thoughtful responses to reviewers.

The only substantial issue I raised in my second review was the fact that climatic niche is essentially an extrinsic, rather than an intrinsic, characteristic of a species candidate – and the issue of whether this could lead to “confirmation” for a species pair that is really a more generalist species able to live in differing climates. I am satisfied with the authors’ response to this. It would seem odd to leave it out altogether, and the fact that it’s embedded with 7 other types of tests means it’s unlikely to pose a problem – and in the case where that variable is the one that makes the difference, researchers can examine that issue separately if desired.

Beyond that, I struggled to find any other faults, and I offer only two suggestions to improve clarity. The authors have produced a well-justified and well-written manuscript, which is remarkable given how much ground it covers, and I think it will have a meaningful impact on the field.

498: replace “the species” with “this species” to make more clear that you’re referring to *M. tavaratra*?

Reply: We thank the reviewer for this suggestion and edited the text accordingly (line 515).

511: replace “species are often structured in space...” with “species often show spatial patterning of variation, which can confound species delimitation if ignored or not represented adequately in the sampling, we first tested... (otherwise the second part of the sentence refers to a noun - the spatial patterning - which wasn’t introduced as such in the first part).

Reply: We thank the reviewer for this suggestion and edited the text accordingly (lines 528-530).

References

Kappeler, Peter M., et al. "Complex social and political factors threaten the world's smallest primate with extinction." *Conservation Science and Practice* 4.9 (2022): e12776.

Final Decision Letter:

23rd August 2024

Dear Tobias,

We are pleased to inform you that your Article entitled "Integrative taxonomy clarifies the evolution of a cryptic primate clade", has now been accepted for publication in *Nature Ecology & Evolution*.

Over the next few weeks, your paper will be copyedited to ensure that it conforms to *Nature Ecology and Evolution* style. Once your paper is typeset, you will receive an email with a link to choose the appropriate publishing options for your paper and our Author Services team will be in touch regarding any additional information that may be required

Due to the importance of these deadlines, we ask you please us know now whether you will be difficult to contact over the next month. If this is the case, we ask you provide us with the contact information (email, phone and fax) of someone who will be able to check the proofs on your behalf, and who will be available to address any last-minute problems . Once your paper has been scheduled for online publication, the Nature press office will be in touch to confirm the details.

Acceptance of your manuscript is conditional on all authors' agreement with our publication policies (see www.nature.com/authors/policies/index.html). In particular your manuscript must not be published elsewhere and there must be no announcement of the work to any media outlet until the publication date (the day on which it is uploaded onto our web site).

Please note that *Nature Ecology & Evolution* is a Transformative Journal (TJ). Authors may publish their research with us through the traditional subscription access route or make their paper immediately open access through payment of an article-processing charge (APC). Authors will not be required to make a final decision about access to their article until it has been accepted. Find out more about Transformative Journals

73Authors may need to take specific actions to achieve compliance with funder and institutional open access mandates. If your research is supported by a funder that requires immediate open access (e.g. according to Plan S principles) then you should select the gold OA route, and we will direct you to the compliant route where possible. For authors selecting the subscription publication route, the journal's standard licensing terms will need to be accepted, including <https://www.nature.com/nature-portfolio/editorial-policies/self-archiving-and-license-to-publish>. Those licensing terms will supersede any other terms that the author or any third party may assert apply to any version of the manuscript.

We welcome the submission of potential cover material (including a short caption of around 40 words) related to your manuscript; suggestions should be sent to Nature Ecology & Evolution as electronic files (the image should be 300 dpi at 210 x 297 mm in either TIFF or JPEG format). Please note that such pictures should be selected more for their aesthetic appeal than for their scientific content, and that colour images work better than black and white or grayscale images. Please do not try to design a cover with the Nature Ecology & Evolution logo etc., and please do not submit composites of images related to your work. I am sure you will understand that we cannot make any promise as to whether any of your suggestions might be selected for the cover of the journal.

You can generate the link yourself when you receive your article DOI by entering it here: <http://authors.springernature.com/share>.

[REDACTED]

P.S. Click on the following link if you would like to recommend Nature Ecology & Evolution to your librarian <http://www.nature.com/subscriptions/recommend.html#forms>

** Visit the Springer Nature Editorial and Publishing website at www.springernature.com/editorial-and-publishing-jobs for more information about our career opportunities. If you have any questions please click here.**